# Oxidative stress-induced FABP5 S-glutathionylation protects against acute lung injury by suppressing inflammation in macrophages

Yuxian Guo[1,2,7], Yaru Liu[1,7], Shihao Zhao[1], Wangting Xu[1], Yiqing Li[1], Pengwei Zhao[3], Di Wang [2], Hongqiang Cheng [1,4], Yuehai Ke [1,5,6✉] & Xue Zhang [1,5✉]

Oxidative stress contributes to the pathogenesis of acute lung injury. Protein S-glutathionylation plays an important role in cellular antioxidant defense. Here we report that the expression of deglutathionylation enzyme Grx1 is decreased in the lungs of acute lung injury mice. The acute lung injury induced by hyperoxia or LPS is significantly relieved in Grx1 KO and Grx1[fl/fl]LysM[cre] mice, confirming the protective role of Grx1-regulated S-glutathionylation in macrophages. Using a quantitative redox proteomics approach, we show that FABP5 is susceptible to S-glutathionylation under oxidative conditions. S-glutathionylation of Cys127 in FABP5 promotes its fatty acid binding ability and nuclear translocation. Further results indicate S-glutathionylation promotes the interaction of FABP5 and PPARβ/δ, activates PPARβ/δ target genes and suppresses the LPS-induced inflammation in macrophages. Our study reveals a molecular mechanism through which FABP5 S-glutathionylation regulates macrophage inflammation in the pathogenesis of acute lung injury.

[1] Department of Pathology and Pathophysiology and Sir Run Run Shaw Hospital, Zhejiang University School of Medicine, 310058 Hangzhou, P.R. China. [2] Institute of Immunology, Zhejiang University School of Medicine, 310058 Hangzhou, P.R. China. [3] Department of Biochemistry, Zhejiang University School of Medicine, 310058 Hangzhou, P.R. China. [4] Department of Cardiology of Second Affiliated Hospital, Zhejiang University School of Medicine, 310058 Hangzhou, P.R. China. [5] Department of Respiratory Medicine of Sir Run Run Shaw Hospital, Zhejiang University School of Medicine, 310058 Hangzhou, P.R. China. [6] Zhejiang Laboratory for Systems and Precision Medicine, Zhejiang University Medical Center, 311121 Hangzhou, China. [7] These authors contributed equally: Yuxian Guo, Yaru Liu. ✉email: yke@zju.edu.cn; zhangxue@zju.edu.cn

Oxidative stress is a result of imbalance between the generation of reactive oxygen species (ROS) and the antioxidant defense systems, leading to cellular damage. It has been found to be associated with the initiation and progression of a variety of human diseases[1]. Redox regulation of cell function often involves the conversion of reactive thiols on specific cysteine residues from reduced to oxidized forms. The major types of thiol modification that have been shown to play an important redox-dependent role include protein S-glutathionylation, sulfenic acid formation, nitrosylation, and disulfide bond formation[2].

In many types of cells, oxidative stress has been shown to drive the S-glutathionylation of free thiol groups (-SH) on the cysteine residues of proteins to form protein-glutathione mixed disulfide adducts (Pr-SSG). Protein S-glutathionylation regulates the structure and function of target proteins, including actin, Ras, integrins, transcription factors (NF-κB and AKT)[3–6], and metabolic enzymes (GAPDH, succinate dehydrogenase, and pyruvate kinase)[7–10]. In a recent study, we revealed that cysteine S-glutathionylation of the highly-conserved Cys188 residue of IL-1β positively regulates its bioactivity by preventing its irreversible ROS-elicited deactivation[11].

Protein S-glutathionylation is a dynamic reversible process. Deglutathionylation is mainly catalyzed by glutaredoxin (Grx), a thiol disulfide oxido-reductase (thioltransferase)[12]. Mammalian cells encode two principal dithiol isoforms of Grx: Grx1 mainly exists in the cytoplasm[13], and Grx2 localizes to mitochondria or the nucleus depending on gene-splicing[14]. Grx1 is involved in the regulation of various cellular functions, maintains a stable cellular redox state, and serves to maintain a reducing environment within the cell under varying environmental conditions. Grx1-regulated protein S-glutathionylation has been implicated in the pathogenesis of many diseases including neurodegenerative disorders[15], cardiovascular diseases[16,17], and several types of cancer[18]. It is becoming increasingly evident that protein S-glutathionylation and Grx1 perform a wide range of antioxidant, anti-inflammatory, and anti-apoptotic functions in the body, participating in acute and chronic inflammatory responses[19,20].

In many inflammatory lung diseases, the importance of protein S-glutathionylation and Grx1 has been investigated. Grx1 expression in the lungs has been found to be predominantly localized in macrophages and bronchial epithelium[21]. In lung epithelial cells, cigarette smoke extract downregulates the Grx1 level[22], and smoke exposure in mice increases protein S-glutathionylation in lavaged cells and lavage fluid (BALF)[21]. Grx1 expression is increased in a mouse model of allergic airway disease[23]. In patients with COPD, Grx1 is decreased and the number of Grx1-positive macrophages is positively correlated with lung function[24].

Recent studies have shown that protein S-glutathionylation and Grx1 are closely associated with acute lung injury (ALI)[19,25]. Nonetheless, the mechanism underlying the Grx1-driven regulation and alleviation of acute lung injury remains unclear, and the precise molecular targets of Grx1 in its pathogenesis have yet to be elucidated. For this purpose, we used a recently-developed redox proteomic strategy for site-specific identification and quantification of S-glutathionylation to determine the proteins in macrophages that are susceptible to redox-dependent regulation. The S-glutathionylation identified a broad set of proteins and specific cystine residues which correlated well with cellular metabolic processes. FABP5 was among the top S-glutathionylation candidates revealed by liquid chromatography with tandem mass spectrometry (LC-MS/MS) and validated using co-immunoprecipitation (Co-IP).

Fatty acid-binding proteins (FABPs) are a class of small molecular mass intracellular lipid chaperones that bind to hydrophobic ligands such as long-chain fatty acids, as well as transporting them and related lipids to specific cellular compartments, including the nucleus, mitochondria, endoplasmic reticulum, and peroxisomes[26–29]. FABP5 is abundantly expressed in keratinocytes, adipocytes, macrophages, T cells, and cancer cells[18,26,30,31]. It plays a key role in lipid-related metabolic processes, cell signaling, cell growth, and differentiation, as well as regulating inflammation[29,32]. FABP5 is translocated into the nucleus in response to retinoic acid (RA), and it enhances RA-induced, PPARβ/δ-mediated transcriptional activation[28]. In recent studies, FABP5 has been shown to play a protective role during bacterial infection of the lung by increasing PPAR-γ activity[33]. FABP5−/− mice present increased numbers of inflammatory cells and lung damage during H1N1 influenza A infection[34].

Here, we report that conditional macrophage-specific genetic deletion of Grx1 confers protection against acute lung injury in mice. LC-MS/MS and Co-IP assays indicated that FABP5 is a major target of Grx1 and is susceptible to redox-dependent regulation in response to oxidative stress. We found that S-glutathionylation promotes the fatty-acid binding and nuclear translocation of FABP5, activating PPARβ/δ and suppressing macrophage inflammation. Collectively, the S-glutathionylation of FABP5 serves as an anti-inflammatory player in response to oxidative stress during acute lung injury, providing a potential therapeutic target for its treatment.

## Results

**S-glutathionylation and Grx1 play a regulatory role in acute lung injury.** Recent studies have revealed that Grx1 regulates protein S-glutathionylation and LPS-induced acute lung injury[19]. To confirm the role of Grx1-regulated S-glutathionylation in acute lung injury, hyperoxia-induced acute lung injury mouse model was used. Hyperoxic acute lung injury is a prominent clinical concern with the increasing use of oxygen therapy. Accumulating evidence indicates that oxidative stress plays an important role in the pathogenesis of hyperoxia-induced acute lung injury[35] although the precise mechanism remains unclear. C57BL/6 mice were randomly grouped and exposed to either normoxia (NOX, 21% $O_2$) or hyperoxia (HOX, 85% $O_2$). The ratio of reduced GSH to oxidized GSH (GSSG) provides a reliable estimate of cellular redox status[36]. A decrease in the GSH/GSSG ratio was found in BALF from mice exposed to HOX (Fig. 1a). Compared with the NOX group, a higher expression level of GSH synthetase (*Gss*) was found in the HOX group (Fig. 1b). These data confirm the association between redox regulation and hyperoxic acute lung injury. Next, key enzymes involved in protein S-glutathionylation were analyzed, the *Glrx* (Grx1) expression level was significantly decreased in the HOX group (Fig. 1c, Supplementary Fig. 1a), whereas *Glrx2* (Grx2), *Txn* (TXN, Thioredoxin), and *Gsto1-1* (GSTO1-1, Glutathione-S-transferase omega 1-1) remained unchanged (Supplementary Fig. 1b). Consistently, Grx1 enzymatic activity was decreased after exposure to HOX (Fig. 1d, Supplementary Fig. 1c). Similar results were obtained in lung tissue after long-time hyperoxia exposure (Supplementary Fig. 1d, e). These findings indicate participation of Grx1 in the pathogenesis of hyperoxia-induced lung injury.

Then, Grx1 KO mice were used to explore the role of S-glutathionylation in the pathogenesis of hyperoxia-induced lung injury. Loss of Grx1 significantly relieved the lung damage caused by hyperoxia (Fig. 1e, f). A similar effect was also found after longer hyperoxia exposure (Supplementary Fig. 1f, g). Leukocytes (Fig. 1g), cytokines (Fig. 1h, Supplementary Fig. 1h) were also significantly reduced in hyperoxia-exposed Grx1 KO mice. We further measured the level of protein S-glutathionylation in Grx1 KO mice induced by

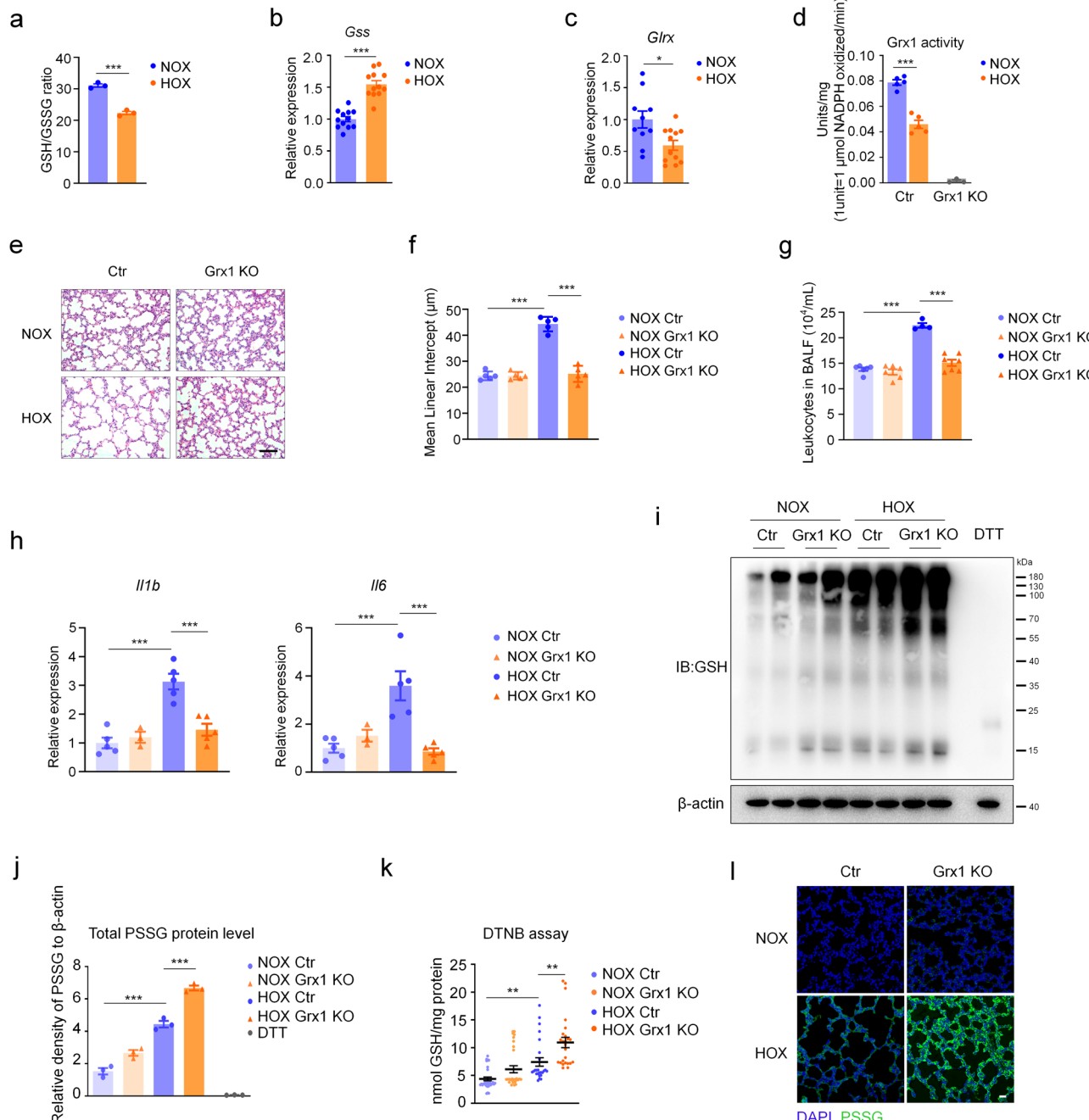

**Fig. 1 Grx1-regulated protein S-glutathionylation alleviates hyperoxia-induced acute lung injury. a** GSH/GSSG ratio in BALF from mice exposed to NOX (*n* = 3) or HOX (*n* = 3), *P* = 0.0003. **b** mRNA levels of *Gss* in lung tissue harvested after exposure to NOX (*n* = 12) or HOX (*n* = 12) as determined by qPCR, *P* < 0.0001. **c** *Glrx* mRNA expression in lung tissue from mice exposed to NOX (*n* = 10) or HOX (*n* = 12), *P* = 0.0125. **d** Grx1 activity (reaction mixtures containing L-CySSG) in lung tissue after exposure to NOX (*n* = 5) or HOX (*n* = 5), negative control (Grx1 KO mice, *n* = 3), $P_{(NOX\ ctr\ vs.\ HOX\ ctr)}$ <0.0001. **e** H&E staining of mice lungs exposed to NOX or HOX (scale bars, 100 μm). **f** Morphometric analyses of lung by mean linear intercept (MLI) in all groups as in (**e**), *n* = 5 in each group, *** *P* < 0.0001. **g** Total numbers of leukocytes in BALF from mice exposed to NOX or HOX, *n* = 6, 6, 4, 7, respectively, *** *P* < 0.0001. **h** mRNA levels of *Il1b* and *Il6* in lung tissue from mice exposed to NOX or HOX as determined by qPCR, *n* = 5, 3, 5, 5, respectively, $P_{(Il1b,\ NOX\ ctr\ vs.\ HOX\ ctr)}$ < 0.0001, $P_{(Il1b,\ HOX\ ctr\ vs.\ HOX\ Grx1\ KO)}$ = 0.0004, $P_{(Il6,\ NOX\ ctr\ vs.\ HOX\ ctr)}$ = 0.0007, $P_{(Il6,\ HOX\ ctr\ vs.\ HOX\ Grx1\ KO)}$ = 0.0004. **i** Protein S-glutathionylation in lung tissue from mice exposed to NOX or HOX *via* non-reducing Western blot (DTT, negative control; β-actin, loading control). **j** Relative expression of protein S-glutathionylation in lung tissue as in (**i**). Optical density is normalized by β-actin, *n* = 3 in each group, ***P* < 0.0001. **k** Protein S-glutathionylation quantified by DTNB assay in lung tissue from mice exposed to NOX or HOX (data shown in nmol GSH/mg protein), *n* = 35, 30, 25, 25, respectively, $P_{(NOX\ ctr\ vs.\ HOX\ ctr)}$ = 0.0065, $P_{(HOX\ ctr\ vs.\ HOX\ Grx1\ KO)}$ = 0.0029. **l** Lung tissue sections from mice exposed to NOX or HOX stained for S-glutathionylated proteins (green). DAPI (blue), Scale bars, 20 μm. All samples were biologically independent and three or more independent experiments were performed. All quantitative data are shown as mean ± SEM and analyzed with a 95% confidence interval. Two-tailed unpaired Student's *t* test for (**a–c**); one-way ANOVA followed by Tukey's post-hoc test for (**d**, **f**, **g**, **h**, **j**, **k**). **P* < 0.05, ***P* < 0.01, ****P* < 0.001. Source data are provided as a Source Data file.

hyperoxia. Western blots with anti-GSH (Fig. 1i, j) and quantitative DTNB [5, 5-dithiobis-(2-nitrobenzoic acid)] enzymatic recycling assays (Fig. 1k) revealed that Grx1 deletion obviously enhanced the overall protein S-glutathionylation in lung tissue after exposure to hyperoxia. As expected, in Grx1 KO lungs, robust elevation in protein S-glutathionylation occurred after hyperoxia exposure in situ, compared to controls (Fig. 1l). These findings strongly suggest the importance of Grx1 and S-glutathionylation in hyperoxia-induced lung injury. We consider that the reduced *Glrx* expression and Grx1 activity in the mouse lungs is one of the mechanisms of protection against oxidative stress.

**Macrophage-specific Grx1 deficiency alleviates the inflammation of acute lung injury.** Numerous cell types contribute to lung structures and functions[37,38]. We next explored which type of cell is more sensitive to oxidative stress during the pathogenesis of acute lung injury. First, the intracellular ROS level was monitored using 2′, 7′-dichlorofluorescein diacetate (DCFH-DA) as the fluorescent probe. Compared to other types of cells, the total ROS generation was markedly increased in macrophages upon hyperoxia exposure (Fig. 2a, Supplementary Fig. 2a, c–g). The GSH/GSSG ratio in macrophages significantly decreased after exposure to hyperoxia (Fig. 2b, Supplementary Fig. 3a–d). In addition, real-time quantitative PCR (qPCR) and immuno-fluorescence staining showed that the Grx1 expression level was markedly decreased in hyperoxia-exposed alveolar macrophages (AMs) (Fig. 2c, d). Moreover, Grx1 deletion remarkably increased the protein S-glutathionytion level in AMs after exposure to hyperoxia (Fig. 2e). Based on the above results, we hypothesize that Grx1 regulated S-glutathionylation in macrophages plays a key role in the pathogenesis of acute lung injury.

To further explore the function of Grx1 in macrophages in the regulation of acute lung injury pathogenesis in vivo, mice with conditional macrophage-specific genetic deletion of Grx1 (Grx1[fl/fl]LysM[cre]) and their wild-type littermate controls (Grx1[fl/fl]) were generated. Lipopolysaccharide (LPS), a characteristic component of the outer membrane of Gram-negative bacteria, was used to induce a mouse model of acute lung injury. LPS is also known to induce oxidative responses[39]. The ROS generation in macrophages was notably increased after treatment with LPS (Fig. 2f, Supplementary Fig. 2b). Histological analyses revealed that LPS-induced pulmonary inflammation was significantly suppressed in Grx1[fl/fl]LysM[cre] mice (Fig. 2g). This protective effect was further confirmed by the reduced leukocytes in BALF (Fig. 2h). The expression levels of pro-inflammatory cytokines were also significantly decreased in BALF (Fig. 2i) and lung tissue (Supplementary Fig. 3e) from Grx1[fl/fl]LysM[cre] mice when exposed to LPS. These data demonstrate that Grx1-regulated protein S-glutathionylation in macrophages plays a protective role in acute lung injury.

**Mass-spectroscopic quantification and functional analysis of ROS-induced cysteine S-glutathionylation in macrophages.** Our above results reveal that Grx1-regulated protein S-glutathionylation in macrophages plays a protective role in acute lung injury. S-glutathionylation is an important regulatory post-translational modification of protein cysteine (Cys) thiols, yet the role of specific cysteine residues as targets of modification is poorly understood. To identify the specific molecular targets and pathways of Grx1 that are susceptible to redox-dependent regulation in the pathogenesis of acute lung injury, we applied a quantitative redox proteomics approach that permits site-specific profiling of S-glutathionylation at a proteome-wide scale in macrophages[40–42] (Fig. 3a). Hydrogen peroxide ($H_2O_2$) has been applied as an oxidative stress inducer for in vitro studies. Bone marrow-derived macrophages (BMDMs) were exposed to 200 μM

$H_2O_2$ for 15 min, then the enrichment and quantitative analysis of glutathionylated proteins were performed. Gene Ontology (GO) analysis was used to obtain a picture of the biological processes that are potentially regulated by S-glutathionylation. As shown in Fig. 3b, the S-glutathionylated proteins were broadly distributed across major cellular processes, metabolic processes, and single-organism processes. Among the top five glutathiony-lated proteins in metabolic processes (Fig. 3c, Supplementary Data 2), the expression of FABP5 in macrophages was markedly higher than other proteins (Fig. 3d). Taken together, these data suggest that the S-glutathionylation of FABP5 participates in the pathogenesis of acute lung injury by regulating macrophages.

**FABP5 is susceptible to S-glutathionylation in response to ROS.** Next, we confirmed in vitro the increased S-glutathionylation of FABP5 found with MS. Compared to other FABP family members, the abundance of *Fabp5* transcripts was extremely high (Fig. 4a, Supplementary Fig. 4a). And there was no compensatory increase in the expression of FABPs in FABP5 KO BMDMs and FABP5[fl/fl]LysM[cre] BMDMs (Supplementary Fig. 4b–d). Western blot analysis showed that the expression level of FABP5 in macrophages was strikingly higher than in other types of cells, including endothelial cells, lung cancer cells, epithelial cells, fibroblasts, and neutrophils (Fig. 4b). However, after $H_2O_2$ exposure, the protein and RNA levels of FABP5 remained almost unchanged (Fig. 4c, d). In contrast, *Fabp3*, *Fabp4*, and *Fabp7* expression were markedly increased in response to $H_2O_2$ (Supplementary Fig. 4e). Based on the above results, we speculate that FABP5 may regulate the function of macrophages through other mechanisms, such as post-translational modifications, without changing its own expression.

Then, FABP5 S-glutathionylation was evaluated by Co-IP with anti-GSH antibody in COS-7 cells overexpressing FABP5 and Grx1. Our results revealed a significant enhancement of FABP5 S-glutathionylation in response to $H_2O_2$, accompanied by dimer and oligomer formation. In addition, Grx1 prevented the increase in $H_2O_2$-induced S-glutathionylation of FABP5 (Fig. 4e). Consistently, Grx1 deficiency in BMDMs amplified the S-glutathionylation of FABP5 (Fig. 4f). In the Co-IP results, we noted that dimers and oligomers of FABP5-SSG were easily detected (Fig. 4f). We hypothesized that the S-glutathionylation of FABP5 promotes the formation of dimers and oligomers, and immunoprecipitation with anti-GSH antibody facilitates the effects by enriching glutathiony-lated proteins. To test our hypothesis, the dimers and oligomers of FABP5 in macrophages were assessed under reducing and non-reducing conditions. Western blot results showed that the dimers and oligomers of FABP5 were gradually increased in a concentration-dependent manner upon $H_2O_2$ exposure in macrophages (Fig. 4g). Similar effects were revealed in COS-7 cells overexpressing FABP5 (Fig. 4h). In addition, we detected an increased S-glutathionylation signal for LPS treatment in BMDMs (Fig. 4i). As expected, elevations in FABP5 S-glutathionylation of lung tissue induced by LPS were more pronounced in Grx1[fl/fl]LysM[cre] mice (Fig. 4j). Overall, these data indicate that FABP5 in macrophages is S-glutathionylated in response to oxidative stress.

**S-glutathionylation facilitates the nuclear translocation and fatty acid binding of FABP5.** Previous studies have demonstrated that FABP5 plays an important role in trafficking fatty acids and related lipids to specific cellular compartments[26–29]. Both saturated and unsaturated long-chain fatty acids (SLCFAs and ULCFAs) bind to FABP5, but play different roles. For example, while SLCFAs block FABP5 and inhibit peroxisome proliferator-activated receptor (PPARβ/δ), ULCFAs are delivered

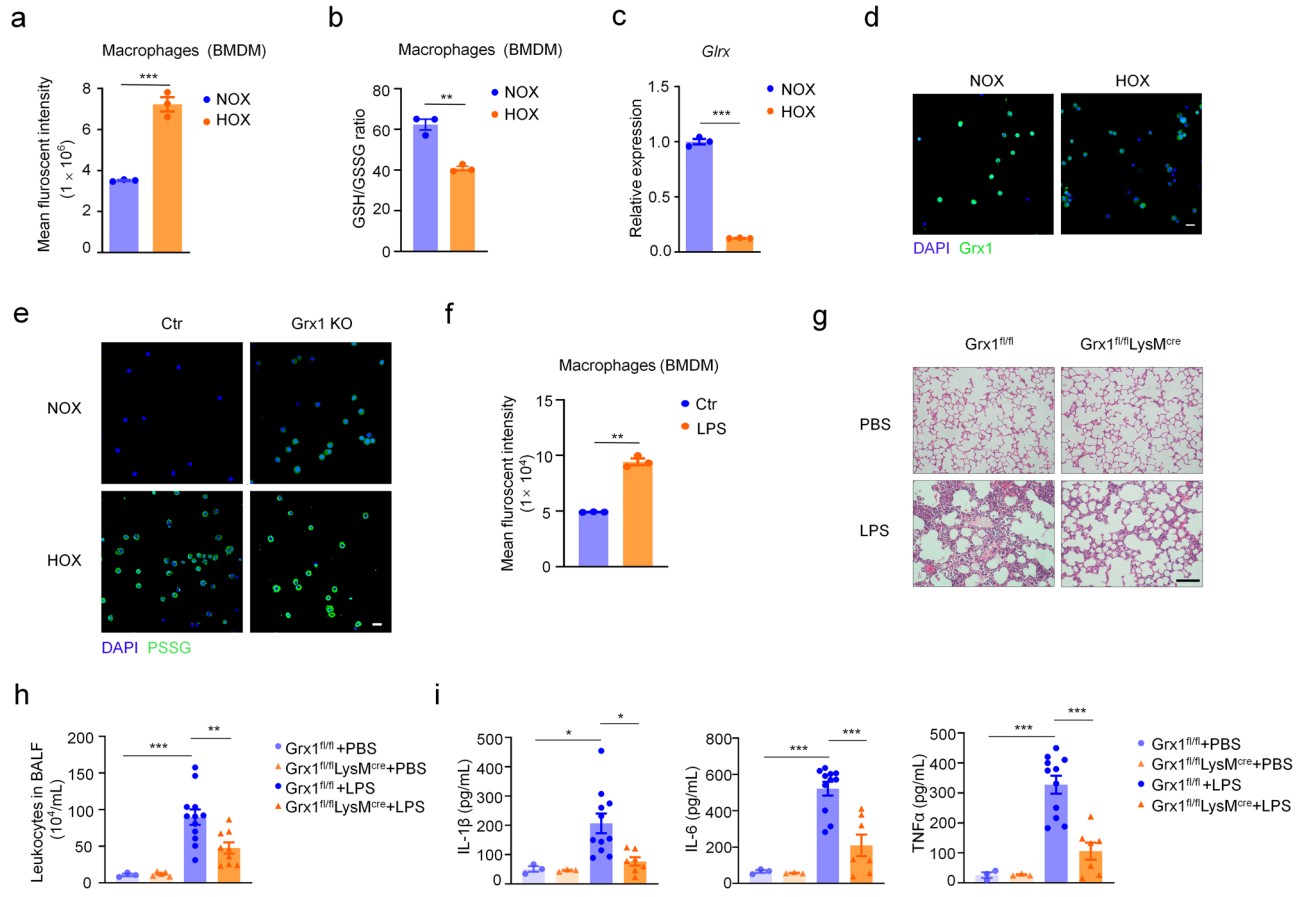

**Fig. 2 Loss of Grx1 in macrophages relieves the inflammation of acute lung injury. a** Flow cytometric analysis of intracellular ROS levels in BMDMs after culture under NOX ($n = 3$) or HOX ($n = 3$) for 15 min, $P = 0.0004$. **b** GSH/GSSG ratio in BMDMs after culture in NOX ($n = 3$) or HOX ($n = 3$) for 2 h, $P = 0.0017$. **c** *Glrx* mRNA expression of AMs from WT mice exposed to NOX ($n = 3$) or HOX ($n = 3$) as evaluated by qPCR, $P < 0.0001$. **d** Immunofluorescence staining of AMs from WT mice exposed to NOX or HOX (green, Grx1; blue, DAPI; scale bars, 40 μm). **e** Representative confocal microscopy images of AMs from control or Grx1 KO mice exposed to NOX or HOX stained for S-glutathionylated proteins (green) using Grx1-catalyzed cysteine derivatization (nuclei are stained blue with DAPI; scale bars, 40 μm). **f** Representative flow cytometry quantification of ROS generation in BMDMs stimulated with LPS (100 ng/mL, 5 min), $n = 3$ in each group, $P < 0.0001$. **g** H&E staining of mouse lung sections 24 h following intratracheal administration of PBS or LPS (5 mg/kg) (scale bars, 100 μm). **h** Total numbers of leukocytes in BALF from mice 24 h after intratracheal administration of PBS or LPS (5 mg/kg), $n = 3, 5, 12, 9$, respectively, $P_{(Grx1fl/fl\ +PBS\ vs.\ Grx1fl/fl\ +LPS)} = 0.0008$, $P_{(Grx1fl/fl\ +LPS\ vs.\ Grx1fl/fl\ LysMcre+LPS)} = 0.009$. **i** Levels of IL-1β, IL-6, and TNF-α in BALF from mice 24 h after intratracheal administration of PBS or LPS (5 mg/kg) as determined by ELISA, $n = 3, 3, 11, 7$, respectively, $P_{(Il-1β,\ Grx1fl/fl\ +PBS\ vs.\ Grx1fl/fl\ +LPS)} = 0.0388$, $P_{(Il-1β,\ Grx1fl/fl\ +LPS\ vs.\ Grx1fl/fl\ LysMcre+LPS)} = 0.018$, $P_{(Il-6,\ Grx1fl/fl\ +PBS\ vs.\ Grx1fl/fl\ +LPS)} = <0.0001$, $P_{(Il-6,\ Grx1fl/fl\ +LPS\ vs.\ Grx1fl/fl\ LysMcre+LPS)} = 0.0002$, $P_{(TNFα,\ Grx1fl/fl\ +PBS\ vs.\ Grx1fl/fl\ +LPS)} < 0.0001$, $P_{(TNFα,\ Grx1fl/fl\ +LPS\ vs.\ Grx1fl/fl\ LysMcre+LPS)} = <0.0001$. All samples were biologically independent and three or more independent experiments were performed. All quantitative data are shown as mean ± SEM and analyzed with a 95% confidence interval. Two-tailed unpaired Student's *t* test for (**a–c**, **f**); one-way ANOVA followed by Tukey's post-hoc test for (**h**, **i**). *$P < 0.05$, **$P < 0.01$, ***$P < 0.001$. Source data are provided as a Source Data file.

by FABP5 to PPARβ/δ to induce its activation[43]. First of all, we investigated whether S-glutathionylation influenced the fatty-acid binding of FABP5. To specifically detect the FABP5 protein binding with fatty acids, biotin-labeled fatty acid was used for immunoprecipitation and captured with streptavidin magnetic beads. Purified recombinant FABP5 protein was incubated with GSH plus $H_2O_2$ or GSSG for 15 min to promote S-glutathionylation. The FABP5 inhibitor BMS309403 has been reported to interact with the fatty-acid-binding pocket within the interior of the protein and competitively inhibits the binding of fatty acids[44,45]. Thus, BMS309403 and 100-fold excess of unlabeled fatty acid were used to compete with biotin-labeled fatty acid. In the presence of GSH plus $H_2O_2$ or GSSG, FABP5 exhibited a much stronger ability to bind to fatty acid (Fig. 5a, b) than untreated controls. In addition, an FABP5 inhibitor, unlabeled fatty acid, and Grx1 protein completely blocked the binding of FABP5 and biotin-labeled fatty acid, indicating that

S-glutathionylation promotes the fatty acid binding of FABP5 (Fig. 5a, c). However, the SLCFAs and ULCFAs showed the same pattern of FABP5 binding, and there was almost no difference between the two groups.

As the main cytoplasmic lipid chaperones, FABPs are known to facilitate the transport and targeting of fatty acids to intracellular sites of metabolism. Next, we examined the subcellular location of FABP5 under oxidative stress. After exposure to $H_2O_2$ for 30 min, increased nuclear accumulation of FABP5 in BMDMs was detected by immunostaining. Moreover, Grx1 deficiency clearly enhanced the nuclear translocation of FABP5, suggesting the importance of S-glutathionylation in FABP5 nuclear accumulation (Fig. 5d, e). To confirm our findings, COS-7 cells were fractionated into cytoplasmic and nuclear components and assessed for relative amounts of FABP5. $H_2O_2$ exposure resulted in a shift of FABP5 into the nucleus, and greater nuclear accumulation occurred in COS-7 cells

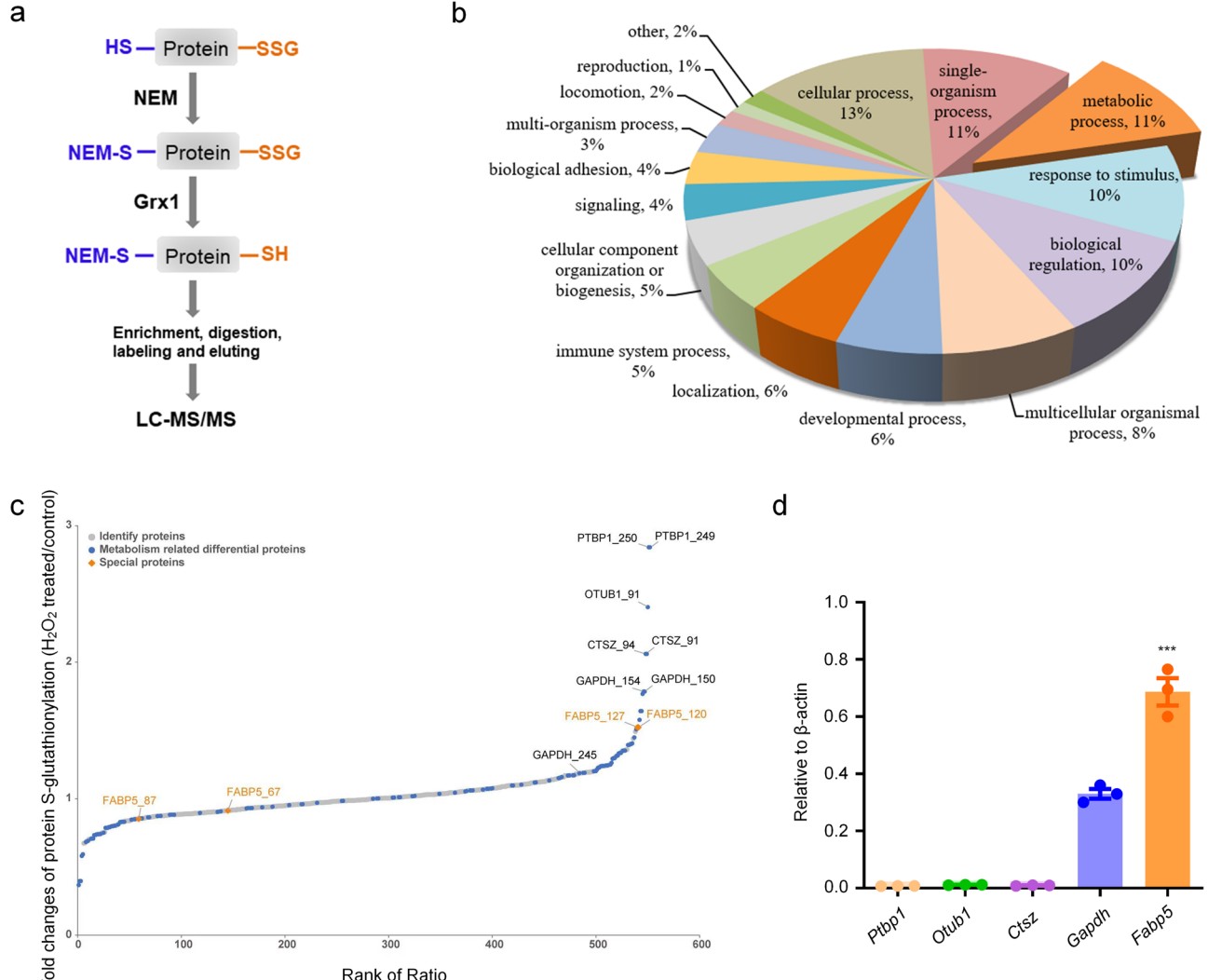

**Fig. 3 Alterations of protein S-glutathionylation modification in macrophages under oxidative conditions. a** Schematic of the enrichment and quantitative analysis for SSG-modified Cys-peptides. **b** Enrichment of protein SSG and LC–MS/MS in BMDMs exposed to $H_2O_2$ (200 μM) for 0, 15 min. Gene ontology annotations for biological process of proteins with significant S-glutathionylation alterations found to be increased or decreased in any stage by >1.2-fold compared to control. The slice for metabolic process is slightly displaced from the rest of the pie chart. **c** Alterations in S-glutathionylation sites under exposure to $H_2O_2$ [shown as fold changes of protein S-glutathionylation ($H_2O_2$ treated/control). Gray, identified protein sites; blue, metabolism-related differential proteins; orange, FABP5]. **d** mRNA levels of *Ptbp1*, *Otub1*, *Ctsz*, *Gapdh*, and *Fabp5* in BMDMs as evaluated by qPCR, $n = 3$ biologically independent samples over 3 independent experiments, data are shown as mean ± SEM and analyzed with a 95% confidence interval, $P < 0.0001$, one-way ANOVA is followed by Tukey's post-hoc test. Source data are provided as a Source Data file.

overexpressing FABP5 (Fig. 5f, Supplementary Fig. 5a, b). Consistently, overexpression of Grx1 substantially inhibited the nuclear accumulation of FABP5 (Fig. 5g). Besides, similar results were obtained from control and Grx1 KO BMDMs (Fig. 5h). LPS treatment also promoted the nuclear accumulation of FABP5 in BMDMs (Supplementary Fig. 5c). The above findings associate the improved fatty acid binding and nuclear translocation of FABP5 with protein S-glutathionylation.

FABP5 is abundantly expressed in macrophages, and the pharmacological inhibition of FABP5 is associated with inflammation, stress, and pain[33,46]. Recent studies illustrated that, upon association with particular ligands, FABP5 translocates from the cytosol to the nucleus and then directly performs its functions within the nucleus[28]. However, the regulatory mechanisms and pathophysiological relevance remained unclear. To further explore the functional significance of FABP5 S-glutathionylation, LPS-induced transcription of the inflammatory cytokines in macrophages was analyzed. As shown in Fig. 5i and

Supplementary Fig. 5d, Grx1 deletion or inhibition prevented the induction of IL-1β, IL-6, and TNFα in LPS-activated BMDMs, revealing the anti-inflammatory effect of protein S-glutathionylation. Conversely, inflammatory cytokines were significantly increased by pretreatment with FABP5 inhibitors (SBFI26 and BMS309403) in LPS-treated BMDMs (Fig. 5j, Supplementary Fig. 5e). SBFI26, a novel a-truxillic acid 1-naphthyl mono-ester, which has been proved to exhibit strong binding and better inhibition of FABP5 than BMS309403[44,45], was used as FABP5 inhibitor to confirm the results obtainted from BMS309403. Furthermore, in Grx1 KO BMDMs, FABP5 inhibitors suppressed the vital role of protein S-glutathionylation in limiting inflammation (Fig. 5k). All these results indicate a potential inhibitory role of FABP5 S-glutathionylation in macrophages during inflammation.

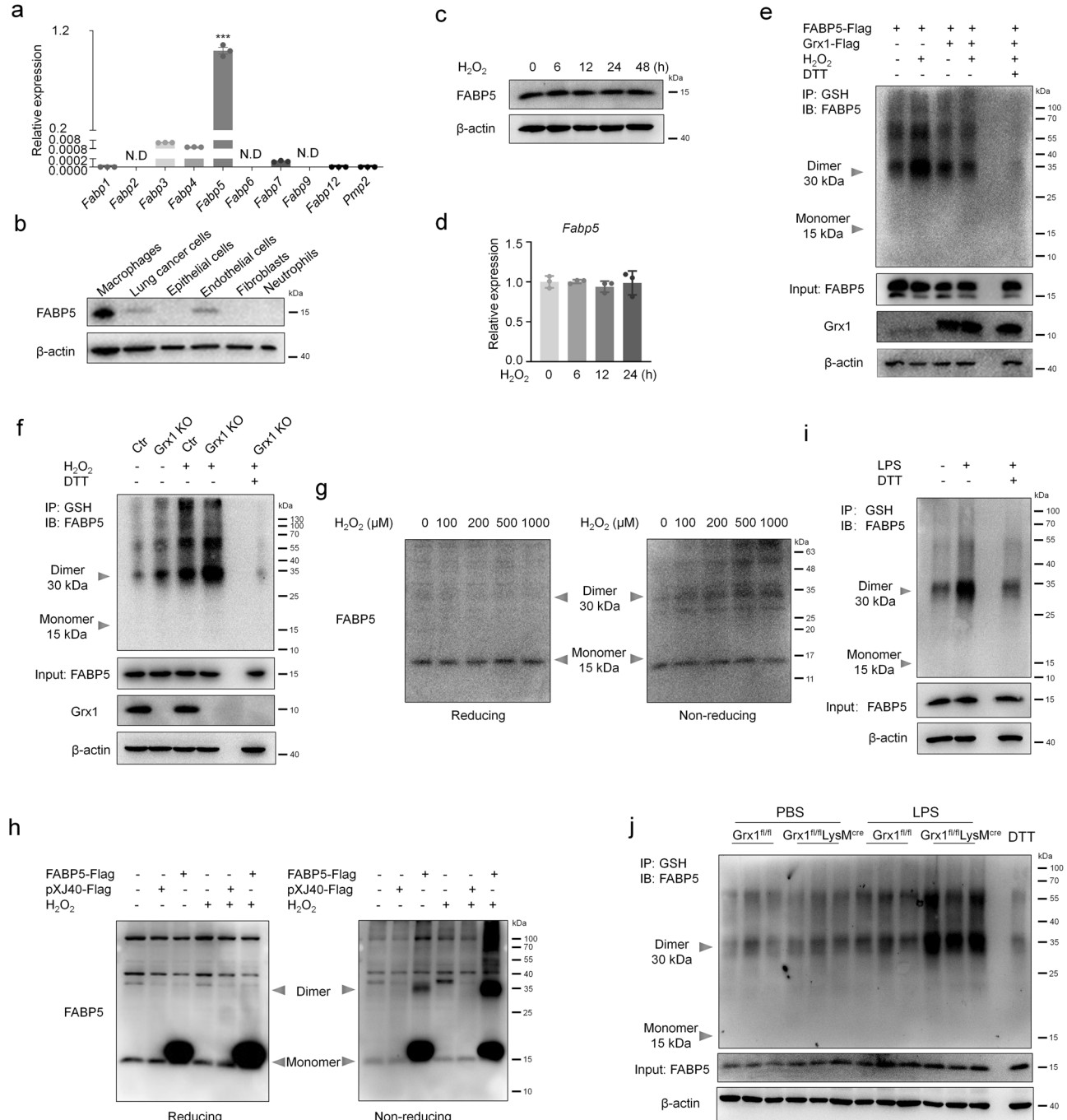

**Fig. 4 FABP5 is S-glutathionylated under oxidative stress conditions. a** mRNA levels of FABP family members in RAW264.7 cells as determined by qPCR, $n = 3$ biologically independent samples over three independent experiments, data are presented as mean ± SEM and analyzed with a 95% confidence interval, $P < 0.0001$, N.D, not detected, one-way ANOVA is followed by Tukey's post-hoc test. **b** Protein levels of FABP5 in macrophages (MH-S), lung cancer cells (A549), epithelial cells (MLE-12), endothelial cells (HUVEC), fibroblasts (HFL1), and neutrophils (primary neutrophils) as evaluated by immunoblot. **c** Levels of FABP5 evaluated by immunoblot in RAW264.7 cells exposed to $H_2O_2$ (200 μM) for indicated time (β-actin, loading control). **d** mRNA levels of *Fabp5* in RAW264.7 cells after exposure to $H_2O_2$ (200 μM) for indicated time as determined by qPCR, $n = 3$ biologically independent samples over three independent experiments, data are shown as mean ± SEM and analyzed with a 95% confidence interval, one-way ANOVA is followed by Tukey's post-hoc test. **e** Co-IP of S-glutathionylation of FABP5 in COS-7 cells co-transfected with pXJ40-3xFlag-FABP5 and pcDNA3.1-3xFlag-Grx1 (or vector) and exposed for 15 min to $H_2O_2$ (200 μM) at 24 h post-transfection (IP, GSH; IB, FABP5). Whole cell lysates confirm the expression of FABP5, Grx1, and β-actin (DTT, negative control). **f** Co-IP showing S-glutathionylation of FABP5 in BMDMs (from control or Grx1 KO mice) after exposure to $H_2O_2$ (200 μM) for 15 min (IP, GSH; IB, FABP5). Whole cell lysates confirm the expression of FABP5, Grx1, and β-actin (DTT, negative control). **g** Immunoblotting analysis of reducing or non-reducing SDS–PAGE of RAW264.7 cells after exposure to $H_2O_2$ (at the indicated concentration) for 15 min. **h** COS-7 cell lysates subjected to SDS–PAGE under reducing or non-reducing conditions after transfection of pXJ40-3xFlag vector or pXJ40-3xFlag-FABP5 and exposure to $H_2O_2$ (200 μM) for 15 min. **i** Co-IP for S-glutathionylation of FABP5 in WT BMDMs after treatment with LPS (100 ng/mL) for 15 min (IP, GSH; IB, FABP5). Whole cell lysates confirm the expression of FABP5 and β-actin. **j** S-glutathionylation of FABP5 in lung tissues from Grx1^fl/fl and Grx1^fl/fl LysM^cre mouse 24 h following intratracheal administration of PBS or LPS. Source data are provided as a Source Data file.

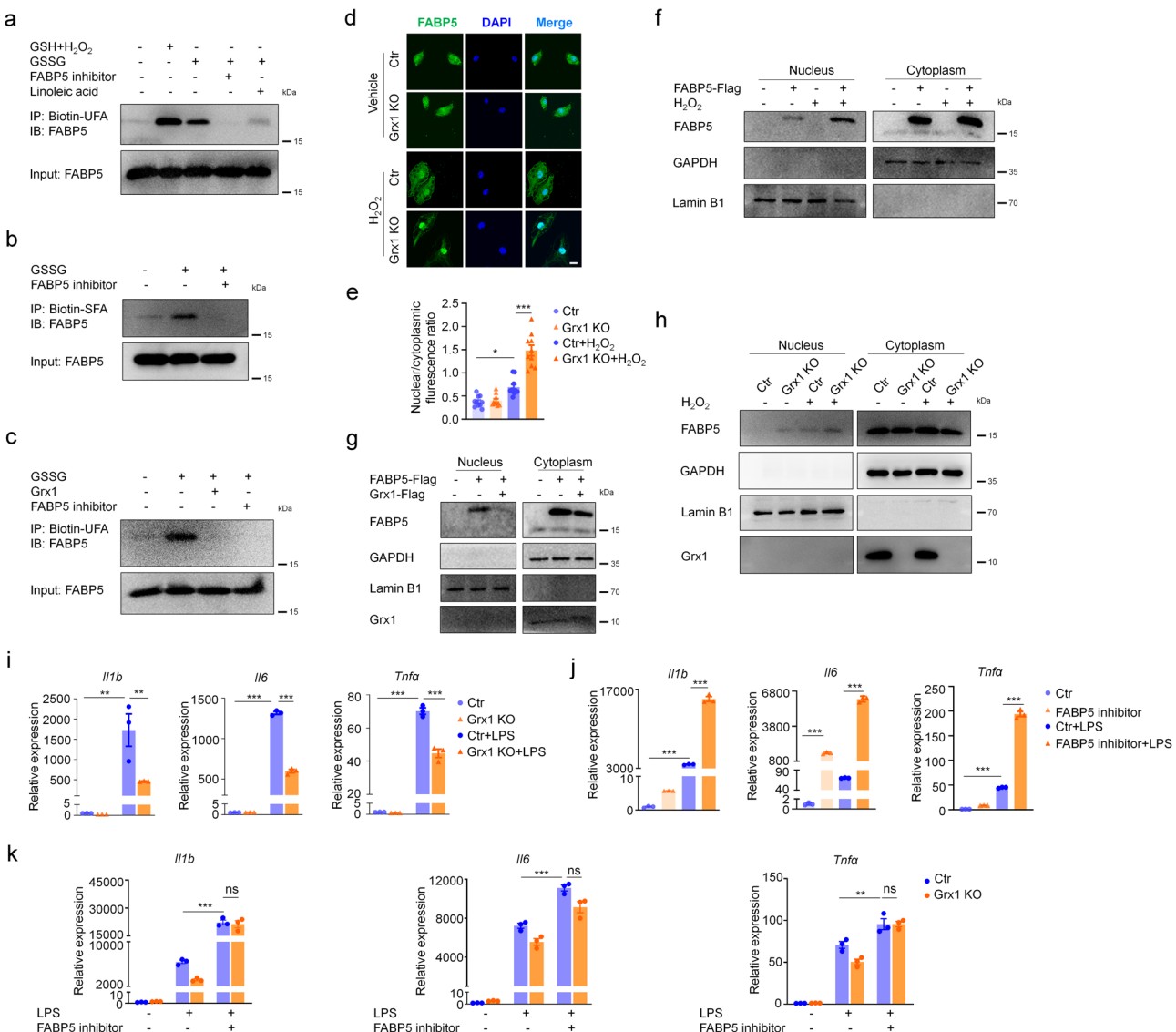

**Fig. 5 S-glutathionylation promotes the nuclear translocation and fatty acid binding of FABP5. a–c** Immunoblot analysis of FABP5 binding to ULCFA (**a**), SLCFA (**b**) or ULCFA with Grx1 treatment (**c**). Purified recombinant FABP5 protein was incubated with GSH plus $H_2O_2$ or GSSG for 15 min, then mixed with Biotin-ULCFA or Biotin-SLCFA in the absence or presence of BMS309403 or linoleic acid or Grx1. FABP5 interacted with the Biotin-ULCFA or Biotin-SLCFA was captured using streptavidin magnetic beads. **d** Immunofluorescence staining of BMDMs after exposure to $H_2O_2$ (200 μM) for 30 min (green, FABP5; blue, DAPI; scale bars, 10 μm). **e** Quantitative analysis of the ratio of nuclear/cytoplasmic fluorescence intensity in BMDMs as in (**d**), $n = 10$ in each group, $P_{(ctr\ vs.\ ctr\ H2O2)} = 0.0172$, $P_{(ctr\ H2O2\ vs.\ Grx1\ KO\ H2O2)} < 0.0001$. **f** Immunoblot analysis of cytoplasmic and nuclear FABP5 in COS-7 cells. COS-7 cells were transfected with pXJ40-3xFlag vector or pXJ40-3xFlag-FABP5 followed by exposure to $H_2O_2$ (200 μM) for 1 h. Lamin B1 (nuclear fraction) and GAPDH (cytoplasmic fraction) are loading controls. **g** Immunoblots with antibodies against the indicated proteins in nuclear and cytoplasmic fractions from COS-7 cells co-transfected with pXJ40-3xFlag-FABP5 (or vector) and pcDNA3.1-3xFlag-Grx1 (or vector) for 24 h, then exposed to $H_2O_2$ (200 μM) for 1 h. **h** Immunoblots against the indicated proteins of cytoplasmic and nuclear extracts from BMDMs exposed to $H_2O_2$ (200 μM) for 1 h. **i** mRNA expression of *Il1b*, *Il6*, and *Tnfα* in BMDMs from mice stimulated with LPS (100 ng/mL) for 4 h, as evaluated by qPCR, $n = 3$ in each group, $P_{(Il1b,\ Ctr\ vs.\ Ctr+LPS)} = 0.0013$, $P_{(Il1b,\ Ctr+LPS\ vs.\ Grx1\ KO+LPS)} = 0.0091$, ***$P < 0.0001$. **j** mRNA levels of *Il1b*, *Il6*, and *Tnfα* in BMDMs pre-treated with an FABP5 inhibitor (SBFI26, 100 μM) for 2 h and stimulated with LPS (100 ng/mL) for 4 h, as determined by qPCR, $n = 3$ in each group, ***$P < 0.0001$. **k** mRNA levels of *Il1b*, *Il6*, and *Tnfα* in BMDMs after treated with SBFI26 (100 μM) for 2 h and stimulation with LPS (100 ng/mL) for 4 h, as evaluated by qPCR, $n = 3$ in each group, $P_{(Il1b,\ Ctr+LPS\ vs.\ Ctr+LPS+inhibitor)} < 0.0001$, $P_{(Il6,\ Ctr+LPS\ vs.\ Ctr+LPS+inhibitor)} < 0.0001$, $P_{(Tnfα,\ Ctr+LPS\ vs.\ Ctr+LPS+inhibitor)} = 0.0046$. All samples were biologically independent and three or more independent experiments were performed. All quantitative data are shown as mean ± SEM and analyzed with a 95% confidence interval. One-way ANOVA followed by Tukey's post-hoc test for (e, i-k). *$P < 0.05$, **$P < 0.01$, ***$P < 0.001$, ns = no significance. Source data are provided as a Source Data file.

**Cys127 is required for the fatty acid binding and nuclear accumulation of FABP5 under oxidative stress.** *Homo sapience* FABP5 has six cysteines, which is many more than the other members of the FABP family (Supplementary Fig. 6a, Fig. 6a). In addition, sequence alignments from diverse species revealed that

five cysteines (Cys43, Cys67, Cys87, Cys120, and Cys127) in FABP5 are conserved (Supplementary Fig. 6b), implying the crucial role of cysteine in the regulation of FABP5 function. In our proteomic analysis, FABP5 was identified to be glutathiony-lated on four cysteine residues: cys67, cys87, cys120, and cys127

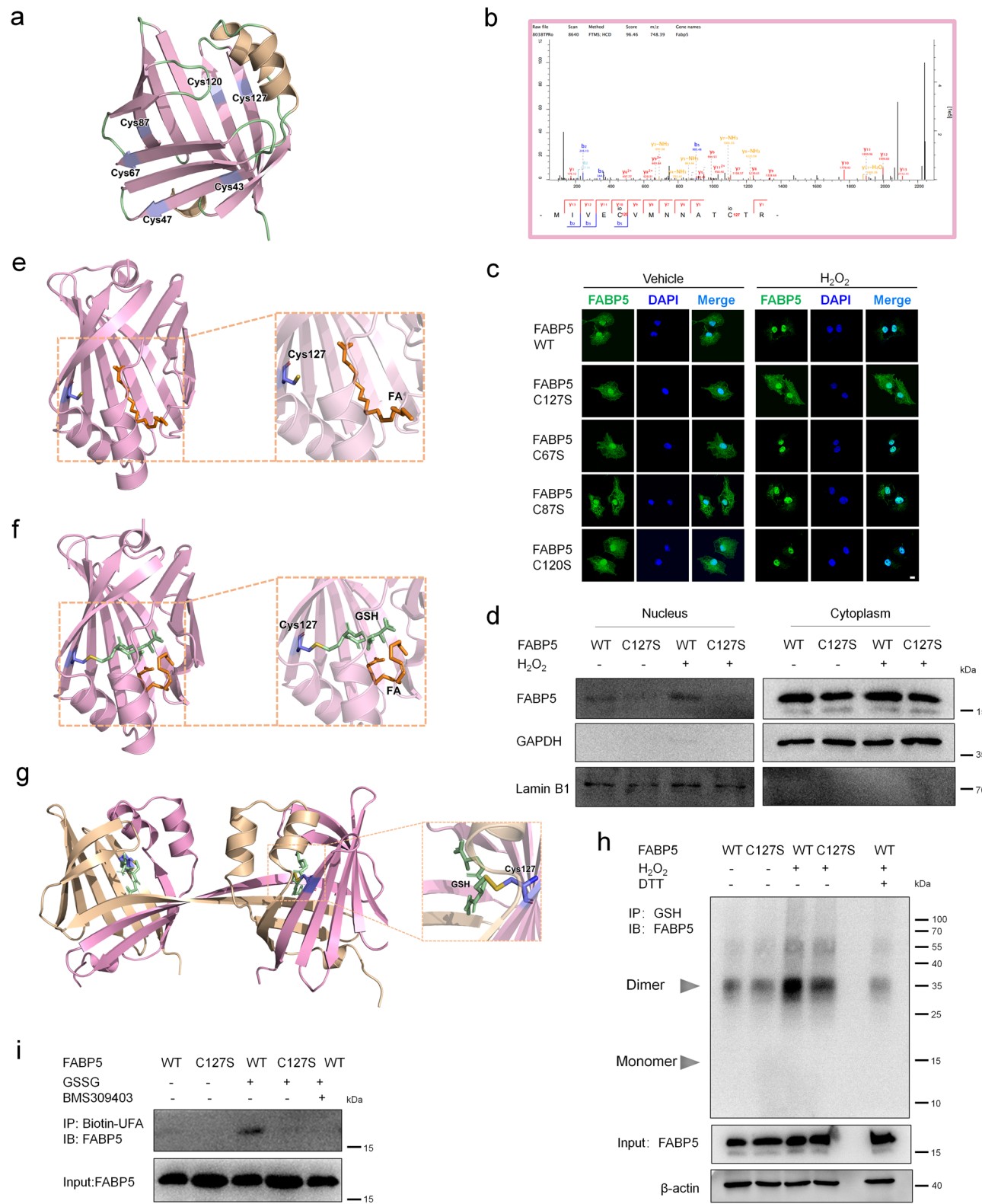

(Fig. 6b, Supplementary Fig. 6c, d). To determine which cysteine residue(s) plays a major role in the regulation of FABP5, we mutated each of the four putative S-glutathionylation sites to serine and assayed their activity individually. As shown in Fig. 6c and Supplementary Fig. 7a, overexpression of FABP5 WT, C67S, C87S, C120S, and C127S did not change the nuclear translocation of FABP5. $H_2O_2$ treatment clearly increased nuclear accumulation of FABP5 (WT, C67S, C87S and C120S). Mutation of Cys127

to Ser (FABP5 C127S) blocked the $H_2O_2$-induced nuclear translocation of FABP5 in COS-7 cells. In addition, the same result was obtained in the nuclear components of COS-7 cells overexpressing FABP5 C127S (Fig. 6d), while the mutation of Cys120 to Ser did not block the nuclear translocation of FABP5 induced by $H_2O_2$ (Supplementary Fig. 7b). On the other hand, *Homo sapience* FABP5 structural data (PDB: 4LKT) showed that Cys127 is located within the ligand-binding pocket, indicating a

**Fig. 6 Cys127 S-glutathionylation of FABP5 potentiates its nuclear translocation and fatty acid binding in response to ROS. a** Model structure of FABP5 and its 6 cysteines, generated by Pymol and based on the crystal structure of *Homo sapiens* FABP5 (PDB ID,4LKT). **b** Mass spectra of a peptide from FABP5 including glutathionylated cysteine 120 and cysteine127. **c** Immunofluorescence staining (with FABP5) and confocal microscopy imaging of COS-7 cells with overexpression of FABP5 WT, C67S, C87S, C120S, and C127S after exposure to $H_2O_2$ (200 μM) for 1 h (green, FABP5; blue, DAPI; scale bars, 10 μm). **d** Immunoblot analysis of cytoplasmic and nuclear FABP5 in COS-7 cells transfected with FABP5 WT or C127S after exposure to $H_2O_2$ (200 μM) for 1 h. Lamin B1 (nuclear fraction) and GAPDH (cytoplasmic fraction) are loading controls. **e** Model structure of FABP5 interaction with an FA (Fatty Acid, Linoleic Acid), generated by Pymol and based on the crystal structure of FABP5 (PDB ID, 4LKT). **f** Docking structure of FABP5 in complex with S-glutathionylated Cys127 interacting with an FA (Fatty Acid, Linoleic Acid). **g** Docking structure of dimer FABP5 (PDB ID, 4AZM, *Homo sapiens*) in complex with reduced glutathione (GSH) on Cys127. **h** Co-IP for S-glutathionylation of FABP5 in COS-7 cells overexpressing pXJ40-3xFlag-FABP5 WT or C127S and exposed to $H_2O_2$ (200 μM) for 15 min (IP, GSH; IB, FABP5). Whole-cell lysates confirm the expression of FABP5 and β-actin (DTT, negative control). **i** Immunoblot analysis of FABP5 fatty acid binding. Purified recombinant FABP5 WT or C127S protein (1 μg) was incubated with GSSG (1 mM) for 15 min, then mixed with Biotin-linoleic acid in the absence or presence of BMS309403 (5 mM) for 30 min. FABP5 that associated with the Biotin-linoleic acid was captured using streptavidin magnetic beads. Source data are provided as a Source Data file.

regulatory role of Cys127 in fatty-acid binding (Fig. 6e). Moreover, molecular docking analysis revealed that monomer FABP5 had a pocket that perfectly accommodated the GSH molecule linked to Cys127, displaying the possibility of Cys127 S-glutathionylation (Fig. 6f). The pocket of dimer FABP5 even had a larger space for GSH molecule linked to Cys127 (Fig. 6g). Glutathionylation of FABP5 at Cys127 was further confirmed by Co-IP in FABP5 WT and C127S-overexpressing cells. In response to $H_2O_2$, FABP5 WT was strongly glutathionylated, whereas mutation of Cys127 to Ser almost completely blocked the $H_2O_2$-induced S-glutathionylation of FABP5 (Fig. 6h, Supplementary Fig. 7e). Mutation of Cys43, Cys47, Cys67, Cys87 or Cys120 to Ser did not affect the S-glutathionylation of FABP5 (Supplementary Fig. 7c, d). In addition, purified recombinant FABP5 C127S protein almost completely abolished the fatty-acid binding of FABP5 (Fig. 6i, Supplementary Fig. 7f). Together, these results clearly indicate that S-glutathionylation at Cys127 promotes the fatty-acid binding and nuclear translocation of FABP5.

**S-glutathionylation of FABP5 inhibits LPS-induced inflammation in macrophages.** To gain insight into the changes in biological processes and pathways caused by FABP5 S-glutathionylation, differentially-expressed genes in macrophages were assessed by RNA-seq. FABP5 KO BMDMs were transfected with FABP5 WT or C127S by nucleofection and treated with LPS for 24 h. Consistent with our results (Fig. 5i–k), regulation of the inflammatory response and cytokine production were enriched in macrophages overexpressing FABP5 C127S relative to FABP5 WT (Fig. 7a–c). Besides, regulation of ROS biosynthetic/metabolic process and response to lipid were also enriched in FABP5 C127S macrophages (Supplementary Fig. 8a, b). After RNA-seq, validation by qPCR was performed on macrophage samples. Not surprisingly, mutation of cysteine 127 markedly increased the LPS-induced production of IL-1β, IL-6, and TNFα in BMDMs (Fig. 7d). Grx1 deletion clearly decreased the levels of IL-1β, IL-6, and TNFα stimulated by LPS. However, almost no difference was detected between WT and Grx1-KO BMDMs after FABP5 C127S overexpression (Fig. 7e), thus demonstrating the central role of FABP5 cysteine 127 in mediating macrophage inflammation through S-glutathionylation.

In order to confirm the function of FABP5 in the regulation of acute lung injury pathogenesis in vivo, mice with macrophage-specific FABP5 deficiency (Fabp5^fl/fl^LysM^cre^) and their wild-type littermate controls (Fabp5^fl/fl^) were generated. Histological analysis and counts of leukocytes were performed 24 h after intratracheal LPS instillation. As shown in Fig. 7f, g, FABP5 deficiency aggravated inflammatory responses in the LPS-induced mouse model of acute lung injury, with substantial destruction of alveolar structure and increased infiltration of inflammatory cells.

Recent studies have demonstrated that FABPs selectively cooperate with the nuclear receptors RARα, ERRα[47], PPARβ/δ, and PPARγ. In contrast to FABP4, FABP5 mobilizes to the nucleus only in response to ligands that activate PPARβ/δ[28]. Next, we further explored the mechanism by which FABP5 S-glutathionylation inhibits the inflammation in macrophages. Pull-down assay was performed by using purified mouse recombinant FABP5 (WT or C127S) protein and cell lysates, S-glutathionylation promoted the binding of FABP5 and PPARβ/δ, and mutation of Cys127 to Ser almost completely abolished the binding of FABP5 and PPARβ/δ (Fig. 8a). Then, the ability of S-glutathionylated FABP5 to activate PPARβ/δ was examined. Transcriptional activation assays using a luciferase reporter driven by a consensus PPAR response element (PPRE) showed that the synthetic PPARβ/δ-selective ligand GW0742, induced transcription of the reporter. $H_2O_2$ enhanced the expression of the PPRE-driven reporter and the response was markedly suppressed when FABP5 C127S was overexpressed (Fig. 8b, c), indicating that the ability of FABP5 to induce reporter expression was enhanced by S-glutathionylation. Then, we performed chromatin immunoprecipitation (ChIP) assays to detect whether FABP5 directly interacts with PPRE. Results revealed that recruitment of FABP5 to PPRE on the promoters of PPARβ/δ target genes, such as ADRP, FIAF, were markedly increased in response to $H_2O_2$ treatment (Fig. 8d). We then set out to examine the ability of S-glutathionylated FABP5 to induce the expression of endogenous PPARβ/δ target genes in macrophages. Fasting-induced adipose factor (Fiaf), adipose differentiation-related protein (Adrp) and carnitine palmitoyltransferase 1a (Cpt1a), were previously shown to be direct PPARβ/δ targets[28,48]. The PPARβ/δ ligand GW0742 increased the mRNA expression of all three of these endogenous PPARβ/δ target genes. Furthermore, pre-incubation with the FABP5 inhibitor BMS309403 blocked the GW0742-induced mRNA expression, indicating the important role of FABP5 in PPARβ/δ activation (Fig. 8e). To further explore the connection between FABP5 S-glutathionylation and human diseases, we examined the S-glutathionylation of FABP5 after transfection of FABP5 G114R, a reported disease-associated mutation[49]. As shown in Fig. 8f, G114R enhanced the S-glutathionylation of FABP5, suggesting a potential role of FABP5 S-glutathionylation in disease. Taken together, our analyses indicate that S-glutathionylation of FABP5 controls inflammatory activity in macrophages, alleviating acute lung injury.

**Discussion**
ROS plays a pivotal role in physiological cellular processes, but the generation of excess ROS causes oxidative damage to molecules and cells in a multitude of pathological conditions, including acute lung injury[50]. The initiation of acute lung injury

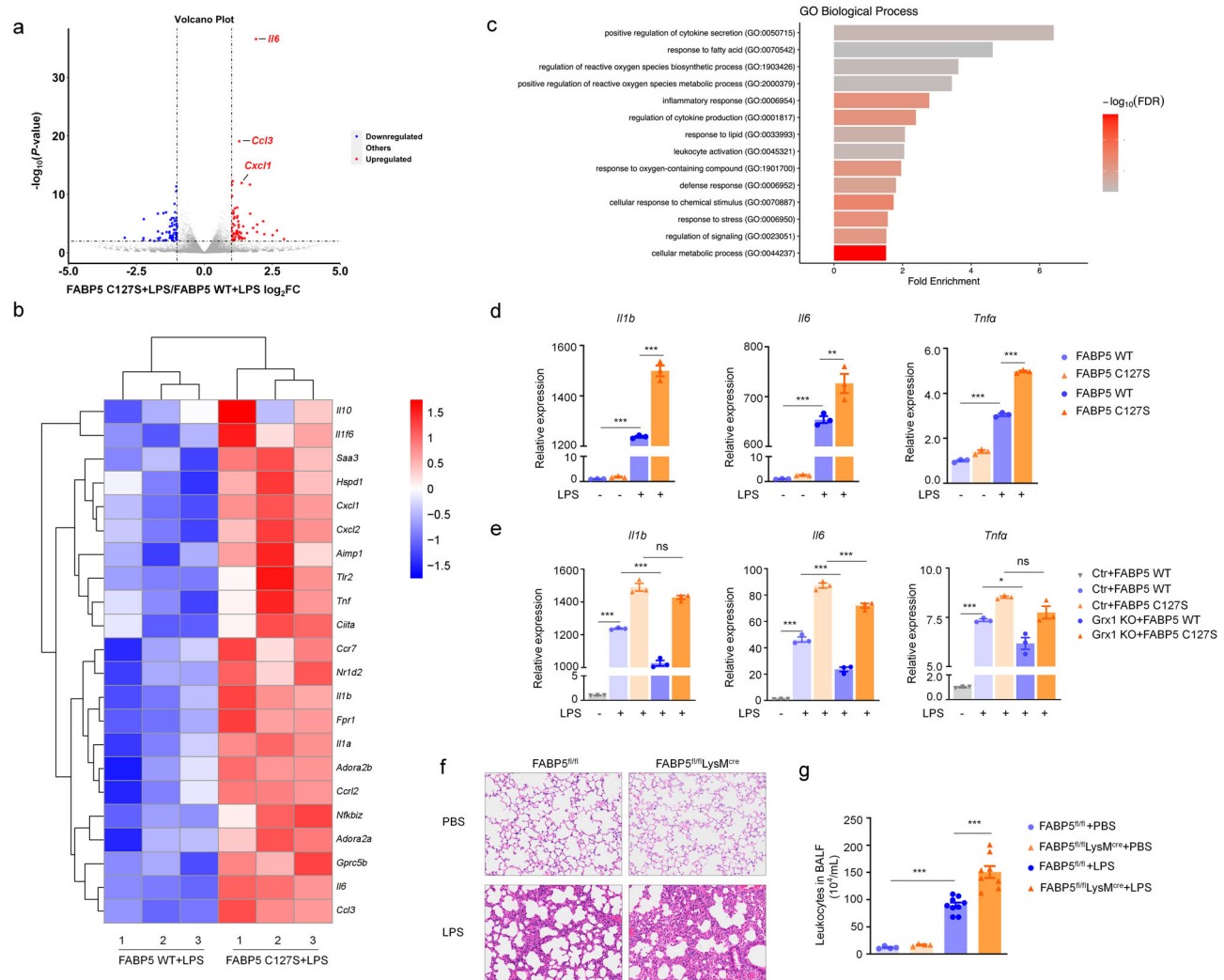

**Fig. 7 S-glutathionylation of FABP5 suppresses macrophage inflammation in vivo. a** Volcano plot of RNA-seq transcriptome data displaying the pattern of gene expression values for FABP5 KO BMDMs with overexpression of FABP5 C127S relative to FABP5 WT after exposure to LPS (500 ng/mL) for 24 h [red and blue dots, significantly up-and down-regulated genes ($P < 0.01$, $|\log_2 FC| > 1$); black dashed lines, boundary for identification of up- or down-regulated genes]. **b** Heatmap of regulation of the inflammatory response and cytokine transcript production in LPS-stimulated (24 h) BMDMs nucleofected with either pXJ40-3xFlag-FABP5 WT or C127S. Data shown are relative to the calculated Z scores across the samples. Red, relatively high levels of expression; blue, relatively low levels of expression. Each column represents one individual (for a total of 3 per group) and each row represents the expression of a single gene. **c** Diagram of GO analysis classifying DEGs into biological process groups using Panther (http://www.pantherdb.org). Some GO categories with FDR <0.05 are included. GO, gene ontology; DEGs, differentially-expressed genes. **d** mRNA expression of *Il1b*, *Il6*, and *Tnfα* in FABP5 KO BMDMs after nucleofection with pXJ40-3xFlag-FABP5 WT or C127S and stimulation with LPS (1 μg/mL) for 24 h, as identified by qPCR, $n = 3$ in each group, $P_{(Il6,\ FABP5\ WT+LPS\ vs.\ FABP5\ C127S+LPS)} = 0.0045$, ***$P < 0.0001$. **e** mRNA expression of *Il1b*, *Il6*, and *Tnfα* in Grx1 KO BMDMs after nucleofection of pXJ40-3xFlag-FABP5 WT or C127S and treatment with LPS (500 ng/mL) for 24 h, as determined by qPCR, $n = 3$ in each group, $P_{(Il6,\ Ctr+FABP5\ C127S\ vs.\ Grx1\ KO+FABP5\ C127S)} = 0.0004$, $P_{(TNFα,\ Ctr+FABP5\ WT\ vs.\ Grx1\ KO+FABP5\ WT)} = 0.0123$, ***$P < 0.0001$(except group *Il6*, Ctr+FABP5 C127S vs. Grx1 KO+FABP5 C127S). **f** H&E staining showing lung histopathological changes in FABP5^fl/fl and FABP5^fl/flLysM^cre mice 24 h after intratracheal administration of PBS or LPS (5 mg/kg). Scale bars, 50 μm. **g** Total numbers of leukocytes in BALF from FABP5^fl/fl and FABP5^fl/flLysM^cre mice 24 h after intratracheal administration of PBS or LPS (5 mg/kg), $n = 4, 4, 9, 8$, respectively, ***$P < 0.0001$. All samples were biologically independent and three or more independent experiments were performed. All quantitative data are shown as mean ± SEM and analyzed with a 95% confidence interval. One-way ANOVA followed by Tukey's post-hoc test for (**d–e**, **g**). *$P < 0.05$, **$P < 0.01$, ***$P < 0.001$. Source data are provided as a Source Data file.

is associated most often with pneumonia, sepsis, multiple transfusions, trauma, or shock, leading to activation of the acute inflammatory response on a systemic level[51]. The influx of neutrophils and macrophages into the alveolar space releases ROS, inflammatory cytokines, and other cytotoxic and pro-inflammatory compounds. ROS upregulates inflammatory cytokines, which perpetuates a vicious cycle by recruiting more inflammatory cells, ultimately leading to profound tissue damage[50,52].

However, the exact redox mechanisms whereby oxidant-induced modifications contribute to acute lung injury are still not clear. In this study, we revealed that the absence of Grx1 in macrophages significantly ameliorated acute lung injury. According to the results of quantitative mass spectrometry screening and in vitro experiments, we demonstrated that FABP5 is susceptible to S-glutathionylation under oxidative stress conditions. Further experiments showed that S-glutathionylation on the Cys-127 residue of FABP5 promoted its fatty acid binding and

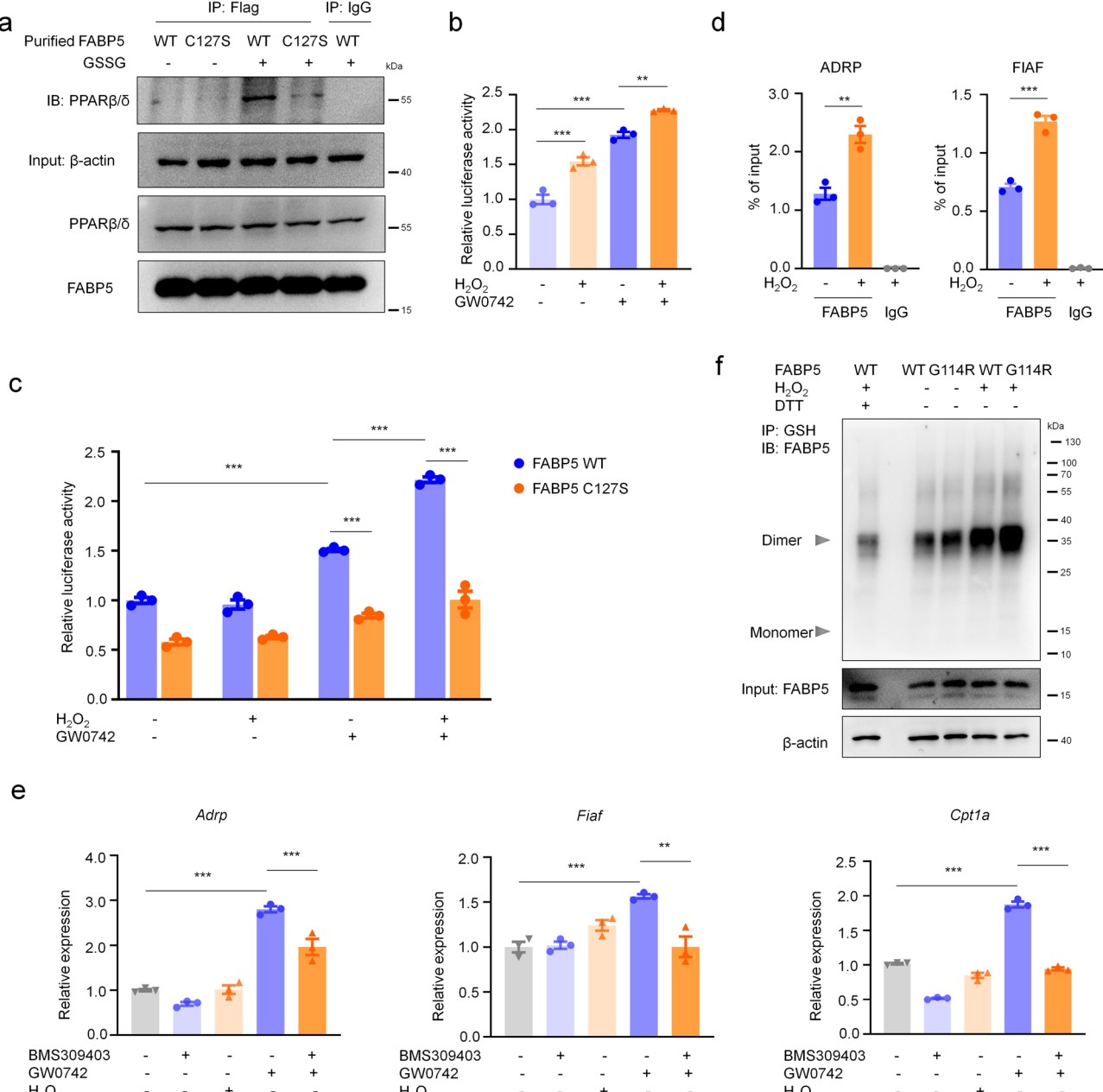

**Fig. 8 FABP5 S-glutathionylation controls macrophage inflammation by activating PPARβ/δ. a** In vitro pull-down assays. The mouse recombinant FABP5 (WT or C127S) protein with 3xFlag tag purified from E. coli treated with GSSG (1 mM) for 15 min was incubated with COS-7 lysates and precipitated with Flag-magnetic beads, respectively. Precipitates were subjected to SDSPAGE and examined by immunoblot with anti PPARβ/δ antibodies. **b** Transactivation assays in Raw264.7 cells transfected with a PPRE-driven luciferase reporter. After nucleofection with PPRE X3-TK-luc and pRL-TK, Raw264.7 cells were serum starved for 8 h, then exposed to $H_2O_2$ (200 μM), GW0742 (1 μM), or both for 24 h, and luciferase activity was determined, $n = 3$ in each group, $P_{(ctr\ vs.\ H2O2)} = 0.0003$, $P_{(ctr\ vs.GW0742)} < 0.0001$, $P_{(GW0742\ vs.H2O2+GW0742)} = 0.0047$. **c** Transactivation assays in COS-7 cells transfected with a PPRE-driven luciferase reporter. After transfection with PPRE X3-TK-luc, FABP5, and pRL-TK, cells were serum starved for 8 h, then treated with $H_2O_2$ (200 μM), GW0742 (1 μM), or both for 24 h, and luciferase activity was measured, $n = 3$ in each group, ***$P < 0.0001$. **d** ChIP analysis of FABP5 recruitment to the PPREs of PPARβ/δ target gene promoters in Raw 264.7 cells after exposure to $H_2O_2$ (200 μM) for 1 h. ChIP signals were quantified by qPCR, $n = 3$ in each group, $P_{(ADRP,\ ctr\ vs.\ H2O2)} = 0.001$, $P_{(FIAF,\ ctr\ vs.\ H2O2)} < 0.0001$. **e** mRNA levels of *Adrp*, *Fiaf*, and *Cpt1a* in RAW264.7 cells after treatment with BMS309403 (50 μM) for 2 h, stimulation with GW0742 (1 μM) for 30 min, and exposure to $H_2O_2$ (200 μM) for 4 h, as determined by qPCR, $n = 3$ in each group, $P_{(Adrp,\ ctr\ vs.\ GW0742)} < 0.0001$, $P_{(Adrp,\ GW0742\ vs.\ GW0742+BMS309403)} = 0.0009$, $P_{(Fiaf,\ ctr\ vs.\ GW0742)} = 0.001$, $P_{(Fiaf,\ GW0742\ vs.\ GW0742+BMS309403)} = 0.001$, $P_{(Cpt1α,\ ctr\ vs.\ GW0742)} < 0.0001$, $P_{(Cpt1α,\ GW0742\ vs.\ GW0742+BMS309403)} < 0.0001$. **f** Co-IP for S-glutathionylation of FABP5 in COS-7 cells overexpressing pXJ40-3xFlag-FABP5 WT or G114R and exposed to $H_2O_2$ (200 μM) for 15 min (IP, GSH; IB, FABP5). Whole-cell lysates confirm the expression of FABP5 and β-actin (DTT, negative control). All samples were biologically independent and three or more independent experiments were performed. All quantitative data are shown as mean ± SEM and analyzed with a 95% confidence interval. One-way ANOVA followed by Tukey's post-hoc test for (**b**–**e**). *$P < 0.05$, **$P < 0.01$, ***$P < 0.001$. Source data are provided as a Source Data file.

nuclear translocation, activating PPARβ/δ and inhibiting macrophage inflammation.

An emerging body of literature has explored the function of Grx1 in diverse respiratory diseases, showing that S-glutathionylation plays different roles upon diverse stimuli and in different cell types[19,25,53]. The relationship between S-glutathionylation and the pathogenesis of acute lung injury is only now emerging[19]. But, thus far, the mechanistic details whereby Grx1 regulates the pathophysiology of acute lung injury remain unclear, and S-glutathionylation targets have yet to be fully understood.

The role of Grx1 and protein S-glutathionylation in LPS-induced acute lung injury has been reported recently[19]. To confirm the role of oxidative stress and redox regulation in acute lung injury, hyperoxia-induced acute lung injury, another commonly-used mouse model of acute lung injury were used. Hyperoxia-induced acute lung injury has been used for many years as a model of oxidative stress mimicking clinical acute lung injury. Excess quantities of ROS are responsible for oxidative stress-related lung injury[54]. Although the stimuli that induce acute lung injury are different, these two mouse models have common features, such as inflammation, ROS and epithelial injury. In our work, exposure to hyperoxia increased the level of protein S-glutathionylation in lung tissue, and lack of Grx1 evidently attenuated hyperoxia-induced acute lung injury, confirming the protective role of Grx1-regulated S-glutathionylation in acute lung injury. Recent studies have shown Grx1 also plays a crucial role in hypoxia-induced oxidative stress[55–57]. Both hyperoxia and hypoxic conditions (such as COPD and smoking)[22,24] favour the increase of ROS, although the specific mechanisms may be different. The major contributor of hyperoxia-induced ROS is activation of the multiprotein enzyme complex NADPH oxidase, while most ROS are generated in cells by the mitochondrial respiratory chain under hypoxic conditions.

Further analysis of ROS and Grx1 expression levels suggested that Grx1 and S-glutathionylation in macrophages play a crucial role in the pathogenesis of acute lung injury. To confirm the importance of macrophage Grx1 in acute lung injury, Grx1$^{fl/fl}$LysM$^{cre}$ mice were generated and treated with LPS. Loss of Grx1 in macrophages relieved LPS-induced acute lung injury. To the best of our knowledge, our study is the first to use a myeloid-specific knockout approach to investigate the role of Grx1 in acute lung injury.

Next, in macrophages, we used a redox proteomic approach for site-specific identification and quantification of protein S-glutathionylation to find precise molecular targets of Grx1 that are susceptible to redox-dependent regulation. Macrophages are extremely plastic cells and can change their functional profile rapidly in response to different stimuli[58]. $H_2O_2$, an endogenous oxidant produced by activated macrophages and many pathophysiological stimuli, has been widely used to induce oxidative stress in different cell lines[3,59]. LPS, the major component of the outer membranes of Gram-negative bacteria, induces the release of ROS. In addition, LPS also induces inflammatory responses and leads to lung injury. These inflammatory and oxidative mediators worsen the cellular and lung tissue injury in mice with acute lung injury[52]. Thus, BMDMs were simulated with $H_2O_2$ to identify the specific cysteine redox switches in macrophages under oxidative conditions. MS analyses revealed that FABP5 is susceptible to S-glutathionylation in response to ROS. It is noteworthy that previously reported sites of S-glutathionylation (GAPDH[7], actin[3], and STIM1[60]) were identified in our analyses, providing additional confidence in our results.

Metabolism controls macrophage activation in regulating their phenotype. It has been shown that M1 macrophages are characterized by enhanced glycolysis and impaired mitochondrial oxidative phosphorylation (OXPHOS)[61,62]. M1 macrophages utilize lipids as precursors for the synthesis of inflammatory molecules[63]. On the contrary, M2 macrophages have an intact TCA cycle and enhanced OXPHOS, and fatty acid oxidation is also used[62]. Through the years, several studies have shown that fatty acid oxidation is necessary for inflammasome activation in M1 macrophages, and glycolysis is also involved in fueling fatty acid oxidation in M2 macrophages[63–65]. As a whole, the role of metabolism in macrophage activation is complicated and needs to be further addressed.

As to FABP5, its expression was notably higher than others (PTBP1, OTUB1, CTSZ, and GAPDH) in macrophages. S-sulfenylation of FABP5 on Cys127 has been reported[66], and S-sulfenylation is one potential intermediate prior to forming S-glutathionylation[12]. FABP-regulated lipid metabolism is closely associated with both metabolic and inflammatory processes, in particular, macrophages and adipocytes. Macrophages express high levels of FABP5 and FABP4[26], and FABP5 levels remain unchanged after FABP4 deficiency, suggesting that they are likely to have distinct functions[67]. Interestingly, although FABP4 and FABP5 bind multiple ligands, only specific compounds trigger their nuclear translocation[68]. FABP4 moves into the nucleus in response to ligands that activate PPARγ but not upon treatment with PPARβ/δ ligands. In contrast, FABP5 mobilizes to the nucleus only in response to ligands that activate PPARβ/δ. Our data showed that the expression level of FABP5 was higher than FABP4 in macrophages, reflecting an important role of FABP5 in the regulation of macrophage function. FABP5 has been reported to facilitate lipid trafficking to the nucleus for lipid-mediated transcriptional regulation. However, FABP5 does not harbor a classical nuclear localization signal (NLS) within its primary sequence. A tertiary NLS, located on the α helical cap of the fatty acid binding pocket, is required for ligand-dependent activation. ULCFAs elicit the translocation of FABP5 by permitting allosteric communication between the ligand-sensing β2 loop and the tertiary NLS within the α1 and α2 helices of the protein, and SLCFAs inhibit NLS formation by destabilizing this activation loop[69]. Based on our data, we speculate that FABP5 Cys127 S-glutathionylation enhances the fatty acid binding and nuclear translocation by facilitating the cytosolic exposure or the stabilization of the NLS that is necessary for nuclear import.

Further RNA-seq analysis and the LPS-induced acute lung injury model in Grx1$^{fl/fl}$LysM$^{cre}$ mice indicated that FABP5 S-glutathionylation suppresses LPS-induced inflammation in macrophages. FABP5 is protective against bacterial infection of the lung[33] and plays an important protective role against inflammation and oxidative lung injury during H1N1 influenza A infection[34]. Pull-down and ChIP assays revealed that S-glutathionylation enhances the ability of FABP5 to activate PPARβ/δ. The effects of PPARβ/δ in the control of the macrophage inflammatory response have been largely studied. The anti-inflammatory properties of PPARβ/δ are mainly based on inhibiting NF-κB signaling as well as expression of pro-inflammatory cytokines (iNOS, COX2, TNFα), chemoattractant molecules (MCP-1, MCP-3) and adhesion molecules (VCAM-1, ICAM-1, and E-selectin) in macrophages[70,71].

Overall, our in vivo and in vitro data show that ROS promotes the S-glutathionylation of FABP5 in macrophages under oxidative conditions. S-glutathionylation enhances the fatty acid binding and nuclear accumulation of FABP5, activates PPARβ/δ, and inhibits the LPS-induced inflammation in macrophages. These data confirm Grx1 as a target of anti-inflammatory drug development and suggest that the S-glutathionylation of FABP5 acts as an anti-inflammatory mediator in macrophages during acute lung injury. Our findings shed light on a mechanism by which macrophages prevent self-hyperactivation under oxidative stress.

## Methods

**Mice.** LysM[Cre] mice (C57BL/6 background) were crossed with Grx1[fl/fl] mice (C57BL/6 background) to generate myeloid cell-conditional Grx1 knockout mice (Grx1 [fl/fl]LysM[cre]) and their wild-type littermate controls (Grx1[fl/fl]). LysM[Cre] mice (C57BL/6 background) were crossed with Fabp5[fl/fl] mice (C57BL/6 background) to generate myeloid cell-conditional FABP5 knockout mice (FABP5[fl/fl] LysM[cre]) and their wild-type littermate controls (FABP5[fl/fl]). FABP5[fl/fl] mice were generated by Viewsolid Biotech Co., Ltd (Beijing, China) using CRISPR-Cas9 technology, performed by co-microinjection of in vitro-translated Cas9nickase mRNA, gRNA, and donor DNA into C57BL/6 zygotes. A pair of gRNA sequences (GGACGATATAA GCGCAGATGG and GCAGCCCCTCATTGCTTCCGG) was designed to flank exon 1 of FABP5. Three founder mice from these injections had a correctly-targeted allele with both loxP sites inserted, which was confirmed by genotyping and sequencing analysis around the targeted regions. F0 founders were crossed to C57BL/6 J mice to generate F1 progeny, which were used to establish the flox colony. Grx1 KO mice (C57BL/6 background) were kindly provided by Dr. Yuanfu Xu (State Key Laboratory of Experimental Hematology, Institute of Hematology and Blood Diseases Hospital, Chinese Academy of Medical Sciences and Peking Union Medical College, Tianjin, China)[3]. FABP5[-/-] mice (C57BL/6 background) were generated by Viewsolid Biotech Co., Ltd. All experiments involving equal treatments in wild-type and mutant samples and animals were conducted by experimenters blind to the conditions. Mice were housed under specific pathogen-free conditions. All mice were cultured in suitable temperature and humidity environment and fed with sufficient water and food (25 °C, suitable humidity (typically 50%), 12 hour dark/light cycle). All animal experiments were performed according to protocols approved by the Institutional Animal Care and Use Committee of Zhejiang University School of Medicine (ETHICS CODE: ZJU20200180).

**Cell cultures.** COS-7 cells (ATCC, CRL-1651), EA.hy926 cells (ATCC, CRL-2922), A549 cells (ATCC, CRM-CCL-185) were cultured in DMEM/high glucose (Gibco) supplemented with 10% FBS (Gibco), penicillin (100 U/mL), and streptomycin (100 mg/mL) (HyClone).

Raw264.7 (ATCC, TIB-71) cells were cultured in DMEM/high glucose supplemented with 10% heat-inactivated FBS, penicillin (100 U/mL), and streptomycin (100 mg/mL).

MLE-12 cells (ATCC, CRL-2110) were grown in DMEM/F12 (Gibco) supplemented with 2% FBS, penicillin (100 U/mL), and streptomycin (100 mg/mL).

HFL1(ATCC, CCL-153) cells were grown in DMEM/high glucose supplemented with 10% FBS, MEM Non-Essential Amino Acids Solution (Thermo Fisher Scientific,11140050), penicillin (100 U/mL), and streptomycin (100 mg/mL).

MH-S cells (ATCC, CRL-2019) were cultured in RPMI 1640 (Gibco) supplemented with 10 mM HEPES, 2 mM L-glutamine, 10% heat-inactivated FBS, penicillin (100 U/mL), and streptomycin (100 mg/mL).

AMs were obtained from mice and gathered via quick adhesion. BMDMs were isolated and differentiated as previously described[72]. Cells were cultured in DMEM/F12 and RPMI 1640, respectively, supplemented with 10% heat-inactivated FBS, penicillin (100 U/mL), and streptomycin (100 mg/mL). FABP5 KO BMDMs were nucleofected with 5 μg pXJ40-3xFlag-FABP5 WT or C127S plasmid using the Amaxa Mouse Macrophage Nucleofector Kit (Lonza, VPA-1009) following the manufacturer's instructions.

HUVECs were isolated from umbilical cord vein by lavaging with 0.2% (w/v) collagenase solution at 37 °C for 15 min, the cells were collected and suspended in complete M199 medium. HUVECs between passages 4 and 6 were cultured in a complete medium containing 53% M199 (Corning Incorporated), 37% human endothelial serum-free medium (Thermo Fisher Scientific), and 15 μg/mL of endothelial cell growth supplement (Sigma-Aldrich).

BMS-309403 (BM0015), CdCl$_2$ (202908), LPS (L3129), and H$_2$O$_2$ (88597) were from Sigma-Aldrich. SBFI26 was kindly provided by Iwao Ojima (Institute of Chemical Biology and Drug Discovery, Stony Brook University, Stony Brook, NY, USA).

**Acute lung injury model.** Eight to ten-week-old Grx1[fl/fl]LysM[cre] or FABP5[fl/fl]LysM[cre] male mice (C57BL/6 background) were challenged with LPS (Sigma-Aldrich, L3129) in sterile saline by intratracheal administration; Grx1[fl/fl] or FABP5[fl/fl] mice (C57BL/6 background) served as controls. Mice received a single dose of LPS (5 mg/kg body weight). After 24 h, mice were sacrificed for pathological examination.

**Hyperoxia-induced lung injury.** Grx1 KO and control mouse pups (C57BL/6 background) were exposed to either 21% O$_2$ (room air) or 85% O$_2$ continuously from postnatal days 1 to 7 or 1 to 14. Dams were rotated between normoxia and hyperoxia conditions every 24 h to avoid oxygen toxicity and to ensure sufficient nutrition for the pups during hyperoxia exposure.

**Bronchoalveolar lavage fluid analysis.** Mice were euthanized 24 h after LPS challenge for BALF analysis and ELISA. The left lung was tied at the bronchus and the right lung lavaged with 0.5 ml of Ca$^{2+}$- and Mg$^{2+}$-free phosphate-buffered saline (PBS). This procedure was repeated three times (total volume ~1.5 mL, recovery >80%). BALF from each mouse was collected in an Eppendorf tube,

cooled on ice, and centrifuged at 300 g at 4 °C for 10 min, and supernatants were stored at −80 °C until analysis. To isolate and purify alveolar macrophages, the pellets were re-suspended in DMEM/F-12 medium supplemented with 10% heat-inactivated FBS, penicillin (100 U/mL), and streptomycin (100 mg/mL) and plated for at least 90 min. Non-adherent cells were discarded, and adherent cells were gathered for further analysis.

**ELISA.** Cytokine concentrations of IL-1β, IL-6, and TNF-α in BALF from Grx1[fl/fl] or Grx1[fl/fl]LysM[cre] mice 24 h after intratracheal administration of PBS or LPS (5 mg/kg) were quantified using mouse ELISA kits according to manufacturer's protocols (Invitrogen, 88-7013-88, 88-7064-88, 88-7324-88). Plates were read at 450 nm on a microplate reader (Molecular Devices).

**Grx1 Activity Assay.** Grx1 activity was assayed as described previously[73–75]. Briefly, lung tissue was lysed in 137 mM Tris-HCl, pH 8.0, 130 mM NaCl, and 1% NP-40. Lysates were then cleared by centrifugation and 100 μg was incubated with reaction mixtures containing Na/K phosphate buffer (0.1 mM, pH 7.5), 0.5 mM GSH (Sigma-Aldrich, G4251), 2 units/mL GSSG reductase (Sigma-Aldrich, G3664), 0.1 mM L-CySSG (Cayman, 17582), 0.2 mM NADPH (Roche, 10107824001) and 1.5 mM EDTA (pH 8.0) or with reaction buffer consisting of 137 mM Tris-HCl buffer (pH 8.0), 0.5 mM GSH (Sigma-Aldrich, G4251), 1.2 units GSSG reductase (Roche, 10105678001), 2.5 mM 2-hydroxyethyl disulfide (Sigma-Aldrich, 380474), 0.35 mM NADPH (Roche, 10107824001), 1.5 mM EDTA (pH 8.0). The reaction was prepared in a 96-well plate (final volume of 200 μL) and proceeded at 30 °C. The consumption of NAPDH was followed spectrophotometrically at 340 nm. In each sample, the spontaneous consumption of NADPH was subtracted. Data are expressed in units (U)/mg protein for lung homogenates in which 1 U equals the oxidation of 1 μmol NADPH/min.

**GSH and GSSG concentration determination.** GSH and GSSG were quantified with a luminescence-based GSH/GSSG-Glo Assay per the manufacturer's instructions (Promega, V6611).

**In situ detection of protein S-glutathionylation.** Pr-SSG was detected in situ in paraffin-embedded lung tissue as previously described[76]. After dewaxing tissue samples in three changes of xylene, they were rehydrated in 100%, 95%, and 75% ethanol. Free thiol groups were then blocked using a buffer that contained 25 mM 4-(2-hydroxyethyl)-1- piperazineethanesulfonic acid, pH 7.4, 0.1 mM EDTA, pH 8.0, 0.01 mM neocuproine (Sigma-Aldrich, N1501), 40 mM N-ethylmaleimide (Sigma-Aldrich, E1271), and 1% Triton (Sigma-Aldrich) for 30 min. After three washes in PBS, S-glutathionylated cysteine groups were reduced by incubation with 13.5 μg/mL human Grx1 (Fitzgerald, 30R-1244), 35 μg/mL GSSG reductase (Roche, 10105678001), 1 mM GSH (Sigma-Aldrich, G4251), 1 mM NADPH (Roche, 10107824001), 18 μmol EDTA, and 137 mM Tris·HCl, pH 8.0, for 20 min. After three washes with PBS, newly reduced cysteine residues were labeled with 1 mM 3-(N-maleimidylpropionyl)biocytin (MPB) (Santa Cruz, sc-216373) for 1 h. Excess MPB was removed by three washes with PBS. Next, tissue samples were incubated with 0.5 μg/mL streptavidin-conjugated Alexa Fluor 488 (Invitrogen, S32354) for 30 min. Nuclei were stained with DAPI. All steps were conducted at room temperature. Slides were photographed under an Olympus BX61 confocal microscope.

**Quantitative Determination of S-Glutathionylated Proteins Using DTNB.** S-glutathionylated proteins in lung tissue were assayed by the DTNB enzymatic recycling method as described previously[77]. After perfusing and lavaging the lung, connective tissue and hemolyzed blood were removed. Lung tissues were homogenized in 5% sulfosalicylic acid (Sigma-Aldrich, S2130), and then treated with 1% NaBH$_4$ (Sigma-Aldrich, 452882). After neutralizing with 30% metaphosphoric acid (Sigma-Aldrich, 79613), cells were centrifuged at 1000 g for 15 min. Add 20 μL supernatant to the 96-well plate. Mix equal volumes of freshly prepared DTNB (Sigma-Aldrich, 452882, 2 mg DTNB in 3 mL KPE buffer) and GR solutions [40 μL GR (250 units mL$^{−1}$) in 3 mL KPE buffer] together and add 120 μl to each well. After 30 s, then add 60 μl of β-NADPH (Sigma-Aldrich, N7505, 2 mg β-NADPH in 3 mL KPE buffer) . Immediately read the absorbance at 412 nm in a microplate reader and take measurements every 30 s for 2 min (5 readings). Calculate the rate of 2-nitro-5-thiobenzoic acid formation (change in absorbance min$^{−1}$). Determine the actual total GSH concentration in the samples by using linear regression to calculate the values obtained from the standard curve.

Preparing GSH to make the top standard concentration (26.4 nM ml$^{−1}$) and then make twofold serial dilutions for a series of standards. Preparing all solutions in 0.1 M KPE buffer (0.1 M potassium phosphate buffer with 5 mM EDTA disodium salt, pH 7.5). Keep all samples and standards on ice during the procedure and all reagents covered in aluminum foil to protect from light.

**Detection of S-glutathionylated FABP5.** Cells were lysed in NP-40 lysis buffer [50 mM Tris-HCl (pH 7.4), 150 mM NaCl, 1% Nonidet P-40, 1 mM EDTA] containing 20 mM N-ethylmaleimide (Sigma-Aldrich, E1271) and protease inhibitor cocktail (Roche, 04693132001). And the lysates were incubated with anti-glutathione (Abcam, ab19534) antibody-conjugating SureBeads Protein G

magnetic beads (Bio-Rad, 161-4023) overnight at 4 °C. Co-IP was conducted on a magnetic rack (Bio-Rad). Samples were analyzed by immunoblotting using anti-FABP5 antibody (Proteintech, 12348-1-AP, 1:500).

**Histopathology**. Lungs were fixed by intratracheal injection of 4% paraformaldehyde in PBS, maintained at 25 cm hydrostatic pressure for 5 min and then immersed in the same fixative for an additional 24 h. The left lung was embedded in paraffin and 5 μm sections were cut for H&E staining. Slides were photographed under Olympus VS200 analyzing by OlyVIA 3.1 or Nikon Eclipse Ci-S.

**Immunofluorescence staining**. Cells were washed with PBS, fixed in 4% paraformaldehyde for 20 min, and permeabilized with 0.1% Triton X-100 for 20 min. After blocking cells with 4% donkey serum for 1 h at room temperature, cells were incubated with antibody overnight at 4 °C. The next day, the cells were washed and incubated with Alexa Fluor® 488 donkey anti-rabbit IgG (Invitrogen, A21206, 1:200) or Alexa Fluor® 488 donkey anti-mouse IgG (Invitrogen, A21202, 1:200) for 1 h at room temperature, and the nuclei were stained with DAPI. Slides were photographed under an Olympus BX61 confocal microscope.

**Molecular docking**. The protein structure of FABP5 complexed with linoleic acid was downloaded from RCSB PDB database (PDBID: 4LKT). Protein structure was prepared with preparation wizard module in Schrödinger pacage (2017-04). The pka of protein and PLM was predicted with propka and Epik method[78], respectively. protonation was determined at pH7.3 according to the prediced pka. Water was removed, and the structure was then optimized with OPLS2005 force field. Then two-step docking was performed. First, the glutathione was docked to the apo form FABP5 with covalent docking CovDock method[79]; the disulfed bond formation was choosen as the reaction type. Cys127 was selected for the reactive residue. The top one pose was optimized with Prime module. The second step was that PLM was docked to the holo form FABP5-GSH. The Glide SP protocol was selected for the non-covalent docking[80]. The top one pose was minimized with OPLS2005 force file. Finally, the representation was prepared with PyMOL[81].

**Quantitative PCR (qPCR)**. Total RNA was extracted using TRIzol reagent then reverse-transcribed to cDNA using the ReverTraAce qPCR RT kit (Toyobo). qPCR was performed on the CFX96 Touch™ Real-Time PCR Detection System (Bio-Rad) using SYBR Green reagent (Roche). The primer sequences are listed in Supplementary Information file.

**Nucleofection of mouse macrophages**. $1 \times 10^6$ FABP5 KO BMDMs were nucleofected with 5 μg pXJ40-3xFlag-FABP5 WT or C127S plasmid using the Amaxa Mouse Macrophage Nucleofector Kit (Lonza, VPA-1009) following the manufacturer's instructions. Briefly, Centrifuge $1 \times 10^6$ BMDMs at 200 g for 10 min at room temperature. Discard supernatant completely so that no residual PBS covers the cell pellet. Resuspend the cell pellet carefully in 100 μl room temperature Nucleofector® Solution per sample. Combine 100 μl of cell suspension with 5 μg plasmid. Transfer cell/DNA suspension into certified cuvette. Select the appropriate Nucleofector® Program Y-001 (Amaxa Nucleofector II). Insert the cuvette with cell/DNA suspension into the Nucleofector® Cuvette Holder and apply the selected program. Add ~500 μl of the pre-equilibrated culture medium to the cuvette and gently transfer the sample into the 12-well plate (final volume of 2 ml media per well). Replace medium 6 hours post Nucleofection and add 500 ng/mL LPS to the fresh medium. 24 hours after treatment, harvest cells by using GenElute Single Cell RNA Purification Kit (Sigma-Aldrich, RNB300) and perform RNA-seq.

**RNA-seq**. RNA integrity was assessed using the RNA Nano 6000 Assay Kit of the Bioanalyzer 2100 system (Agilent Technologies, CA, USA). For RNA sequencing, a total of 3 μg RNA/sample was used as input for RNA sample preparation. Sequencing libraries were generated using the NEBNext® UltraTM RNA Library Prep Kit for Illumina® (NEB) following the manufacturer's recommendations, and index codes were added to attribute sequences to each sample. Clustering of the index-coded samples was performed on a cBot Cluster Generation System using the TruSeq PE Cluster Kit v3-cBot-HS (Illumina) according to the manufacturer's instructions. After cluster generation, the library preparations were sequenced on an Illumina Novaseq platform and 150 bp paired-end reads were generated. Raw reads were processed through quality control and genome mapping and further analyzed by the DESeq2 package (1.20.0). Library preparation, clustering, and sequencing were done by Novogene Co., Ltd.

**Nuclear and cytoplasmic fractionation**. BMDMs or COS-7 cells were first treated with or without $H_2O_2$ for the indicated times and washed in PBS. Cells were harvested and kept at 4 °C. Nuclear and cytoplasmic fractions were separated using a Nuclear and Cytoplasmic Extraction Kit (Thermo Fisher Scientific, 78833).

**Western blot**. Cells were lysed in RIPA lysis buffer. Total cell lysates were used for Western blot using antibodies against FABP5 (Cell Signaling Technology, 39926, 1:1000), FABP5 (Proteintech, 12348-1-AP, 1:500), Grx1 (Abcam, ab45953, 1:250),

β-actin (Huabio, M1210-2, 1:2000), PPARβ/δ (Santa Cruz, sc-74517, 1:500), Lamin A/C (Cell Signaling Technology, 4777, 1:2000) and Lamin B1 (Proteintech, 66059-1-Ig, 1:1000). Anti-mouse IgG (Cell Signaling Technology, 7076, 1:1000) and anti-rabbit IgG (Cell Signaling Technology, 7074, 1:1000) were the secondary antibodies.

**ROS assay**. Intracellular ROS levels were assessed using a 2′,7′-dichlorofluorescein diacetate (DCFH-DA) fluorescent probe (Sigma-Aldrich, D6683) and analyzed on an ACEA NovoCyte Flow Cytometer. Data were analyzed using FlowJo software (Tree Star).

**Chromatin Immunoprecipitation**. Chromatin Immunoprecipitation assay (ChIP) was performed using Pierce Magnetic ChIP Kit (Thermo Fisher Scientific, 26157) according to manufacturer's instructions. And DNA-bound protein was immunoprecipitated using anti-FABP5 (Proteintech, 12348-1-AP) and anti-IgG (Thermo Fisher Scientific, 26157) antibodies. DNA enrichment was quantified by qPCR. The following primers were used:

mFIAF-PPRE (forward primer 5′-GGCAGACCCAGAAAGATGG-3′, reverse primer 5′-CCGATTGGATGAGAGGAAAG-3′)

mADRP-PPRE (forward primer 5′-GCTGGGGATTACAGACCAGA-3′, reverse primer 5′-TCTTGGGGTTTTGGAAAATG-3′)

**Transactivation assays**. COS-7 cells were cultured in 12-well plates and co-transfected with PPRE X3-TK-luc, pRL-TK, pXJ40-3xFlag-FABP5 WT, or C127S, according to the Lipofectamine reagent protocol (Invitrogen, 11668-019). Raw264.7 cells were nucleofected with PPRE X3-TK-luc and pRL-TK by using Amaxa Mouse Macrophage Nucleofector Kit (Lonza, VPA-1009) according to the protocol of the manufacturer. PPRE X3-TK-luc is a reporter construct containing three copies of PPRE (PPRE is peroxisome proliferator response element) upstream of a thymidine kinase promoter fused to a luciferase gene. PPRE X3-TK-luc was a gift from Bruce Spiegelman (Addgene plasmid #1015; http://n2t.net/addgene:1015;RRID:Addgene_1015)[82]. Six hours following transfection, medium was replaced by fresh cell culture medium. Eighteen hours later, cells were serum starved for 8 h, then exposed to $H_2O_2$ (200 μM), GW0742 (1 μM), or both for 24 h, and luciferase activity was determined by Dual-Luciferase® Reporter Assay System (Promega, E1910) using Promega GloMax 20/20 (Promega).

**Recombinant protein purification**. Recombinant protein purification was performed as described previously[83]. Wild-type FABP5 was cloned into pGEX6P1 and expressed as glutathione-S-transferase (GST) fusion proteins with a TEV protease cleavage site in between. FABP5 C127S was generated with a Fast Mutagenesis System kit (Transgen, FM111) based on pGEX6P1-FABP5 WT. GST fusion proteins were expressed in E. coli BL21 (DE3) at 16 °C to achieve maximal soluble expression. Cells were collected by centrifugation and washed three times with cold PBS. The cells were lysed by sonication in lysis buffer (20 mM Tris–HCl, pH 7.5, 500 mM NaCl, 1 mM EDTA, 0.5% Triton-X100, protease inhibitor cocktail from Roche) and centrifuged at 12,000 g for 15 min. GST-Sepharose resin (0.2 mL; GE Healthcare) pre-equilibrated with 20 mL TEV protease cleavage buffer (10 mM Tris–HCl, pH 8.0, 150 mM NaCl, 0.1% NP-40, 1 mM DTT) was added to the supernatant and rotated at 4 °C for 2 h. Next, beads were washed three times with TEV protease cleavage buffer, and then the recombinant protein was eluted from the resin by incubation overnight at 4 °C with 10 μg/mL TEV protease to cleave off the desired protein from the GST tag, which was still bound to the GST-Sepharose resin after the overnight cleavage reaction.

**In vitro pull-down assays**. The recombinant protein with Flag tag was eluted from the resin by incubation overnight at 4 °C with 10 μg/mL TEV protease to cleave off the desired protein from the GST tag. After purification, 2 μg of recombinant FABP5-3xFlag or FABP5 C127S-3xFlag protein were incubated with anti-Flag (Cell Signaling Technology, 14793 S) antibody-conjugating SureBeads Protein A magnetic beads (Bio-Rad, 161-4013) overnight at 4 °C. After washing three times, FABP5 or FABP5 C127S-conjugating magnetic beads were treated with 1 mM GSSG for 15 min. COS-7 cells, seeded onto 10-mm dishes, were lysed with a lysis buffer (50 mM Tris-HCl, pH 7.4, 150 mM NaCl, 1% Triton X-100, 1% sodium deoxycholate, 0.1% SDS, 1 mM EDTA and protease inhibitor cocktail) by pipetting and sonication. After centrifugation, cleared cell lysates were incubated with GSSG treated FABP5 or FABP5 C127S-conjugating magnetic beads for 4 hours at 4 °C. Following incubation, the beads were washed five times with lysis buffer and protein samples were eluted by boiling in 2× SDS sample buffer for Western blot analysis.

**Fatty acid binding assay**. Purified recombinant FABP5 protein (1 μg) was incubated with 1 mM GSSG (Sigma-Aldrich, G6654) or GSH (250 μM) plus $H_2O_2$ (250 μM) for 15 min, then mixed with 50 μM Biotin-Linoleic acid (Cayman, No. 10010623) or Biotin-Dodecanoic acid (Cayman, No. 25714) in the absence or presence of BMS309403 (5 mM) or linoleic acid (1 mM) or Grx1 [Human Grx1 (13.5 μg/mL), GSSG reductase (35 μg/mL), GSH (1 mM), EDTA (18 μmol), NADPH (1 mM), or Tris-HCl (137 mM, pH 8.0)] for 30 min. Samples were incubated with Streptavidin-coupled Dynabeads (Invitrogen, 65801D) for 30 min at room temperature on a magnetic rack (Bio-Rad) and were analyzed by

immunoblotting using anti-FABP5 antibody (Proteintech, 12348-1-AP; Cell Signaling Technology, 39926).

**Enrichment of Protein SSG and LC–MS/MS**. Enrichment of Protein SSG and LC–MS/MS were performed as previously described[40,41]. BMDMs were harvested form 6 mice, respectively. After $H_2O_2$ exposure for 15 min, BMDMs were rinsed twice and harvested in lysis buffer (250 mM HEPES, 1% Triton X-100, and 0.1% Protease Inhibitor Cocktail, pH 7.0) containing 100 mM N-ethylmaleimide. Each group used a pooled sample of six BMDM samples from different mice. The lysates were centrifuged at 13,000 g for 10 min at 4 °C and the soluble protein fraction was retained. The alkylation reaction was carried out at 55 °C in darkness for 30 min in the presence of 2% SDS. To obtain protein, 6 volumes of pre-cooled acetone were added and kept at –20 °C for 4 h, then washed three times with pre-cooled acetone. Purified proteins were re-suspended in buffer (250 mM HEPES, 8 M urea, 0.1% SDS, pH 7.5) and replaced with buffer I (25 mM HEPES, 1 M urea, pH 7.5) three times in an ultrafiltration centrifugal tube (Merck, Amicon Ultra-0.5 mL, 10 KDa). The protein concentration was determined using the bicinchoninic acid assay (BCA).

For the reduction of SSG-modified proteins, 480 μg of the alkylated samples were prepared at a final concentration of 1 μg/μL in 25 mM HEPES containing 1 M urea (pH 7.5) followed by the addition of 2.5 μg/mL Grx1M (C14S mutant from *Escherichia coli*, Cayman, No. 11534), 0.25 mM GSSG, 1 mM NADPH, and 4 U/mL GR. Samples were incubated at 37 °C for 10 min, immediately placed on ice, and transferred to a 0.5-mL Amicon Ultra 10 KDa filter. Excess reagents were removed by buffer exchange with 3 × 8 M urea (pH 7.0) resulting in a final volume of 30–40 μL. The protein concentration of the deglutathionylated samples was measured by the BCA assay before enrichment.

Three hundred micrograms of reduced samples were taken for each enrichment and the final volume readjusted to ~120 μl of 25 mM HEPES buffer containing 0.2% SDS and loaded to Handee Mini-Spin columns containing 30 mg of preconditioned thiopropyl sepharose 6B resin. Enrichment was carried out in a thermomixer at room temperature with shaking at 850 rpm for 2 h.

The samples were processed according to the manufacturer's protocol for the 6-plex iodoTMT kit (Thermo Fisher Scientific). The protein samples were further diluted and digested with trypsin (Promega) following the manufacturer's instructions.

After resin washing, tryptic digestion, IodoTMT-labeling, and DTT elution[42], the peptides were dissolved in a final volume of ~25 μL water and 20 mM DTT was added to prevent the oxidation of cysteines prior to LC-MS/MS analysis.

The MS/MS data were processed using the MaxQuant search engine (v.1.5.2.8). Tandem mass spectra were searched against the SwissProt mouse database (16839 entries) concatenated with a reverse decoy database. Trypsin/P was specified as the cleavage enzyme allowing up to 2 missing cleavages. The mass tolerance for precursor ions was set at 20 ppm in First search and 5 ppm in Main search, and the mass tolerance for fragment ions was set at 0.02 Da. Acetylation on protein N-terminal, oxidation on Met, and iodo TMT-6plex var were variable modifications. Iodo TMT-6plex quantification was performed. The false discovery rate (FDR) was adjusted to <1% and the minimum score for peptides was set at >40. IodoTMT-based proteomics and analysis was done by Jingjie PTM Biolab Co., Ltd. (Hangzhou).

**Plasmids**. Wild-type 3xFlag-FABP5 was constructed by cloning cDNA into the pXJ40-3xFlag vector containing the amino-terminal Flag tag using HindIII and KpnI sites. Other FABP5-related mutants were generated with the Fast Mutagenesis System kit (Transgen, FM111) on the basis of their respective WT constructs. pcDNA3.1-3xFlag-Grx1 was made by inserting Grx1 cDNA into the pcDNA3.1-3xFlag vector using BamHI and XhoI sites.

**Sequence alignments**. Multiple sequence alignments were conducted using the ClustalX2 program, and the GeneDoc 2.7 program was used to edit the sequence alignment.

**Statistical analysis**. All results are presented as the mean ± SEM. Two-tailed unpaired Student's t-test (for two group comparison) or one-way ANOVA followed by Tukey's post-hoc test (for multi-group comparison) were performed using GraphPad Prism 8; P value < 0.05 was considered statistically significant. All experiments were repeated independently at least three times with similar results.

**Reporting summary**. Further information on research design is available in the Nature Research Reporting Summary linked to this article.

## Data availability

The RNA-seq data generated in this study have been deposited in the NCBI gene expression Omnibus (GEO) under accession code GSE182238. The raw Mass Spectrum data generated in this study have been deposited in the PRIDE database under accession code PXD027965. Mass Spectrum data that supporting our findings have been deposited in Supplementary Data 1, 2. Structural models used in Fig. 6a, e, f were accessed from the Protein Data Bank under the accession code 4LKT. Structural models used in Fig. 6g were accessed from the Protein Data Bank under the accession codes 4AZM. The remaining data are available within the Article, Supplementary Information or Source Data file. Source data are provided with this paper.

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

## Acknowledgements

The authors thank Dr. Yuanfu Xu (State Key Laboratory of Experimental Hematology, Institute of Hematology and Blood Diseases Hospital, Chinese Academy of Medical Sciences and Peking Union Medical College, Tianjin, China) for providing Grx1$^{fl/fl}$ mice (C57BL/6 background); and Xueping Zhou (Laboratory Animal Center, Zhejiang University, Hangzhou, China) for mouse care. We thank Dr. Iwao Ojima (Institute of Chemical Biology and Drug Discovery, Stony Brook University, Stony Brook, NY, USA) for generously providing the SBFI26 reagent; Dr. Ximing Xu (Innovation Platform of Marine Drug Screening & Evaluation, Qingdao National Laboratory for Marine Science and Technology, Qingdao, Shandong, China) for his help in computational docking analysis; Danlu Jiang (Zhejiang University School of Medicine) for her help in RNA-seq analysis; Shuangshuang Liu (Core Facilities, Zhejiang University School of Medicine, Hangzhou, China) for microscopy and instrumentation support; and Jingjie PTM Biolab Co. Ltd. for assistance with the mass spectrometry analysis. This work was supported by the National Natural Science Foundation of China (31870901 to X.Z. and 81873418 to Y.K.), the Key Research and Development Project of the Ministry of Science and Technology of China (2016YFA0501800 to Y.K.), Natural Science Foundation of Zhejiang Province (LY18H010001 to X.Z.).

## Author contributions

X.Z., Y.K., and Y.G. wrote the manuscript and designed figures. X.Z., Y.K., and Y.G. designed experiments. Y.G., Y. Liu, S.Z., W.X., Y. Li and P.Z. conducted experiments and acquired data. Y.G., X.Z., Y. Liu, Y.K., H.C. and D.W. analyzed and interpreted data. X.Z. and Y.G. edited the manuscript. All authors approved the final manuscript.

## Competing interests

The authors declare no competing interests.
