## [Peer Review File · Nature Communications]

REVIEWER COMMENTS

Reviewer #1 (Remarks to the Author):

In the article “S-glutathionylation of FABP5 protects against acute lung injury by suppressing inflammation in macrophages”, the authors provide convincing evidence that glutathionylation of Cys127 of FABP5 is required for its nuclear translocation, fatty acid binding and subsequent activation of PPARbeta in macrophages. Grx1 was identified as responsible for the de-glutathionylation of FABP5 and selective knockout of Grx1 in macrophages resulted in increased glutathionylation and subsequent nuclear localisation of FABP5. The glutathionylation of Cys127 of FABP5 resulted in PPARbeta activation and prevented the LPS induced inflammatory response (as measured by IL-1beta, IL-6 and TNFalpha) in BMDMs.

Overall the article presents novel data on a specific redox modification of FABP5 that regulates an important physiological mechanistic response to stress. The methodology employed throughout the article is technically sound and the results would support the authors hypothesis. A wide range of techniques are employed in the article include using selective knockout models, a redox proteomic approach to identify Grx1 targets, RNA-seq analysis and qPCR, blotting, Co-IP, microscopy, flow cytometry.

I only have some minor comments that the authors may wish to clarify or address.

1. In the initial experiment the authors subject mice to normoxia or hyperoxia to investigate the role of hyperoxic acute lung injury. It is interesting that some of the general results obtained in this hyperoxic experiment are very similar to those mentioned in the introduction with exposure to cigarette smoke (25) and COPD (27) i.e. reduction in Grx1 expression and decreased GSH:GSSG ratio, could the authors comment in the discussion on whether hypoxia would provoke similar responses and why?
2. The authors use a H₂O₂ concentration of 200uM for 10 mins to identify glutathionylated proteins and for other experiments, why this dose and time?
3. The redox proteomic approach is well performed but the full list of identified glutathionylated peptides and proteins should be included in a supplementary file. Potentially further substrates of Grx1 are of great value and if I can read Fig3C correctly (resolution needs improving), Cys249/250 of PTBP1 would appear to be the top target of Grx1 in this system. The resolution of the MS/MS fragmentation spectra in Fig 6D needs to be improved, as this peptide contains both Cys120 and Cys127 of FABP5 to identify which one is glutathionylated. The quantification of the glutathionylation of all Cys residues reported as glutathionylated in FABP5 could be presented in the supplementary information. The immunolocalisation study in Fig6E would indicate that only the C127S mutation prevented nuclear accumulation of FABP5 but considering the 2 Cys residues on the peptide detected and its proximity to the GSH binding domain perhaps Fig6F could include the C120S mutation?

4. The formation of FABP5 dimers and oligomers in Fig4E non-reducing and reducing gels would suggest they are via disulphide bridges dependent on the other Cys residues in FABP5. But in Fig4E and 4I we do not see any FABP5 monomer as glutathionylated, does this mean that the protein needs to be in the dimer/oligomer formation for glutathionylation? As the authors already have the mutant forms of FABP5 with each Cys residue mutated, it would be very quick and might be interesting to determine which ones are required for dimer/oligomer formation and whether this is required for glutathionylation of Cys127.

5. Why in Fig4I is there a dimer of FABP5 that is glutathionylated in the DTT treated negative control lane?

6. Finally, could the authors include in the discussion a brief line on the mechanism of how PPARbeta activation results in an anti-inflammatory response.

Overall a very interesting and novel article that identified a specific redox mechanism of FABP5 responsible for the potential prevention of acute lung injury.

Reviewer #2 (Remarks to the Author):

Overall: This manuscript addresses the important area of anti-inflammatory regulation in acute lung injury, focusing on a novel regulatory mechanism involving site-specific S-glutathionylation of FABP5 on Cys-127 and the role of Grx1 in reversing this oxidative modification. The overall data support the interpretation that S-glutathionylation of FABP5 on Cys127 enhances fatty acid binding and nuclear translocation by facilitating cytosolic exposure or stabilization of the NLS that is necessary for nuclear import and suppression of inflammatory responses. Many modern techniques have been utilized in this study, including quantitative redox proteomics, mouse models with cell-specific molecular modifications, and molecular docking analysis. Overall this is an impressive study that provides fresh insights. However, there are substantial shortcomings that detract from the quality of the manuscript, including organization of the manuscript, presentation of some of the data, quality/accuracy of certain items of data, interpretation of some of the results, accuracy/clarity of some descriptions in the narrative, and level of resolution of many of the figures. Specific comments are delineated below.

Specific comments:

1. **Abstract, line 9:** “Here we report that Grx1-mediated S-glutathionylation in macrophages alleviates the inflammation of acute lung injury.” **Comment:** Although it appears to be a minor problem with choice of words, “Grx1-mediated” should be replaced with “Grx1-regulated,” because “mediated” suggests that *Grx1 is catalyzing the formation of protein-SSG*. Instead, as the authors understand, the primary role of Grx1 is to catalyze de-glutathionylation of protein-SSG. There are unique circumstances where Grx1 may catalyze S-glutathionylation (e.g., when the GSH/GSSG ratio is extremely low (≤ 1) or when hydroxyl radicals are abundant); but such considerations do not apply here. There are several places in the manuscript where this ambiguity should be corrected (see below).
Results, lines 154-156, “These data demonstrate that Grx1-mediated [replace with regulated] protein S-glutathionylation in macrophages plays a protective role in acute lung injury.”
2. **Introduction, lines 45-47,** “Grx1 is involved in the regulation of various cellular functions, maintains a stable cellular redox state, and produces a reducing environment within the cell under varying environmental conditions.” **Comment:** “produces a reducing environment” is an *inaccurate* statement because it implies that the enzyme changes the position of equilibrium. Instead Grx1 facilitates homeostasis by catalyzing primarily de-glutathionylation. The sentence could read, “... serves to maintain a reducing environment...”
3. **Introduction, lines 85-86.** “Here, we report that macrophage-restricted ablation of Grx1 confers protection against acute lung injury in mice. **Comment:** This finding and others such as LPS-induced diminution of Grx1 in macrophages should be included in the Abstract, because they help convey the regulatory role of Grx1 as de-glutathionylation catalyst, namely absence or diminution in Grx1 content generally leads to increased glutathionylation.
4. **Results, section one, focus on hypoxia** – The initial part of the Results section is not well integrated with the major portion of the manuscript, and it stands out as being incomplete, having only implicated Grx1 and S- glutathionylation in general, without the mechanistic depth of the study of LPS-induced activation of macrophages presented in the rest of the manuscript. The one commonality is the observation that *loss of Grx1 substantially diminished the lung damage caused by hyperoxia*. The loss of Grx1 and consequent enhancement of protein glutathionylation as a protective mechanism is the common thread that should be developed more effectively in the first section as a segue to the main body of the manuscript, which nicely evolves the specific role of S-glutathionylation of FABP5-Cys127.
5. **Results: problematic GSH/GSSG ratios (Fig. 1A, Fig. 2B).** The authors state (lines 101-102) that the GSH/GSSG ratio “provides a reliable estimate of cellular redox status.” This can be true if the ratio is measured accurately. Typically the GSH/GSSG ratio is expected to be near 100 or greater for non-oxidatively stressed cells; however, the value for the normal oxygen control (Fig. 1A, NOX) is less than 10! Such a value is usually interpreted as severe oxidative stress. Likewise, the value shown for the

unstimulated macrophages is 16 (Fig. 2B), whereas the usual value is >100. For example, the study cited below reports a resting value of 200.

K. Alam, et al., Glutathione-Redox Balance Regulates c-rel-Driven IL-12 Production in Macrophages: Possible Implications in Antituberculosis Immunotherapy, *J. Immunol.* 2010, 184 (6) 2918-2929; DOI: <https://doi.org/10.4049/jimmunol.0900439>.

What precautions were taken to prevent oxidation of GSH to GSSG during processing of samples? Did the authors run a standard curve of various GSH/GSSG ratios to confirm that a range of values (e.g., 1-100) could be measured accurately with the luminescence-based assay that they used (Promega)?

6. Results: problematic representation of Grx1 activity (Fig. 1E). There is a good quantitative correlation of the diminution of Grx1 activity (Fig. 1E) with the diminution of its expression (Fig. 1C) in terms of fold-change, but expression is shown as relative (1.0 = 100%), whereas activity is presented as specific activity (units/mg). The value of ~100 units/mg shown in Fig. 1E is not possible for a crude tissue sample or cell homogenate, because the maximum specific activity of pure Grx1 is ~100 units per mg. Should the y-axis of Fig. 1E read as "% Activity?"

It is remarkable that the authors recognize the specificity of Grx1 for selectively reducing glutathione-containing mixed disulfides and they rely on this specificity for their proteomic mass spec procedure in which Grx1 is used to remove GSH from protein-SSG. However, under Methods for Grx1 activity they have referenced two articles; one (28) does not describe an assay for Grx activity and the other (73) describes an old assay of thioltransferase activity that utilizes non-specific, non-glutathione containing substrates. The appropriate assay for Grx1 utilizes cysteine-SSG as substrate, which serves as a mimic for all protein-SSGs, without the potential steric hindrance due to surrounding protein residues (see e.g., the reference cited below which characterizes Grx2 relative to Grx1, using Cys-SSG as substrate).

Gallogly MM, Starke DW, Leonberg AK, Ospina SM, Mieyal JJ., Kinetic and mechanistic characterization and versatile catalytic properties of mammalian glutaredoxin 2: implications for intracellular roles. *Biochemistry.* 2008 Oct 21;47(42):11144-57. doi: 10.1021/bi800966v.

7. Results, Fig. 2/ Why were several different approaches used to assess changes in protein glutathionylation (Western blot and DTNB recycling *versus* immunocytochemistry)?
8. Results, lines 140-142, "Based on the above results, we hypothesize that Grx1 in macrophages plays a key role in the pathogenesis of acute lung injury." Comment: This statement is not readily understood without reading the rest of the manuscript, where the role of Grx1 in removing GSH from a specific protein-SSG interferes with the protective effect of that protein-SSG. Without this understanding it appears that you are hypothesizing that Grx1 action leads *directly* to pathogenic events. Some more elaboration is necessary here.
9. Results, Fig. 3C: What does treatment / control ratio mean in Fig. 3C - what is being measured? The legend does not provide an explanation.
10. Results, lines 191-192: Why was the artificial COS7 construct substituted for macrophages to study the glutathionylation of FABP5? How does this COS7 model compare to the natural abundance of Grx1 and FABP5 in macrophages? It is important to be more transparent here. Was the COS7 construct used because it was not possible to document FABP5-SSG directly with macrophages? Rather, glutathionylation of FABP5 in macrophages is only inferred by observation of oligomerization of the protein, presumed to occur *via* inter-protein disulfide bonds facilitated by the transitory formation of FABP5-SSG as a precursor?
11. Results, Fig.6E: It is difficult to discern any differences among the panels. More explanation is necessary, along with a quantitative analysis of the perceived differences. Fig. 6F provides more definitive data, but the approach is different (Western blot) and the experiment is conducted with the artificial COS7 construct.
12. General note about the figures: The resolution is relatively poor and the very small labels are often difficult to read even at higher magnification.

13. Results, lines 284-286 (Fig. 6I), “In response to H₂O₂, FABP5 WT was strongly glutathionylated, whereas glutathionylation was significantly decreased with FABP5 C127S (Fig. 6I, S6C).” Comment: It would be very helpful to quantify the results of Fig. 6I. Was the extent of glutathionylation diminished by more than 25%, indicating a greater propensity for glutathionylation of C127 relative to the other 3 sites?

Note: Fig. S6 does not have a panel C.

14. Results, lines 338-340, “As shown in Fig. 7J, G114R enhanced the S-glutathionylation of FABP5, suggesting a potential role of FABP5 glutathionylation in disease.” Comment: This statement appears to contradict the key observations/conclusions of this study, so it needs to be re-phrased or further explained. If S-glutathionylation of FABP5 leads to diminution of the inflammatory response and G114R-mutation of FABP5 leads to greater glutathionylation of FABP5, then the mutation should be more anti-inflammatory (*i.e.*, protective), inhibiting rather than promoting disease.

15. Discussion, lines 370-376: This paragraph does a better job of tying hypoxia to the rest of the manuscript. But it reveals a deficiency in mechanistic detail, only implicating that S-glutathionylation and Grx1 are involved, but not pursuing how they are involved and what are the key molecular targets. This shortcoming contrasts with the depth of the second focus of the manuscript on LPS-activation of macrophages.

16. Discussion, lines 379-380, “Not surprisingly, loss of Grx1 in macrophages relieved LPS-induced acute lung injury.” Comment: This may not be surprising to the authors or to a rather narrow audience. However, general knowledge conveys that loss of Grx1 exacerbates oxidative stress and allows increased protein-SSG formation, which more often than not decreases the function of proteins or accelerates their degradation. *The difference in acute lung injury is that particular S-glutathionylation events are protective rather than deleterious. This distinction needs more emphasis in many places in the manuscript.

17. Discussion, lines 416-418, “As to FABP5 ... its S-glutathionylation has not been reported previously.” Comment: It is important in this context to acknowledge that S-sulfenylation of FABP5 on Cys127 *has been reported*. This is significant, because it suggests that formation of FABP5-Cys127-SOH by reaction of FABP5 with H₂O₂ likely serves as a precursor to FABP5-Cys127-SSG (see citation below).

Yang J, Gupta V, Carroll KS, Liebler DC. Site-specific mapping and quantification of protein S-sulphenylation in cells. *Nat Commun.* 2014 Sep 1;5:4776. doi: 10.1038/ncomms5776. PMID: 25175731; PMCID: PMC4167403.

Indeed, the reaction (Pr-SOH + GSH → Pr-SSG + H₂O) has been identified as a likely common mechanism of S-glutathionylation of proteins. (As described in reference 13, cited in this manuscript).

18. Discussion, lines 448-451, “These data suggest that the glutathionylation of FABP5 acts as an anti-inflammatory mediator in macrophages during ALI and we propose a new mechanism by which macrophages prevent self-hyperactivation under oxidative stress.” Comment: The regulatory role of Grx1 in this context implicates Grx1 as a target of anti-inflammatory drug development, as suggested previously (see citation, below).

Gorelenkova Miller O, Cole KS, Emerson CC, Allimuthu D, Golczak M, Stewart PL, Weerapana E, Adams DJ, Miayal JJ. Novel chloroacetamido compound CWR-J02 is an anti-inflammatory glutaredoxin-1 inhibitor. *PLoS One.* 2017 Nov 20;12(11):e0187991. doi: 10.1371/journal.pone.0187991. PMID: 29155853; PMCID: PMC5695812.

19. Methods, lines 585-592 - Fatty Acid Binding Assay: This appears to be only a qualitative assay. Was FABP5-C127S tested in this assay? What is the concentration dependence of the binding? K_D value?

Minor Comments:

- Results, line 104, “Compared with the NOX group, a **markedly** higher expression level of GSH synthetase ... (Fig. 1B). A ~1.5-fold increase would usually not be described as “markedly higher.” It is a relatively modest increase.
- Results (Fig. 1F, 1G) - It is remarkable that there is no *acute* lung damage associated with KO of Grx1 under NOX conditions. Are there long-term effects?
- Results, lines 149-151 (Fig. 2G) – It would be useful here to describe the Grx1^{fl/fl}LysM^{cre} mouse model and its utility.
- Discussion, lines 359-362, “Further experiments showed that **Cys-127-mediated S-glutathionylation of FABP5** promoted its fatty acid binding and nuclear translocation, activating PPAR β/δ and inhibiting macrophage inflammation.” This is phrased awkwardly. It should be restated as, “... S-glutathionylation on the Cys-127 residue of FABP5 promoted”
- Discussion, lines 399-401, “The S-glutathionylation of GAPDH promotes the formation of a disulfide bond between Cys150 and Cys154” Note: The cited reference 7 features the roles of **Cys152** and **Cys156**.

Reviewer #3 (Remarks to the Author):

Author:

The paper presents a role for oxidative stress-induced protein S-glutathionylation in regulation of acute lung injury (ALI). The authors report that glutaredoxin-1 (Grx1)-regulated S-glutathionylation in macrophages mediates inflammation associated with ALI. Grx1 KO mice are protected from inflammation in mouse models of acute lung injury. Using a proteomics approach, the authors show that the fatty acid-binding protein 5 (FABP5) is susceptible to S-glutathionylation under oxidative conditions; and that S-glutathionylation of Cys127 in FABP5 activates its fatty acid binding ability and nuclear translocation. RNA-seq data confirmed that S-glutathionylation of FABP5 inhibited the LPS-induced inflammation in macrophages. FABP5 S-glutathionylation activates PPAR β/δ in macrophages. The authors concluded that FABP5 S-glutathionylation is a novel molecular mechanism by which macrophages control inflammation in ALI.

Major concerns.

After establishing the phenotype of Grx1 KO in the hyperoxia model, it is not clear why the authors switch to an LPS model for assessment of the phenotype of macrophage conditional Grx1 deficient mice.

Given that hyperoxia exposure and LPS, as used for the in vivo modeling in this paper, are also commonly used as in vitro model reagents, it is not clear why the authors implemented hydrogen peroxide for the biochemical characterization in the in vitro model.

After establishing the phenotypes of the KO and conditional macrophage grx1-deleted mice using in vivo models the authors then switch to model in vitro oxidative stress models to establish that FABP5 is a target for glutathionylation in vitro (in macrophages) and present associated anti-inflammatory mechanisms involving activation of PPARs. This is obviously the fusion of two independently derived data sets, and more attention should be made in describing the logical transition between the two approaches.

Final in vivo characterization is performed to demonstrate the phenotype of FABP5 knockout mice as susceptible to inflammation in ALI. As additional final validation, it would be helpful if the authors could also demonstrate using the in vivo models, either hyperoxia or LPS-induced injury, that FABP5 is glutathionylated in the lung tissue or lung macrophages of exposed animals.

Some of the immunoblot panels eg 5 A-C, 5E-G, and 6, F, J have resolution issues or are presented with high background.

Figure 7 is largely illegible in part due to small size and poor resolution. Please reformat especially Figs 7 A-C for clarity.

Similarly, Fig 6 panels should be provided at adequate size and resolution. Fig. 6B data is illegible in current presentation.

Abstract. Define Grx1 in abstract.

The abstract mainly highlights the biochemical characterizations. Please add a description of the phenotypes of Grx1 KO and of macrophage-specific Grx1 KO, to the abstract.

Abstract “Here we report that Grx1-mediated S-glutathionylation in macrophages alleviates the inflammation of acute lung injury.” This sentence may be misleading. Please clarify that S-glutathionylation is protective and Grx1 is counterregulatory and responsible for deglutathionylation.

Line 85 “macrophage-restricted ablation of Grx1”... suggest editing as “conditional macrophage-specific genetic deletion of Grx1”

Fig 1E. Grx1 is established a pathogenic mediator in hyperoxia, in studies of the Grx1 KO which have reduced inflammation. However, in wt mice Grx1 enzyme activity is actually reduced by hyperoxia by about 50%. Please clarify how this concurs with the hypothesis that Grx1 promotes inflammation in hyperoxia by antagonizing protein glutathionylation.

Results line 107. Please define these genes for the reader.

The methods section describes the exposures to Grx1^{fl/fl} LysM^{cre} or FABP5^{fl/fl} LysM^{cre} mice, but a section on mouse breeding appears to be missing. Also, there was no description of the source of Grx1 KO mice described in Figure 1. Also provide details of Fabp5 conditional KO mice presented in Fig.7. Please describe all mouse breeding strategies, induction, and source of materials in the methods section. Also provide information on animal care and compliance with institutional guidelines in Methods section as well.

Reviewer #4 (Remarks to the Author):

Review for NCOMMS-20-48677:

S-glutathionylation of fatty acid-binding protein 5 protects against acute lung injury by suppressing inflammation in macrophages

Major Comments:

1. The authors showed that the glutathionylation of FABP5 promotes the formation of dimer and oligomers of FABP5 which were gradually increased in a concentration-dependent manner upon H₂O₂ exposure. Which state predominantly exists in macrophages in response to oxidative stress? Each form is functionally the same? Which form is involved in translocation from the cytosol to the nucleus to activate PPAR β/δ -mediated signaling pathway and to suppress inflammation?

Monomer and/or oligomer FABP5 directly interact(s) with PPAR β/δ to activate transcription? Is the glutathionylated FABP5 contained in the transcriptional complex consisting of PPAR β/δ , coactivators (PGC-1 α/β) and other cofactors? How do the glutathionylated FABP5 monomer and/or oligomers transfer fatty acid (ligand) to PPAR β/δ in the nucleus? The authors should conduct ChIP assay and/or pull-down assay to evaluate whether direct protein-protein interactions between S-glutathionylated FABP5 and PPAR β/δ or other cofactor(s) would be detected in the nucleus.

2. Although the authors showed that FABP5 glutathionylation controls macrophage inflammation by activating PPAR β/δ , molecular mechanisms underlying anti-inflammation by the glutathionylated FABP5 in macrophages remain unclear. The authors should describe it in more detail.

3. The overexpression of FABP5 in COS7 cells might not be appropriate to characterize the function of FABP5 in macrophages under oxidative stress, because expression level of FABP5 and status of its oligomer formation in COS7 cells were not comparable with those in macrophages (RAW264.7 and BMDM).

4. Cells used in the present study were exposed to H₂O₂ at various concentrations (200 μ M to 500 μ M) under oxidative stress. The authors should explain what is the significance of the experiments conducted at different concentration of H₂O₂.

Minor Comments:

1. In Fig.1, 4, 6 and 7, molecular weight markers for proteins should be described in the western blotting data.

2. In Fig. 1F and 2G regarding histological analysis of lung, the images only in a limited area are not convincing. The authors need to show a wide field of view at a lower magnification.

3. In Fig. 4F, although glutathionylated FABP5 forms dimer and oligomer, the authors used the monomeric structure to build glutathionylated FABP5 models. The FABP5 dimer adopts a domain swapping structure (PDB ID: 4AZM). The glutathionylated FABP5 model should be also constructed by using the dimer structure along with the monomer.

4. In Fig. S4 legend, the sentence “mRNA levels of Fabp3 (A), Fabp4 (B), and Fabp7 (C) in RAW264.7 cells” should be changed to “mRNA levels of Fabp3 (B), Fabp4 (C), and Fabp7 (D) in RAW264.7 cells”.

5. In Fig. 5, although BMS309403 is generally used as a selective FABP4 inhibitor (Bioorg. Med. Chem. Lett. 17, 3511, 2007 and Nature 447, 959, 2007), the authors used two compounds, BMS309403 and SBF126, as FABP5 inhibitors. Why did the authors use BMS309403 as FABP5 inhibitor? What are differences between the two inhibitors? Reference regarding these inhibitors should be described. Although BMS309409 is used as FABP5 inhibitor in 222 line, BMS309403 is used in line 224 and Fig. 5. The authors should describe which is correct.

6. In Fig. 5C, EDTA (18 μ mol) should be changed to description of concentration.

7. In Fig. 5D, the images of only in a limited area analyzed by the immunofluorescence staining are not convincing, and the differences between the groups are not clear. The authors need to quantify positive (FABP5 nuclear translocated) cells in each group.

8. In Fig. 5G, all the bands detected in the nucleus of BMDM are weak and unclear.

9. In Fig. 7A and 7C, the font size is too small to read.

10. As mice null for one FABP compensate by increasing expression of other FABPs, careful interpretation of the data is required when using FABPs KO mice. Therefore, the authors need to describe detailed information of FABP5 KO BMDMs (Fig. 7A-D) and Fabp5^{fl/fl}LysM^{cre} mice (Fig. 7F and 7G). At least, the expression levels of other FABPs should be shown.

11. The abbreviations should be used consistently throughout the manuscript. For example, ALI or “acute lung inflammation” is mixed together in the text (lines 62, 86, 91 and 449). In addition, Pr-SSG (line 34) and PSSG (line 510), UFA (line 953) and SFA (line 958) were used in Fig. 5 legend, ULCFA and SLCFA were used in the text. They should be revised correctly.

12. In line 249, mainly should be changed to abundantly, because FABP5 is also abundantly expressed in keratinocytes, adipocytes, T cells, and cancer cells.

13. In line 267, “FABP5” should be changed to “Homo sapience FABP5”.

14. In line 318, ERR α should be added, because the paper concerning FABP5-ERR α crosstalk in the nucleus was published recently (Oncotarget 9, 31753-31770, 2018).

15. In lines 336-340, mutation of FABP5 (G114R) is found in the diseases accompanied by inflammation? What is the relationship between this mutation and oxidative stress and/or inflammation?

16. In line 358, “the fatty acid-binding protein” might be deleted.

17. In lines 398-405, the part regarding S-glutathionylation of GAPDH might not be necessarily described in Discussion.

18. Materials and Methods is partly incomplete. More detailed information of all the mice, reagents and procedures used in this study should be provided. For example, the experiments regarding PPRE-driven reporter assay, nucleofection, preparation of FABP5 KO BMDM with overexpression of FABP5 C127S and FABP5 WT, ELISA and molecular docking should be described in more detail.

In conclusion, although the findings reported here are potentially quite interesting and significant, sufficient mechanistic details and major revision are needed to substantiate the findings at this time.

Ms. No. NCOMMS-20-48677A

Ms. Title: S-glutathionylation of fatty acid-binding protein 5 protects against acute lung injury by suppressing inflammation in macrophages

Point-by-point responses to the reviewers' comments on manuscript

Reviewer #1 (Remarks to the Author):

In the article "S-glutathionylation of FABP5 protects against acute lung injury by suppressing inflammation in macrophages", the authors provide convincing evidence that glutathionylation of Cys127 of FABP5 is required for its nuclear translocation, fatty acid binding and subsequent activation of PPARbeta in macrophages. Grx1 was identified as responsible for the de-glutathionylation of FABP5 and selective knockout of Grx1 in macrophages resulted in increased glutathionylation and subsequent nuclear localisation of FABP5. The glutathionylation of Cys127 of FABP5 resulted in PPARbeta activation and prevented the LPS induced inflammatory response (as measured by IL-1beta, IL-6 and TNFalpha) in BMDMs.

Overall the article presents novel data on a specific redox modification of FABP5 that regulates an important physiological mechanistic response to stress. The methodology employed throughout the article is technically sound and the results would support the authors hypothesis. A wide range of techniques are employed in the article include using selective knockout models, a redox proteomic approach to identify Grx1 targets, RNA-seq analysis and qPCR, blotting, Co-IP, microscopy, flow cytometry.

We greatly appreciate the positive comments from the reviewer.

I only have some minor comments that the authors may wish to clarify or address.

1. In the initial experiment the authors subject mice to normoxia or hyperoxia to investigate the role of hyperoxic acute lung injury. It is interesting that some of the general results obtained in this hyperoxic experiment are very similar to those mentioned in the introduction with exposure to cigarette smoke (25) and COPD (27) i.e. reduction in Grx1 expression and decreased GSH:GSSG ratio, could the authors comment in the discussion on whether hypoxia would provoke similar responses and why?

We thank the reviewer for this insightful suggestion. Grx1 belongs to the oxidoreductase family and actively maintains cellular redox homeostasis using GSH substrates. We show that the Grx1 expression level and GSH/GSSG ratio are significantly decreased after exposure to hyperoxia. Similarly, recent studies have shown that the Grx1 level is decreased according to severity of COPD (Peltoniemi, M. J. et al., 2006, *Respir Res.* 7, 133), and cigarette smoke

extract decreases the Grx1 level in lung epithelial cells and increases protein S-glutathionylation in BALF (Kuipers, I. et al., 2011, *Am J Respir Cell Mol Biol.* 45, 931-937).

Smoking and COPD can cause hypoxia, where the body does not get enough oxygen to meet the needs of all its organs and tissues. Hypoxic conditions favour the increase of reactive oxygen species (ROS) and oxidative stress. With increased levels of ROS, intracellular GSSG will accumulate, and the GSH/GSSG ratio will decrease. On the other hand, when oxidative stress occurs, cells attempt to counteract the oxidant effects and restore the redox balance by activation or silencing of genes encoding defensive enzymes, transcription factors, and structural proteins. Therefore, we speculate that Grx1, as an important component of the endogenous antioxidant defense system, would play a crucial role in hypoxia induced oxidative stress.

The correlation between hypoxia and Grx1/S-glutathionylation has been investigated *in vitro* and *in vivo*. Hypoxia-mediated oxidative stress promotes S-glutathionylation of MCU at cysteine 97 and modulates MCU activity (Dong, Z. et al., 2017, *Mol Cell.* 65, 1014-1028). Hypoxia decreases the glutathione redox potential in the mitochondrial matrix and the cytoplasm (Swain, L. et al., 2016, *Circ Res.* 119, 1004–1016). Glutathione supplementation potentiates hypoxic apoptosis by S-Glutathionylation of p65-NFκB (Qanungo, S. et al., 2007, *J. Biol. Chem.* 282, 18427-18436). Exposure of rat myocardium to hypoxia increases the S-glutathionylation of the α1 subunit of Na,K-ATPase in crude ventricular homogenates (Petrushanko, I. Y. et al., 2012, *J Biol Chem.* 187, 32195–32205) .

We have added some comments about hypoxia in the discussion section (**Page 20, Line 413-416**).

2. The authors use a H₂O₂ concentration of 200uM for 10 mins to identify glutathionylated proteins and for other experiments, why this dose and time?

We appreciate the reviewer for raising this question. H₂O₂ has been applied as an oxidative stress inducer for the *in vitro* studies. However, at high concentrations (millimolar range), H₂O₂ exerts toxic effects leading to rapid cell disruption. Moreover, high concentration or long exposure of H₂O₂ induces irreversible protein oxidation, including cysteine sulfinic acid and sulfonic acid modifications.

In the pre-experiments, we analyzed the S-glutathionylation of FABP5 in different time and concentration (**as indicated below**), and selected the H₂O₂ concentration of **200uM** for **15 mins** to perform the Co-IP assays and other experiments.

3. The redox proteomic approach is well performed but the full list of identified glutathionylated peptides and proteins should be included in a supplementary file.

We thank the reviewer for pointing out this negligence. We have supplemented the full list of identified S-glutathionylated peptides and proteins in the **supplementary Table 1, 2**.

Potentially further substrates of Grx1 are of great value and if I can read Fig3C correctly (resolution needs improving), Cys249/250 of PTBP1 would appear to be the top target of Grx1 in this system.

As pointed out by the reviewer, the resolution of **Fig. 3c** has been improved. S-glutathionylation of PTBP1 at Cys249/250 ranked first.

The resolution of the MS/MS fragmentation spectra in Fig 6D needs to be improved, as this peptide contains both Cys120 and Cys127 of FABP5 to identify which one is glutathionylated. The quantification of the glutathionylation of all Cys residues reported as glutathionylated in FABP5 could be presented in the supplementary information.

We appreciate this helpful suggestion. We have improved the resolution of **Fig. 6d**. As suggested by the reviewer, the quantification of the S-glutathionylation of 4 Cys residues (Cys67, Cys87, Cys120, and Cys127) identified as S-glutathionylated in FABP5 has been **added below**. According to mass shift of modification in Fig. 6d, both Cys120 and Cys127 were S-glutathionylated. Furthermore, **Fig. 6e, f, k, Supplementary Fig. 6c, d, g, h** indicated that only C127S mutation inhibited the fatty acid binding and nuclear accumulation of FABP5 under oxidation conditions.

Protein accession	Position	Amino acid	Protein description	Gene name	Modified sequence	Control	H ₂ O ₂	H ₂ O ₂ /Control Ratio
Q05816	127	C	Fatty acid-binding protein, epidermal	Fabp5	MIVEC(1)VMNNATC(1)TR	0.764	1.165	1.523
Q05816	120	C	Fatty acid-binding protein, epidermal	Fabp5	MIVEC(1)VMNNATC(1)TR	0.764	1.165	1.523
Q05816	67	C	Fatty acid-binding protein, epidermal	Fabp5	TTVFSC(1)NLGEK	1.054	0.963	0.913
Q05816	87	C	Fatty acid-binding protein, epidermal	Fabp5	TETVC(1)TFQDGalVQHQQWDGK	1.096	0.934	0.853

The immunolocalisation study in Fig6E would indicate that only the C127S mutation prevented nuclear accumulation of FABP5 but considering the 2 Cys residues on the peptide detected and its proximity to the GSH binding domain perhaps Fig6F could include the C120S mutation?

As suggested by the reviewer, we examined cytoplasmic and nuclear FABP5 in COS7 cells transfected with FABP5 WT or C120S after exposure to H₂O₂ (200 μM) for 1 h. **As indicated below** (also see in revised **Supplementary Fig. 6d**), mutation of Cys120 to Ser (FABP5 C120S) cannot block the H₂O₂-induced nuclear translocation of FABP5 in COS7 cells, confirming that Cys127 is required for the nuclear accumulation of FABP5 under oxidative stress.

4. The formation of FABP5 dimers and oligomers in Fig4E non-reducing and reducing gels would suggest they are via disulphide bridges dependent on the other Cys residues in FABP5. But in Fig4E and 4I we do not see any FABP5 monomer as glutathionylated, does this mean that the protein needs to be in the dimer/oligomer formation for glutathionylation? As the authors already have the mutant forms of FABP5 with each Cys residue mutated, it would be very quick and might be interesting to determine which ones are required for dimer/oligomer formation and whether this is required for glutathionylation of Cys127.

We appreciate this important issue raised by the reviewer. In **Fig. 4e, i**, Co-IP results showed that the dimers and oligomers of FABP5 were increased in response to H₂O₂ and LPS. However, the monomer of FABP5 was almost undetectable. **In figure below**, the long-exposure images of Co-IP results exhibited a weak signal of FABP5 monomer. Recent studies have shown that S-glutathionylation triggers protein oligomerization by a thiol–disulfide exchange mechanism (Demasi, M. et al., 2008, *Free Radical Bio Med.* 44,

1180-1190; Martina, J.A. et al., 2021, *EMBO J.* 40, e105793). So, we speculate that the S-glutathionylation of FABP5 promotes the formation of dimers and oligomers, and immunoprecipitation with anti-GSH antibody facilitates the effects by enriching S-glutathionylated proteins.

As suggested by the reviewer, we performed Co-IP and WB to determine which cysteines are required for dimer/oligomer formation and whether this is required for glutathionylation of Cys127. After transfection of pXJ40-Flag vector, pXJ40-Flag-FABP5 WT, C43S, C47S, C67S, C87S, C120S or C127S, and exposure to H₂O₂ (200 μM) for 15 min, COS7 cell lysates were subjected to SDS-PAGE under reducing and non-reducing conditions. **As shown below**, C43S, C47S, C87S, C120S and C127S mutations inhibited the formation of dimers and oligomers. Next, after transfected with pXJ40-Flag-FABP5 WT, C43S, C47S, C67S, C87S or C120S, COS7 cells were exposed to H₂O₂ (200 μM) for 15 min. **As shown below** (also see in revised **Fig. 6j**, **Supplementary Fig. 6e, f**), compared with FABP5 WT, mutation of C43S, C47S, Cys67, Cys87 or Cys120 to Ser didn't influence the S-glutathionylation of FABP5 obviously.

These data suggest that Cys43, Cys47, Cys87, Cys120 and Cys127 are all involved in the dimer/oligomer formation of FABP5. However, Cys127 is required for the S-glutathionylation of FABP5. S-glutathionylation facilitates the dimer/oligomer formation of FABP5, but dimer/oligomer formation is not required for the S-glutathionylation of Cys127.

5. Why in Fig4I is there a dimer of FABP5 that is glutathionylated in the DTT treated negative control lane?

We thank the reviewer for pointing out this issue. S-glutathionylation has been reported to trigger protein oligomerization by a thiol–disulfide exchange mechanism (Demasi, M. et al., 2008, *Free Radical Bio Med.* 44, 1180-1190; Martina, J.A. et al., 2021, *EMBO J.* 40, e105793). So, we speculate that the glutathionylation of FABP5 promotes the formation of dimers and oligomers, and immunoprecipitation with anti-GSH antibody facilitates the effects by enriching glutathionylated proteins.

In Co-IP assays for FABP5 S-glutathionylation, we noticed that 10 mM DTT, a commonly used concentration, didn't completely reduce the dimer of FABP5. So, in **Fig. 4i, j and 6j**, 50 mM DTT was used as a negative control. However, there is still a weak signal, which is the dimer of S-glutathionylated FABP5, cannot reduced by 50 mM DTT.

Although DTT treatment reduced the increased signal induced by LPS to the level of the control group, there is a dimer of FABP5 that cannot be completely reduced. DTT is frequently used to reduce the intramolecular and intermolecular disulfide bonds of proteins and peptides. However, DTT cannot reduce buried (solvent-inaccessible) disulfide bonds. Therefore, we speculate that the disulfide bonds in the dimer and oligomer of FABP5 detected by Co-IP assays may be too complex to be completely reduced by DTT.

On the other hand, it has been reported that ligand-bound human FABP5 dimerizes via a domain-swapping mechanism (Sanson, B. et al., 2014, *Acta Cryst.* 70, 290–298). Dimers and oligomers of protein are also formed by non-covalent bonds or non-disulfide bonds. So, we surmise that the dimers detected in control and DTT groups may not due to the formation of disulfide bonds.

6. Finally, could the authors include in the discussion a brief line on the mechanism of how PPARbeta activation results in an anti-inflammatory response.

Thanks for pointing this out and as suggested by the reviewer, we added a brief summary of PPAR β/δ and inflammation in the discussion (**Page 23, Line 480-485**).

Our data indicate that S-glutathionylation enhances the ability of FABP5 to activate PPAR β/δ . The effects of PPAR β/δ in the control of the macrophage inflammatory response have been largely studied. The anti-inflammatory properties of PPAR β/δ are mainly based on inhibiting NF- κ B signaling as well as expression of pro-inflammatory cytokines (iNOS, COX2, TNF α), chemoattractant molecules (MCP-1, MCP-3) and adhesion molecules (VCAM-1, ICAM-1, and E-selectin) in macrophages (Chinetti-Gbaguidi, G., et al., 2017, *Biochimie.* 136, 59-64; Magadum, A., et al., 2018, *Int J Mol Sci.* 19, 2013).

Overall a very interesting and novel article that identified a specific redox mechanism of FABP5 responsible for the potential prevention of acute lung injury.

We are grateful that you recognize the importance of our work and appreciate your comments.

Reviewer #2 (Remarks to the Author):

Review of FABP5-SSG as anti-inflammatory agent in macrophages/ Nature Commun. 2020

Overall: This manuscript addresses the important area of anti-inflammatory regulation in acute lung injury, focusing on a novel regulatory mechanism involving site-specific S-glutathionylation of FABP5 on Cys-127 and the role of Grx1 in reversing this oxidative modification. The overall data support the interpretation that S-glutathionylation of FABP5 on Cys127 enhances fatty acid binding and nuclear translocation by facilitating cytosolic exposure or stabilization of the NLS that is necessary for nuclear import and suppression of inflammatory responses. Many modern techniques have been utilized in this study, including quantitative redox proteomics, mouse models with cell-specific molecular modifications, and molecular docking analysis. Overall this is an impressive study that provides fresh insights. However, there are substantial shortcomings that detract from the quality of the manuscript, including organization of the manuscript, presentation of some of the data, quality/accuracy of certain items of data, interpretation of some of the results, accuracy/clarity of some descriptions in the narrative, and level of resolution of many of the figures. Specific comments are delineated below.

Specific comments:

1. Abstract, line 9: “Here we report that Grx1-mediated S-glutathionylation in macrophages alleviates the inflammation of acute lung injury.” Comment: Although it appears to be a minor problem with choice of words, “Grx1-mediated” should be replaced with “Grx1-regulated,” because “mediated” suggests that Grx1 is catalyzing the formation of protein-SSG. Instead, as the authors understand, the primary role of Grx1 is to catalyze de-glutathionylation of protein-SSG. There are unique circumstances where Grx1 may catalyze S-glutathionylation (e.g., when the GSH/GSSG ratio is extremely low (< 1) or when hydroxyl radicals are abundant); but such considerations do not apply here. There are several places in the manuscript where this ambiguity should be corrected (see below).

Results, lines 154-156, “These data demonstrate that Grx1-mediated [replace with regulated] protein S-glutathionylation in macrophages plays a protective role in acute lung injury.”

We thank the reviewer for pointing out this important issue. We have replaced all the “mediated” with “regulated” (Page 1, Line 12; Page 9, Line 163)

2. Introduction, lines 45-47, “Grx1 is involved in the regulation of various cellular functions, maintains a stable cellular redox state, and produces a

reducing environment within the cell under varying environmental conditions.”
Comment: “produces a reducing environment” is an inaccurate statement because it implies that the enzyme changes the position of equilibrium. Instead Grx1 facilitates homeostasis by catalyzing primarily de-glutathionylation. The sentence could read, “... serves to maintain a reducing environment...”

We appreciate the insightful suggestions. We have corrected it (**Page 3, Line 45**).

3. Introduction, lines 85-86, “Here, we report that macrophage-restricted ablation of Grx1 confers protection against acute lung injury in mice. Comment: This finding and others such as LPS-induced diminution of Grx1 in macrophages should be included in the Abstract, because they help convey the regulatory role of Grx1 as de-glutathionylation catalyst, namely absence or diminution in Grx1 content generally leads to increased glutathionylation.

Thank you for pointing this out. We have added these findings in the Abstract (**Page 1, Line 8-12**).

4. Results, section one, focus on hypoxia – The initial part of the Results section is not well integrated with the major portion of the manuscript, and it stands out as being incomplete, having only implicated Grx1 and S-glutathionylation in general, without the mechanistic depth of the study of LPS-induced activation of macrophages presented in the rest of the manuscript. The one commonality is the observation that loss of Grx1 substantially diminished the lung damage caused by hyperoxia. The loss of Grx1 and consequent enhancement of protein glutathionylation as a protective mechanism is the common thread that should be developed more effectively in the first section as a segue to the main body of the manuscript, which nicely evolves the specific role of S-glutathionylation of FABP5-Cys127.

We fully understand the reviewer’s concern on this important issue. Recent studies have indicated that Grx1 regulates protein S-glutathionylation and LPS-induced acute lung injury (Aesif, S.W., et al., 2011, *Am J Respir Cell Mol Biol.* 44, 491-499.). To confirm the role of oxidative stress and redox regulation in acute lung injury, we analyzed the *Grx* expression, GSH/GSSG ratio, protein S-glutathionylation and lung damage after hyperoxia exposure by using Grx1 KO mice. Hyperoxia-induced acute lung injury and LPS-induced acute lung injury are both commonly-used mouse model of acute lung injury. LPS and hyperoxia exposure leads to the generation of ROS such as H₂O₂, inflammation and cell damage, which can cause acute lung injury. Thus, in the initial part of the results, hyperoxia-induced mouse model of acute lung injury was used to verify the association of Grx1-regulated S-glutathionylation and acute lung injury. Our data revealed that loss of Grx1 enhanced the overall

protein S-glutathionylation in lung tissue, protecting against lung damage caused by hyperoxia. We consider that the reduced *Grx* expression and Grx1 activity in the mouse lungs is one of the mechanisms of protection against oxidative stress in the pathogenesis of acute lung injury. As suggested by the reviewer, we have added a summary in the first section of result (**Page 7, Line 131-132**).

5. Results: problematic GSH/GSSG ratios (Fig. 1A, Fig. 2B). The authors state (lines 101-102) that the GSH/GSSG ratio “provides a reliable estimate of cellular redox status.” This can be true if the ratio is measured accurately. Typically the GSH/GSSG ratio is expected to be near 100 or greater for nonoxidatively stressed cells; however, the value for the normal oxygen control (Fig. 1A, NOX) is less than 10! Such a value is usually interpreted as severe oxidative stress. Likewise, the value shown for the unstimulated macrophages is 16 (Fig. 2B), whereas the usual value is >100. For example, the study cited below reports a resting value of 200.

K. Alam, et al., Glutathione-Redox Balance Regulates c-rel–Driven IL-12 Production in Macrophages: Possible Implications in Antituberculosis Immunotherapy, *J. Immunol.* 2010, 184 (6) 2918-2929; DOI: <https://doi.org/10.4049/jimmunol.0900439>.

What precautions were taken to prevent oxidation of GSH to GSSG during processing of samples? Did the authors run a standard curve of various GSH/GSSG ratios to confirm that a range of values (e.g., 1-100) could be measured accurately with the luminescence-based assay that they used (Promega)?

We appreciate this concern suggested by the reviewer. To evaluate the cellular redox status, GSH and GSSG levels were quantified with a luminescence-based GSH/GSSG-Glo Assay per the manufacturer's instructions (Promega, V6611). The oxidized glutathione lysis reagent releases reduced and oxidized glutathione and contains N-Ethylmaleimide (NEM), which reacts with GSH to produce a form that cannot contribute to luminescent signal. Besides, we included a vehicle (or untreated) control and a background control (no cells) on each plate. The vehicle control enables observation of solvent effects on GSH/GSSG. The no-cells control gives a measure of background signal from the assay chemistry, which can be subtracted from vehicle and test signals to give net values.

We carefully checked all the steps of this experiment and found that it may be a problem with the standard curve. In the pre-experiments, a standard curve was generated by serial twofold dilutions of 320 μ M glutathione (**Figure A, as indicated below**). However, as recommended in the instruction book, we didn't perform a standard curve with every experiment. We repeated this assay and run a standard curve at the same time. The value of our samples

measured by the luminometer was consistent with the previous results, but the standard curve obtained was quite different from the previous one (**Figure B, as indicated below**). It is possible that we made a mistake while diluting the dilution of the GSH standard in the pre-experiment. Using the revised standard curve, we recalculated the GSH/GSSG ratio (**Figure C, as indicated below**), the value for macrophages and other cells is near 100. However, the GSH/GSSG ratio in BALF was relative low. The BALF from each mouse was collected by lavaging right lungs with 0.5 ml of Ca²⁺- and Mg²⁺-free PBS, and this procedure was repeated three times. It is possible that a small part of GSH was oxidized to GSSG during processing of samples. We have put these results in the revised **Fig. 1a, 2b, Supplementary Fig. 3a-d**.

A

B

C

6. Results: problematic representation of Grx1 activity (Fig. 1E). There is a good quantitative correlation of the diminution of Grx1 activity (Fig. 1E) with the diminution of its expression (Fig. 1C) in terms of fold change, but expression is shown as relative (1.0 = 100%), whereas activity is presented as specific activity (units/mg). The value of ~100 units/mg shown in Fig. 1E is not possible for a crude tissue sample or cell homogenate, because the maximum specific activity of pure Grx1 is ~100 units per mg. Should the y-axis of Fig. 1E read as “% Activity?”

Thank you for pointing out this mistake. The title of y-axis (1 Unit=1 **μmol** NADPH/min/mg) was mislabeled. The value of Grx1 activity shown in **previous Fig. 1E** was for the oxidation of 1 **nmol** NADPH/min/mg protein. We have recalculated the Grx1 activity by converting **nmol** to **μmol** (as indicated **below** and also see in **revised Supplementary Fig. 1c**).

It is remarkable that the authors recognize the specificity of Grx1 for selectively reducing glutathione containing mixed disulfides and they rely on this specificity for their proteomic mass spec procedure in which Grx1 is used to remove GSH from protein-SSG. However, under Methods for Grx1 activity they have referenced two articles; one (28) does not describe an assay for Grx activity and the other (73) describes an old assay of thioltransferase activity that utilizes non-specific, non-glutathione containing substrates.

Thank you for pointing this out. We have corrected these mistakes (**Page 27, Line 547**).

The appropriate assay for Grx1 utilizes cysteine-SSG as substrate, which serves as a mimic for all protein-SSGs, without the potential steric hindrance due to surrounding protein residues (see e.g., the reference cited below which characterizes Grx2 relative to Grx1, using Cys-SSG as substrate).

Gallogly MM, Starke DW, Leonberg AK, Ospina SM, Mieyal JJ., Kinetic and mechanistic characterization and versatile catalytic properties of mammalian glutaredoxin 2: implications for intracellular roles. *Biochemistry*. 2008 Oct 21; **47**(42):11144-57. doi: 10.1021/bi800966v

We thank the reviewer for bringing up this critical concern. As described in our Materials and Methods, 2-hydroxyethyl disulfide (HED) was used as substrate to measure the activity of Grx1. According to the method suggested by the reviewer, we utilized cysteine-SSG (L-CySSG) to perform the Grx1 activity assay. We have added this detailed protocol in our revised manuscript (**Page 27, Line 549-552**).

As indicated below, similar results were obtained by using cysteine-SSG in lung tissue after hyperoxia exposure. We have put these results in our **revised Fig. 1d, Supplementary Fig.1d**, replacing previous data of Grx1 activity (now in **revised Supplementary Fig. 1c, e**).

7. Results, Fig. 2/ Why were several different approaches used to assess changes in protein glutathionylation (Western blot and DTNB recycling versus immunocytochemistry)?

We thank the reviewer for pointing this out. In **Fig. 1a-h**, our data show that Grx1 is involved in the pathogenesis of hypoxia-induced lung injury, and Grx1 deficiency significantly ameliorated acute lung injury. As the reviewer mentioned above, Grx1 catalyzes de-glutathionylation of protein-SSG. However, there are unique circumstances where Grx1 may catalyze S-glutathionylation. In addition, oxidative stress promotes the formation of a diverse range of cysteine oxidative PTMs, including s-nitrosylation (-SNO), S-sulfinylation (-SO₂H) and S-sulfonylation (-SO₃H). Therefore, it is important to assess the level of protein S-glutathionylation after exposure to hypoxia and confirm that the increased S-glutathionylation is regulated *by Grx1*.

Furthermore, each method has its own advantages and disadvantages. Non-reducing western blot with anti-GSH has low sensitivity; the samples of DTNB assay should be fresh and DTNB reagent is easily reduced when exposed to light and leads to increased color intensity; the results of immunocytochemistry staining is difficult to be quantified.

8. Results, lines 140-142, “Based on the above results, we hypothesize that Grx1 in macrophages plays a key role in the pathogenesis of acute lung injury.” Comment: This statement is not readily understood without reading the rest of the manuscript, where the role of Grx1 in removing GSH from a specific protein-SSG interferes with the protective effect of that protein-SSG. Without this understanding it appears that you are hypothesizing that Grx1 action leads directly to pathogenic events. Some more elaboration is necessary here.

We fully understand the reviewer’s concern and appreciate the insightful suggestions. We have replaced the “Grx1” with “Grx1 regulated S-glutathionylation” (**Page 8, Line 148**).

9. Results, Fig. 3C: What does treatment/control ratio mean in Fig. 3C-what is being measured? The legend does not provide an explanation.

Thank you for pointing this out. We have changed the y-axis label to “fold changes of protein S-glutathionylation (H₂O₂ treated/control)”, and we have added detailed information in the figure legend of **Fig. 3c**.

10. Results, lines 191-192: Why was the artificial COS7 construct substituted for macrophages to study the glutathionylation of FABP5? How does this COS7 model compare to the natural abundance of Grx1 and FABP5 in macrophages? It is important to be more transparent here. Was the COS7 construct used because it was not possible to document FABP5-SSG directly with macrophages? Rather, glutathionylation of FABP5 in macrophages is only inferred by observation of oligomerization of the protein, presumed to occur via inter-protein disulfide bonds facilitated by the transitory formation of FABP5-SSG as a precursor?

We fully understand the concerns proposed by the reviewer. As shown in **Fig. 4e**, the S-glutathionylation of FABP5 was evaluated by Co-IP **in COS7 cells**. Consistently, Grx1 deficiency **in BMDMs** enhanced the S-glutathionylation of FABP5 (**Fig. 4f**). In **Fig. 4h**, western blot results showed that the dimers and oligomers of FABP5 were increased upon H₂O₂ exposure **in COS7 cells**. Similar effects were revealed **in BMDMs** (**Fig. 4g**). In **Fig. 5g**, overexpression of Grx1 **in COS7 cells** substantially inhibited the nuclear accumulation of FABP5 induced by H₂O₂ exposure. Similar results were obtained from control and Grx1 KO **BMDMs** (**Fig. 5h**). In **Fig. 6e** and **Supplementary Fig. 6c**,

immunofluorescence staining and confocal microscopy imaging was performed to detect the nuclear translocation of FABP5 WT and mutations **in COS7 cells**. Likewise, FABP5 nuclear translocation has been detected **in BMDMs** by immunofluorescence staining (**Fig. 5d**). In **Fig. 7j**, overexpression of FABP5 C127S markedly suppressed the H₂O₂-induced expression of the PPRE-driven reporter **in COS7 cells**. **In RAW264.7 cells**, we also measured the ability of FABP5 to induce reporter expression and a similar results was obtained (**Figure A, as indicated below**, also see in revised **Fig.7I**). In **Fig. 7h (Figure B, as indicated below)**, purified recombinant mouse FABP5 protein interacts with PPAR β/δ from **COS7 cell** lysates. Consistently, in **Fig. 7k (Figure C, as indicated below)**, CHIP assays reveal that S-glutathionylated FABP5 directly interacts with PPRE **in Raw264.7 cells**.

A

B

C

Macrophages are known to be difficult to transfect. The low transfection efficiency makes it difficult to analyze the function of FABP5 S-glutathionylation. Moreover, macrophages are relatively small, which cannot show the optical images of intracellular structures when we examine the subcellular location of FABP5 under oxidative stress. In the pre-experiments, we have measure the level of FABP5 S-glutathionylation in 297T, Hela and COS7 cells, which are commonly used tool to study gene function. **As indicated below**, in COS7 cells, a significant enhancement of FABP5 glutathionylation was detected after H₂O₂ treatment. Besides, in recent studies, COS7 cells has been used to examine the distribution of FABP5 in live cells treated with various ligands (Schug, T. T., et al., 2007, *Cell*, 129, 723-733). Therefore, in this work, COS7 cells were transfected with FABP5 or Grx1 plasmids to perform Co-IP and immunocytochemistry staining.

We performed Co-IP and WB to determine which cysteins are required for dimer/oligomer formation and whether this is required for S-glutathionylation of FABP5. After transfection of pXJ40-Flag vector, pXJ40-Flag-FABP5 WT, C43S, C47S, C67S, C87S, C120S or C127S, and exposure to H₂O₂ (200 μM) for 15 min, COS7 cell lysates were subjected to SDS-PAGE under reducing and non-reducing conditions. **As shown below**, C43S, C47S, C87S, C120S and C127S mutations inhibited the formation of dimers and oligomers. Next, after transfected with pXJ40-Flag-FABP5 WT, C43S, C47S, C67S, C87S or C120S, COS7 cells were exposed to H₂O₂ (200 μM) for 15 min. **As shown below** (also see in revised **Fig. 6j** and **Supplementary Fig. 6e, f**), compared with FABP5 WT, mutation of Cys43, Cys47, Cys67, Cys87 or Cys120 to Ser didn't

influence the S-glutathionylation of FABP5 obviously. We have added the results in the revised manuscript (**Page 15, Line 309-312**).

These data suggest that Cys43, Cys47, Cys87, Cys120 and Cys127 are all involved in the dimer/oligomer formation of FABP5. However, Cys127 is required for the S-glutathionylation of FABP5. S-glutathionylation facilitates the dimer/oligomer formation of FABP5, but dimer/oligomer formation is not required for the S-glutathionylation of Cys127.

*11. Results, Fig.6E: It is difficult to discern any differences among the panels. More explanation is necessary, along with a quantitative analysis of the perceived differences. Fig. 6F provides more definitive data, but the approach is different (Western blot) and the experiment is conducted with the artificial COS7 construct.*

We understand the concerns and appreciate the suggestions from reviewer. We have improved the resolution of **Fig. 6e** and quantified the nuclear accumulation of FABP5 as shown below (**Supplementary Fig. 6c**). Overexpression of FABP5 WT, C67S, C87S, C120S and C127S did not change the nuclear translocation of FABP5. $H_2O_2$ treatment clearly increased nuclear accumulation of FABP5 (WT, C67S, C87S and C120S). Mutation of Cys127 to Ser (FABP5 C127S) blocked the $H_2O_2$-induced nuclear translocation of FABP5 in COS7 cells. We have supplemented explanation in the manuscript (**Page 14-15, Line 292-297**).

Furthermore, we analyzed the nuclear translocation of FABP5 C120S after treatment with $H_2O_2$ by western blot (**as indicated below**, also see in revised **Supplementary Fig. 6d**). Consistently, mutation of Cys120 to Ser did not inhibit the $H_2O_2$-induced nuclear accumulation of FABP5, conforming the important role of Cys127 in the regulation of $H_2O_2$-induced FABP5 nuclear accumulation.

In macrophages, we have shown that Grx1-regulated S-glutathionylation Grx1 deficiency clearly enhanced the nuclear translocation of FABP5 (**Fig 5d, e, h**). The low transfection efficiency of macrophage makes it difficult to analyze the nuclear translocation of FABP5 C127S under oxidative stress. In **Fig. 7a-e**, FABP5 KO BMDMs and Grx1 KO BMDMs were used to analysis the changes in biological processes and pathways caused by C127S after nucleofection with FABP5 WT and C127S. Consistent with our results obtained from COS7 cells, regulation of the inflammatory response and cytokine production were enriched in macrophages overexpressing FABP5 C127S relative to FABP5 WT, demonstrating the central role of FABP5 cysteine 127 in mediating macrophage inflammation through S-glutathionylation.

12. General note about the figures: The resolution is relatively poor and the very small labels are often difficult to read even at higher magnification.

We appreciate this important issue raised by the reviewer. We have improved the resolution of all figures.

13. Results, lines 284-286 (Fig. 6l), “In response to H₂O₂, FABP5 WT was strongly glutathionylated, whereas glutathionylation was significantly decreased with FABP5 C127S (Fig. 6l, S6C).” Comment: It would be very helpful to quantify the results of Fig. 6l. Was the extent of glutathionylation diminished by more than 25%, indicating a greater propensity for glutathionylation of C127 relative to the other 3 sites?

We appreciate the reviewer’s suggestion. **As shown below**, mutation of Cys127 to Ser almost completely blocks the H₂O₂-induced S-glutathionylation of FABP5. We have revised the description of results (**Page 15, Line 309-312**) and put the quantification of previous Fig. 6l (now in **revised Fig. 6j**) in **Supplementary Fig. 6g**.

Note: Fig. S6 does not have a panel C.

Thank you for pointing this out. We have corrected it.

14. *Results, lines 338-340*, “As shown in Fig. 7J, G114R enhanced the S-glutathionylation of FABP5, suggesting a potential role of FABP5 glutathionylation in disease.” *Comment*: This statement appears to contradict the key observations/conclusions of this study, so it needs to be re-phrased or further explained. If S-glutathionylation of FABP5 leads to diminution of the inflammatory response and G114R-mutation of FABP5 leads to greater glutathionylation of FABP5, then the mutation should be more anti-inflammatory (i.e., protective), inhibiting rather than promoting disease.

We thank the reviewer for pointing out this issue. We show that S-glutathionylation of FABP5 occurs in response to oxidative stimuli, promoting its fatty acid binding ability and nuclear translocation. In macrophages, we demonstrate that S-glutathionylation of FABP5 activates PPAR β/δ and suppresses the LPS-induced inflammation. FABP5 G114R, a mutation identified from schizophrenia and autism spectrum disorder, enhanced the S-glutathionylation of FABP5. FABP5 is expressed in brain, and altered mRNA expression level of FABP5 was detected in psychiatric illnesses. A wide body of evidence has been accumulating for aberrant reactive oxygen species and inflammation in schizophrenia and autism spectrum disorder (Sawa, A. et al., 2016, *Schizophr Res.* 176, 1-2; Pangrazzi, L. et al., 2020, *Int J Mol Sci.* 21, 3293). However, the precise mechanism remains largely unclear.

Data to date indicate that FABP5 is multi-functional protein, plays a key role in lipid-related metabolic processes, cell signaling, cell growth, and differentiation, as well as regulating inflammation. As an intracellular lipid chaperone that binds to fatty acids, FABP5 plays an important role in trafficking fatty acids and related lipids to specific cellular compartments. Disturbances of lipid metabolism have been implicated in psychiatric illnesses. Recently, fatty acids, particularly ω 3 and ω 6 polyunsaturated fatty acids (PUFAs) have attracted attention in the pathophysiology of schizophrenia and ASD (Peet M., et al., 1995, *J. Psychiatr. Res.* 29, 227–232; Arvindakshan M., 2003, *Schizophr. Res.* 62, 195–204.). As PUFAs are extremely lipophilic molecules, FABPs are needed for their intracellular trafficking.

Thus, we speculate that, in psychiatric illnesses, G114R enhances the S-glutathionylation of FABP5, affecting the lipid metabolism in the brain, not just the anti-inflammatory ability of macrophages (or microglia). The detailed molecular mechanisms need to be further studied.

15. *Discussion, lines 370-376*: This paragraph does a better job of tying hypoxia to the rest of the manuscript. But it reveals a deficiency in mechanistic detail, only implicating that S-glutathionylation and Grx1 are involved, but not pursuing how they are involved and what are the key molecular targets. This

shortcoming contrasts with the depth of the second focus of the manuscript on LPS-activation of macrophages.

We appreciate the reviewer for pointing this out. As explained in Comments 4, the role of Grx1 and protein S-glutathionylation in LPS-induced acute lung injury has been reported recently. Different animal models of experimental lung injury have been used to investigate mechanisms of acute lung injury. However, none of these models fully reproduces the features of human lung injury. Thus, in the initial part of the results, hyperoxia-induced mouse model of acute lung injury was used to verify the association of Grx1-regulated S-glutathionylation and acute lung injury. Although the stimuli that induce acute lung injury are different, they have common features, such as inflammation, ROS and epithelial injury. Our data demonstrate that loss of Grx1 enhances the overall protein S-glutathionylation in mouse lungs, protecting against acute lung injury.

After confirming the role of Grx1-regulated S-glutathionylation in ALI, we next explored which type of cell is more sensitive to oxidative stress during the pathogenesis of acute lung injury. We have added some comments in the discussion section (**Page 20, Line 402-413**).

*16. Discussion, lines 379-380, “Not surprisingly, loss of Grx1 in macrophages relieved LPS-induced acute lung injury.” Comment: This may not be surprising to the authors or to a rather narrow audience. However, general knowledge conveys that loss of Grx1 exacerbates oxidative stress and allows increased protein-SSG formation, which more often than not decreases the function of proteins or accelerates their degradation. *The difference in acute lung injury is that particular S-glutathionylation events are protective rather than deleterious. This distinction needs more emphasis in many places in the manuscript.*

We appreciate the reviewer’s suggestion. We have corrected it (**Page 20, Line 420**) and emphasized it in many places in the manuscript.

17. Discussion, lines 416-418, “As to FABP5 ... its S-glutathionylation has not been reported previously.” Comment: It is important in this context to acknowledge that S-sulfenylation of FABP5 on Cys127 has been reported. This is significant, because it suggests that formation of FABP5-Cys127-SOH by reaction of FABP5 with H2O2 likely serves as a precursor to FABP5-Cys127-SSG (see citation below).

Yang J, Gupta V, Carroll KS, Liebler DC. Site-specific mapping and quantification of protein S-sulphenylation in cells. *Nat Commun.* 2014 Sep 1, 5 :4776. doi: 10.1038/ncomms5776. PMID: 25175731; PMCID: PMC4167403.

Indeed, the reaction (Pr-SOH + GSH → Pr-SSG + H₂O) has been identified as a likely common mechanism of S-glutathionylation of proteins. (As described in reference 13, cited in this manuscript).

We appreciate for pointing out our negligence. S-sulfenylation is one potential intermediate prior to forming S-glutathionylation. Yang J et al. have reported that in RKO colon adenocarcinoma cells, FABP5 C127 was more highly S-sulfenylated after H₂O₂ treatment, whereas S-sulfenylation on C120 remained almost unchanged, which is consistent with our results of FABP5 glutathionylation in macrophages. We have corrected and added further description in the revised manuscript (**Page 22, Line 451-453**).

18. Discussion, lines 448-451, “These data suggest that the glutathionylation of FABP5 acts as an antiinflammatory mediator in macrophages during ALI and we propose a new mechanism by which macrophages prevent self-hyperactivation under oxidative stress.” Comment: The regulatory role of Grx1 in this context implicates Grx1 as a target of anti-inflammatory drug development, as suggested previously (see citation, below).

Gorelenkova Miller O, Cole KS, Emerson CC, Allimuthu D, Golczak M, Stewart PL, Weerapana E, Adams DJ, Mieyal JJ. Novel chloroacetamido compound CWR-J02 is an anti-inflammatory glutaredoxin-1 inhibitor. *PLoS One*. 2017 Nov 20; 12(11):e0187991. doi: 10.1371/journal.pone.0187991. PMID: 29155853; PMCID: PMC5695812.

We thank the reviewer for this insightful suggestion. Our data suggest that Grx1 inhibitor enhances the S-glutathionylation of FABP5, suppresses inflammation in macrophages and protects against acute lung injury. CWR-J02, a chloroacetamido compound identified from a rapid screening approach, inhibits the activity of Grx1 *via* covalent modification, serving as a starting point for developing more effective Grx1 inhibitors, and/or anti-inflammatory agents for *in vivo* and therapeutic applications. We have added the discussion in the revised manuscript (**Page 24, Line 490-491**).

19. Methods, lines 585-592 - Fatty Acid Binding Assay: This appears to be only a qualitative assay. Was FABP5-C127S tested in this assay? What is the concentration dependence of the binding? KD value?

We thank the reviewer for this suggestion. We have tested the fatty acid binding ability of FABP5 C127S. Purified recombinant FABP5 C127S protein almost completely abolished the fatty-acid binding of FABP5 under oxidative stress (**Fig. 6k**).

To measure the K_d value of FABP5 and fatty acid under oxidative conditions, MicroScale Thermophoresis (MST) analysis was performed by using the Monolith NT.115 instrument (Nano Temper, Munich, Germany). The affinity (K_d) of FABP5 and linoleic acid was calculated at **340.02±64.903 nM**, and the K_d value of FABP5 C127S and linoleic acid was **412.91±73.293 nM** (as indicated below).

Next, purified mouse recombinant FABP5 protein was incubated with GSSG for 15 min to promote S-glutathionylation. However, protein aggregation was detected when we tried to measure the affinity of S-glutathionylated FABP5 and linoleic acid by MST analysis (as indicated below). We speculate that the S-glutathionylation of FABP5 promotes its dimer/ oligomer formation.

MST traces of FABP5-SSG binding to linoleic acid

MST traces of FABP5 binding to linoleic acid

FABP5 protein aggregation make it difficult for us to measure the K_d value of S-glutathionylated FABP5 and linoleic acid by MST analysis. Then, we detected the concentration-dependence of the binding by Western Blot (**As indicated below, also see in revised Supplementary Fig. 6h**).

Minor Comments:

Results, line 104, “Compared with the NOX group, a **markedly** higher expression level of GSH synthetase ... (Fig. 1B). A ~1.5-fold increase would usually not be described as “markedly higher.” It is a relatively modest increase.

Thank you for pointing this out. We have corrected it (**Page 6, Line 107**).

Results (Fig. 1F, 1G) - It is remarkable that there is no acute lung damage associated with KO of Grx1 under NOX conditions. Are there long-term effects?

We appreciated the reviewer for this insightful point. Grx1 deficiency blocks the de-glutathionylation of protein-SSG. However, compared to HOX group, ROS level (oxidative stress) in the lung is low in NOX conditions. Thus, under NOX conditions, the overall protein S-glutathionylation in the lung tissue of Grx1 KO mice is similar to the control group (**Fig.1 i-l**).

Our data show enhanced S-glutathionylation alleviates acute lung injury and we did not detect obvious lung damage in Grx1 KO mice under NOX conditions. We will further explore the long-term effects.

Results, lines 149-151 (Fig. 2G) – It would be useful here to describe the Grx1fl/fl LysMcre mouse model and its utility.

We appreciate the reviewer's suggestion. When we submitted the manuscript, we choose the **double-blind peer review**. So, the reviewers cannot see this part of information for the moment. We have carefully checked all relevant information to ensure that it has been included in the submitted manuscript.

Discussion, lines 359-362, "Further experiments showed that Cys-127-mediated S-glutathionylation of FABP5 promoted its fatty acid binding and nuclear translocation, activating PPAR β/δ and inhibiting macrophage inflammation." This is phrased awkwardly. It should be restated as, "... S-glutathionylation on the Cys-127 residue of FABP5 promoted"

We appreciate this helpful suggestion. We have corrected it (**Page 19, Line 392-393**).

Discussion, lines 399-401, "The S-glutathionylation of GAPDH promotes the formation of a disulfide bond between Cys150 and Cys154" Note: The cited reference 7 features the roles of Cys152 and Cys156.

Thank you for pointing this out. The S-glutathionylation of GAPDH promotes the formation of a disulfide bond between Cys152 (**Cys150 in mouse** and Cys152 in Human) and Cys156 (**Cys154 in mouse** and Cys156 in Human). However, as suggested by reviewer 4, we have deleted the description of GAPDH-SSG.

Reviewer #3 (Remarks to the Author):

Author:

The paper presents a role for oxidative stress-induced protein S-glutathionylation in regulation of acute lung injury (ALI). The authors report that glutaredoxin-1 (Grx1)-regulated S-glutathionylation in macrophages mediates inflammation associated with ALI. Grx1 KO mice are protected from inflammation in mouse models of acute lung injury. Using a proteomics approach, the authors show that the fatty acid-binding protein 5 (FABP5) is susceptible to S-glutathionylation under oxidative conditions; and that S-glutathionylation of Cys127 in FABP5 activates its fatty acid binding ability and nuclear translocation. RNA-seq data confirmed that S-glutathionylation of FABP5 inhibited the LPS-induced inflammation in macrophages. FABP5 S-glutathionylation activates PPAR β/δ in macrophages. The authors concluded that FABP5 S-glutathionylation is a novel molecular mechanism by which macrophages control inflammation in ALI.

We greatly appreciate the comments from the reviewer.

Major concerns.

After establishing the phenotype of Grx1 KO in the hyperoxia model, it is not clear why the authors switch to an LPS model for assessment of the phenotype of macrophage conditional Grx1 deficient mice.

We understand the reviewer's concern on this important issue. Recent studies have indicated that Grx1 regulates protein S-glutathionylation and LPS-induced acute lung injury (Aesif, S.W., et al., 2011, Am J Respir Cell Mol Biol. 44, 491-499.). To confirm the role of Grx1 and oxidative stress in acute lung injury, we used hyperoxia-induced acute lung injury model and evaluated Grx1 expression, Grx1 activity, GSH/GSSG ratio, protein S-glutathionylation and lung damage of Grx1 KO mice. LPS administration or prolonged hyperoxia exposure leads to the generation of ROS such as H₂O₂ (Fig. 2a, f), inflammation and cell damage, which can cause acute lung injury. LPS-induced acute lung injury and hyperoxia-induced acute lung injury are both widely used mouse model of acute lung injury. Thus, in the initial part of the results, hyperoxia-induced mouse model of acute lung injury was used to verify the association of Grx1-regulated S-glutathionylation and acute lung injury. Our data reveal that loss of Grx1 enhanced the overall protein S-glutathionylation in lung tissue, protecting against acute lung injury.

In the initial part of the results, hyperoxia model was used to verify the role of Grx1-regulated S-glutathionylation in acute lung injury. Next, to further explore the function of Grx1 in macrophages in the regulation of acute lung injury, conditional Grx1 deficient mice were used in LPS-induced acute lung injury. Here, we use the LPS model, mainly because it is more stable and simpler

than the hyperoxia model. As suggested by the reviewer, we have added a comment in the first section of result (**Page 20, Line 402-413**)

Given that hyperoxia exposure and LPS, as used for the in vivo modeling in this paper, are also commonly used as in vitro model reagents, it is not clear why the authors implemented hydrogen peroxide for the biochemical characterization in the in vitro model.

LPS administration or prolonged hyperoxia exposure leads to the generation of ROS (such as H₂O₂, **Fig. 2a, f**) and oxidative stress in the lungs, which can cause acute lung injury. In this work, H₂O₂ was applied as an oxidative stress inducer for in vitro studies. Exposure to H₂O₂ is a commonly used procedure to trigger cellular oxidative stress (Céline Ransy., et al., 2020, Int J Mol Sci. Dec; 21(23): 9149.). Compared with LPS and hyperoxia, H₂O₂ is a more convenient and efficient oxidative stress inducer. After H₂O₂ treatment, we performed LC-MS/MS (**Fig. 3a-c**), Co-IP (**Fig. 4e, f**), fatty acid binding assay (**Fig. 5a-c**) and western blot analysis of nuclear and cytoplasmic fractions (**Fig. 5f-h**) to investigate the level and the function of glutathionylated FABP5 under oxidative stress.

On the other hand, LPS, a characteristic component of the outer membrane of Gram-negative bacteria, was used to induce inflammatory responses, when we analyze the level of inflammatory cytokines to further explore the functional significance of FABP5 S-glutathionylation in macrophages. In the pre-experiments of this work, we have tried to use hyperoxia or H₂O₂ to induce the inflammatory response of macrophages. However, compared with LPS (**Fig. 5i-k**), Hyperoxia and H₂O₂ slightly induced the expression of inflammatory cytokines (**as indicated below**), and the difference between FABP5 WT and FABP5 C127S cannot be easily detected. For H₂O₂ and LPS, each has a particular emphasis, H₂O₂ has a major ability to cause oxidative stress. LPS has a strong ability to induce inflammation responses accompanied by oxidative stress.

mRNA expression of *Il1b*, *Il6* and *Tnfa* in Raw264.7 cells exposed to H₂O₂ (200 μM) for 4 h.

mRNA expression of *Il1b*, *Il6* and *Tnfa* in BMDMs exposed to H₂O₂ (200 μM) for 4 h.

mRNA expression of *Il1b*, *Il6* and *Tnfa* in Raw264.7 cells exposed to hyperoxia for 2 h.

After establishing the phenotypes of the KO and conditional macrophage grx1-deleted mice using in vivo models the authors then switch to model in vitro oxidative stress models to establish that FABP5 is a target for glutathionylation in vitro (in macrophages) and present associated anti-inflammatory mechanisms involving activation of PPARs. This is obviously the fusion of two independently derived data sets, and more attention should be made in describing the logical transition between the two approaches.

We thank the reviewer for this insightful suggestion. After establishing the phenotypes of the KO and conditional macrophage Grx1-deleted mice. We reveal that Grx1-regulated protein S-glutathionylation in macrophages plays a protective role in acute lung injury (**Fig. 1e-h, Fig. 2g-i**). S-glutathionylation is an important regulatory posttranslational modification of protein cysteine (Cys) thiols, yet the role of specific cysteine residues as targets of modification is poorly understood. To identify the specific molecular targets and pathways of Grx1 that are susceptible to redox-dependent regulation in the pathogenesis of acute lung injury, we applied a quantitative redox proteomics approach that permits site-specific profiling of S-glutathionylation at a proteome-wide scale in

macrophages. The S-glutathionylated proteins were broadly distributed across major cellular processes, metabolic processes, and single-organism processes. Among the top five metabolism related differential proteins, the expression of FABP5 in macrophages was markedly higher than other proteins (**Fig. 3c**).

Next, As shown in **Fig 4J**, we detected FABP5 S-glutathionylation in lung tissues from LPS-exposed Grx1^{fl/fl} and Grx1^{fl/fl}LysM^{cre} mice. Grx1 deficiency in macrophages obviously enhanced the S-glutathionylation of FABP5 in lung tissue after exposure to LPS.

It is possible that other S-glutathionylated proteins participate in this process. However, our findings strongly suggest that the S-glutathionylation of FABP5 in macrophage plays a central role in regulating acute lung injury. We have added further description in the revised manuscript (**Page 9, Line 168-169**).

Final in vivo characterization is performed to demonstrate the phenotype of FABP5 knockout mice as susceptible to inflammation in ALI. As additional final validation, it would be helpful if the authors could also demonstrate using the in vivo models, either hyperoxia or LPS-induced injury, that FABP5 is glutathionylated in the lung tissue or lung macrophages of exposed animals.

We appreciate these comments and constructive suggestions from the reviewer. **As shown below** (also see in **revised Fig. 4j**), we measured the level of FABP5 S-glutathionylation in lung tissues of LPS-exposed Grx1^{fl/fl} and Grx1^{fl/fl}LysM^{cre} mice. We observed that Grx1 deletion of macrophage obviously enhanced the S-glutathionylation of FABP5 in lung tissue after exposure to LPS, suggesting that the S-glutathionylation of FABP5 participates in the pathogenesis of acute lung injury by regulating macrophages.

Some of the immunoblot panels eg 5 A-C, 5E-G, and 6, F, J have resolution issues or are presented with high background.

Thank you for pointing this out. We have improved the resolution of **Fig. 5a-c**, **5e-g** (also see in **revised Fig. 5g-h**), **6f**, **6j**. As shown below, we have repeated the assay of previous Fig. 5G (also see in **revised Fig. 5h**).

Figure 7 is largely illegible in part due to small size and poor resolution. Please reformat especially Figs 7 A-C for clarity.

We appreciate the reviewer for pointing out this issue. We have reformatted **Fig. 7a-c**.

Similarly, Fig 6 panels should be provided at adequate size and resolution. Fig. 6B data is illegible in current presentation.

Thank you for pointing this out. We have reformatted **Fig. 6** and improved its resolution.

Abstract. Define Grx1 in abstract.

The abstract mainly highlights the biochemical characterizations. Please add a description of the phenotypes of Grx1 KO and of macrophage-specific Grx1 KO, to the abstract.

We appreciate the reviewer's suggestion. We have added a description of phenotypes in the Abstract (**Page 1, Line 8-11**).

Abstract "Here we report that Grx1-mediated S-glutathionylation in macrophages alleviates the inflammation of acute lung injury." This sentence may be misleading. Please clarify that S-glutathionylation is protective and Grx1 is counterregulatory and responsible for deglutathionylation.

We appreciate the reviewer for pointing this out. We have rewritten this sentence. (**Page 1, Line 8-12**)

Line 85 “macrophage-restricted ablation of Grx1”... suggest editing as “conditional macrophage-specific genetic deletion of Grx1”

Thank you for pointing this out. We have corrected it (**Page 4, Line 85**).

Fig 1E. Grx1 is established a pathogenic mediator in hyperoxia, in studies of the Grx1 KO which have reduced inflammation. However, in wt mice Grx1 enzyme activity is actually reduced by hyperoxia by about 50%. Please clarify how this concurs with the hypothesis that Grx1 promotes inflammation in hyperoxia by antagonizing protein glutathionylation.

We appreciate this helpful suggestion. In WT mice, reduced Grx1 expression and enzyme activity in lung tissue were detected after hyperoxia exposure (**Fig. 1c, d**). As demonstrated in our work, Grx1 promotes inflammation in the pathogenesis of hyperoxia-induced lung injury by antagonizing protein S-glutathionylation. We consider that the reduced Grx1 expression and activity is one of the mechanisms of protection against oxidative stress. It increases S-glutathionylation levels of certain proteins, promotes the process of anti-oxidative stress and thus partially decreases inflammation in the lung. However, it cannot reverse the oxidative stress and lung inflammation induced by hyperoxia. Besides, S-glutathionylation is reversible, S-glutathionylated protein can still be reduced by active Grx1 in the lung.

In Grx1 KO mice, loss of Grx1 completely blocks deglutathionylation reactions, significantly increases the S-glutathionylation of proteins (such as FABP5), and thus decreases the hyperoxia-induced lung inflammation.

Results line 107. Please define these genes for the reader.

We appreciate this helpful suggestion, we have defined these genes in the revised manuscript. (**Page 6, Line 110-113**)

The methods section describes the exposures to Grx1^{fl/fl} LysMcre or FABP5^{fl/fl} LysMcre mice, but a section on mouse breeding appears to be missing. Also, there was no description of the source of Grx1 KO mice described in Figure 1. Also provide details of Fabp5 conditional KO mice presented in Fig.7. Please describe all mouse breeding strategies, induction, and source of materials in the methods section. Also provide information on animal care and compliance with institutional guidelines in Methods section as well.

We thank the reviewer for pointing out this issue. When we submitted the manuscript, we choose the **double-blind peer review**. So, the reviewers cannot see this part of information for the moment. We have carefully checked

all relevant information to ensure that it has been included in the submitted manuscript.

Reviewer #4 (Remarks to the Author):

Major Comments:

1. The authors showed that the glutathionylation of FABP5 promotes the formation of dimer and oligomers of FABP5 which were gradually increased in a concentration-dependent manner upon H₂O₂ exposure. Which state predominantly exists in macrophages in response to oxidative stress? Each form is functionally the same? Which form is involved in translocation from the cytosol to the nucleus to activate PPAR β/δ -mediated signaling pathway and to suppress inflammation?

We thank the reviewer for pointing out this important issue. In **Fig. 4g**, dimers and oligomers of FABP5 were gradually increased in a concentration-dependent manner upon H₂O₂ exposure. In low concentrations of H₂O₂ (such as 200 μ M, the concentration was used in our experiments), most FABP5 are still monomer, the percentage of dimer and oligomers are slightly increased. Furthermore, in the results of nuclear translocation assay (**Fig. 5f-h**) and fatty acid binding assay (**Fig. 5a-c**), the signals of FABP5 we detected by WB are monomer. In the pull-down assay of FABP5 and PPAR β/δ , as shown in the input, purified recombinant FABP5 is monomer (**as indicated below**, also see in **revised Fig. 7h**).

On the other hand, in **Fig. 4e and 4i**, Co-IP results showed that the dimers and oligomers of FABP5 were increased in response to H₂O₂ and LPS. However, the monomer of FABP5 was almost undetectable. **In figure below**, the long-exposure images of Co-IP results exhibited a weak signal of FABP5 monomer. Recent studies have shown that S-glutathionylation triggers protein oligomerization by a thiol–disulfide exchange mechanism (Demasi, M. et al., 2008, *Free Radical Bio Med.* 44, 1180-1190; Martina, J.A. et al., 2021, *EMBO J.* 40, e105793). So, we speculate that the S-glutathionylation of FABP5 promotes the formation of dimers and oligomers, and immunoprecipitation with

anti-GSH antibody or overexpression (Fig. 4e, f, i, 6j and 7m) facilitates the effects by enriching glutathionylated proteins.

Then, to determine which cysteins are required for dimer/oligomer formation and whether this is required for glutathionylation of FABP5, we performed Co-IP and WB assays. After transfection of pXJ40-Flag vector, pXJ40-Flag-FABP5 WT, C43S, C47S, C67S, C87S, C120S or C127S, and exposure to H₂O₂ (200 μ M) for 15 min, COS7 cell lysates were subjected to SDS-PAGE under reducing and non-reducing conditions. As shown below, C43S, C47S, C87S, C120S and C127S mutations inhibited the formation of dimers and oligomers. Next, after transfected with pXJ40-Flag-FABP5 WT, C43S, C47S, C67S, C87S or C120S, COS7 cells were exposed to H₂O₂ (200 μ M) for 15 min. **As shown below** (also see in revised **Supplementary Fig. 6e, and f**), compared with FABP5 WT, mutation of C43S, C47S, Cys67, Cys87 or Cys120 to Ser didn't influence the S-glutathionylation of FABP5 obviously. These data suggest that Cys43, Cys47, Cys87, Cys120 and Cys127 are all involved in the dimer/oligomer formation of FABP5. However, Cys127 is required for the S-glutathionylation of FABP5. S-glutathionylation facilitates the dimer/oligomer formation of FABP5, but dimer/oligomer formation is not required for the S-glutathionylation of Cys127.

Monomer and/or oligomer FABP5 directly interact(s) with PPARβ/δ to activate transcription? Is the glutathionylated FABP5 contained in the transcriptional complex consisting of PPARβ/δ, coactivators (PGC-1α/β) and other cofactors? How do the glutathionylated FABP5 monomer and/or oligomers transfer fatty acid (ligand) to PPARβ/δ in the nucleus? The authors should conduct CHIP assay and/or pull-down assay to evaluate whether direct protein-protein interactions between S-glutathionylated FABP5 and PPARβ/δ or other cofactor(s) would be detected in the nucleus.

We appreciate this helpful suggestion. First of all, pull-down assay of FABP5 and PPAR β/δ has been performed by using purified mouse recombinant FABP5 (WT or C127S) protein and cell lysates, **as shown above** (also see in **revised Fig. 7h**), S-glutathionylation promotes the binding of FABP5 and PPAR β/δ , and mutation of Cys127 to Ser almost completely abolishes the binding of FABP5 and PPAR β/δ . As shown in the input, purified recombinant FABP5 is monomer.

Next, we performed chromatin immunoprecipitation (ChIP) assays to detect whether FABP5 directly interacts with PPARE (a consensus PPAR response element). Results revealed that recruitment of FABP5 to PPARE on the promoters of PPAR β/δ target genes, such as ADRP, FIAF, were markedly increased in response to H₂O₂ treatment (**as indicated below**, also see in the **revised Fig. 7k**). Some other coactivators such as PGC-1 α/β may also be involved in this transcriptional complex. However, it takes a long time to purchase the antibodies during the current pandemic. We will explore the mechanisms in the next step of our research.

A recent study has established the presence of a ligand-sensitive tertiary nuclear localization signal (NLS), consisting of Lys-24, Arg-33, and Lys-34, located on the α 1 and α 2 helices of FABP5. FABP5 does not harbor a classical NLS within its primary sequence, certain “activating” fatty acids elicit FABP5's translocation by permitting allosteric communication between the ligand-sensing β 2 loop and a tertiary nuclear localization signal within the α -helical cap of the protein (Armstrong, E.H. et al., 2014, *J Biol. Chem.* 289, 14941-14954). We assume that the S-glutathionylation of FABP5 enhances the fatty acid binding activity, driving cytosolic exposure of the NLS, and increases the nuclear translocation of FABP5.

2. Although the authors showed that FABP5 glutathionylation controls macrophage inflammation by activating PPAR β/δ , molecular mechanisms

underlying anti-inflammation by the glutathionylated FABP5 in macrophages remain unclear. The authors should describe it in more detail.

We appreciate this important issue raised by the reviewer. As demonstrated above, pull-down assay indicates the interaction between FABP5 and PPAR β/δ under oxidative stress (**Fig. 7h**). Transcriptional activation assays suggest that the ability of FABP5 to induce the reporter expression driven by PPRE (a consensus PPAR response element) is enhanced by S-glutathionylation (**Fig.7i, j**). Furthermore, CHIP assays reveal that S-glutathionylated FABP5 directly interacts with PPRE (**Fig. 7k**). In addition, FABP5 inhibitor blocked the GW0742-induced mRNA expression of endogenous PPAR β/δ target genes in macrophages, suggesting the important role of FABP5 in PPAR β/δ activation (**Fig. 7l**). Collectively, these results suggest that S-glutathionylated FABP5 directly interacts with PPAR β/δ , activates PPAR β/δ and induces the expression of endogenous PPAR β/δ target genes in macrophages (**Page 16-17, Line 347-351; 357-361**)

The effects of PPAR β/δ in the control of the macrophage inflammatory response have been largely studied. The anti-inflammatory properties of PPAR β/δ are mainly based on inhibiting NF- κ B signaling as well as expression of pro-inflammatory cytokines (iNOS, COX2, TNF α), chemoattractant molecules (MCP-1, MCP-3) and adhesion molecules (VCAM-1, ICAM-1, and E-selectin) in macrophages (Chinetti-Gbaguidi, G., et al., 2017, *Biochimie*. 136, 59-64; Magadum, A., et al., 2018, *Int J Mol Sci*. 19, 2013). We will further explore the mechanisms in the next step of our research.

3. The overexpression of FABP5 in COS7 cells might not be appropriate to characterize the function of FABP5 in macrophages under oxidative stress, because expression level of FABP5 and status of its oligomer formation in COS7 cells were not comparable with those in macrophages (RAW264.7 and BMDM).

We appreciate this important issue raised by the reviewer. As show in Fig. 4E, the S-glutathionylation of FABP5 was evaluated by Co-IP **in COS7 cells**. Consistently, Grx1 deficiency **in BMDMs** enhanced the S-glutathionylation of FABP5 (Fig. 4F). In Fig. 4H, western blot results showed that the dimers and oligomers of FABP5 were gradually increased in a concentration-dependent manner upon H₂O₂ exposure **in COS7 cells** overexpressing FABP5. Similar effects were revealed **in BMDMs** (Fig. 4G). In Fig. 5F and 5G, overexpression of Grx1 **in COS7 cells** substantially inhibited the nuclear accumulation of FABP5 induced by H₂O₂ exposure. Similar results were obtained from control and Grx1 KO **BMDMs** (Fig. 5H). In Fig. 6E, immunofluorescence staining and confocal microscopy imaging was performed to detect the nuclear translocation of FABP5 WT and mutations **in COS7 cells**. Likewise, FABP5 nuclear translocation has been detected **in BMDMs** by immunofluorescence

staining (Fig. 5D). In Fig. 7J, overexpression of FABP5 C127S markedly suppressed the H₂O₂-induced expression of the PPRE-driven reporter **In COS7 cells**. **In RAW264.7 cells**, we also measured the ability of FABP5 to induce reporter expression and a similar results was obtained (**as indicated below**, also see in revised Fig. 7I). In Fig. 7H, purified recombinant mouse FABP5 protein interacts with PPARβ/δ from **COS7 cell** lysates. Consistently, in Fig. 7L, CHIP assays reveal that S-glutathionylated FABP5 directly interacts with PPRE **in Raw264.7 cells**.

A

B

C

Macrophages are known to be difficult to transfect. The low transfection efficiency makes it difficult to analyze the function of FABP5 S-glutathionylation. Moreover, macrophages are relatively small, which cannot show the optical images of intracellular structures when we examine the subcellular location of FABP5 under oxidative stress. In the pre-experiments, we have measure the level of FABP5 s-glutathionylation in 297T, Hela and COS7 cells, which are commonly used tool to study gene function. **As indicated below**, in COS7 cells, a significant enhancement of FABP5 glutathionylation was detected after H₂O₂ treatment. Besides, in recent studies, COS7 has been used to examine the distribution of FABP5 in live cells treated with various ligands (Schug, T. T., et al., 2007, *Cell*, 129, 723-733). Therefore, in this work, COS7 cells were transfected with FABP5 or Grx1 plasmids to perform Co-IP and immunocytochemistry staining.

4. Cells used in the present study were exposed to H₂O₂ at various concentrations (200 μM to 500 μM) under oxidative stress. The authors should explain what is the significance of the experiments conducted at different concentration of H₂O₂.

We appreciate the reviewer for raising this question. H₂O₂ has been applied as an oxidative stress inducer for the *in vitro* studies. However, at high concentrations (millimolar range), H₂O₂ exerts toxic effects leading to rapid cell disruption. Moreover, high concentration or long exposure of H₂O₂ induces irreversible protein oxidation, including cysteine sulfinic acid and sulfonic acid modifications. In the pre-experiments, we have analyzed the S-glutathionylation of FABP5 in different time and concentration (**as indicated below**), and selected the H₂O₂ concentration of 200uM for 15 mins to perform the Co-IP assays and other experiments.

In previous **Fig. 4C, 4D** and **7K**, 500 μM H_2O_2 was used to treat the cells. As indicated below, we repeated the experiments by using 200 μM H_2O_2 , and similar results were obtained. We have put these results in our revised **Fig. 4c, d** and **7I**, replacing previous data.

Minor Comments:

1. In Fig.1, 4, 6 and 7, molecular weight markers for proteins should be described in the western blotting data.

We appreciate the reviewer's suggestion. We have reformatted the Figures.

2. In Fig. 1F and 2G regarding histological analysis of lung, the images only in a limited area are not convincing. The authors need to show a wide field of view at a lower magnification.

We thank the reviewer for this helpful suggestion. **As shown below**, a wide field of view has been shown.

3. In Fig. 4F, although glutathionylated FABP5 forms dimer and oligomer, the authors used the monomeric structure to build glutathionylated FABP5 models. The FABP5 dimer adopts a domain swapping structure (PDB ID: 4AZM). The glutathionylated FABP5 model should be also constructed by using the dimer structure along with the monomer.

We understand the concerns and appreciate the suggestions from reviewer. We have performed the molecular docking analysis by using the FABP5 dimer structure (PDB ID: 4AZM) (**As shown below**, also see in **revised Fig. 6i**).

4. In Fig. S4 legend, the sentence “mRNA levels of Fabp3 (A), Fabp4 (B), and Fabp7 (C) in RAW264.7 cells” should be changed to “mRNA levels of Fabp3 (B), Fabp4 (C), and Fabp7 (D) in RAW264.7 cells”.

Thank you for pointing this out. We have corrected it. (**Supplementary Figure legend, 4e**)

5. In Fig. 5, although BMS309403 is generally used as a selective FABP4 inhibitor (*Bioorg. Med. Chem. Lett.* 17, 3511, 2007 and *Nature* 447, 959, 2007), the authors used two compounds, BMS309403 and SBF126, as FABP5 inhibitors. Why did the authors use BMS309403 as FABP5 inhibitor? What are differences between the two inhibitors? Reference regarding these inhibitors should be described.

We understand the reviewer’s concern on this important issue. Few specific FABP5 inhibitors have been described. BMS309403 is a competitive inhibitor of FABPs with reported IC₅₀ values of 350 nM for FABP5 (Sulsky, R. et al., 2007, *Bioorg Med Chem Lett.* 17, 3511-3515; Furuhashi, M. et al., 2008, *Nat Rev Drug Discov.* 7, 489-503). Although BMS309403 is identified as binding FABP4 with high affinity, it has been used as an inhibitor of FABP5 (Xia, S.L. et al., 2015, *Oncotarget.* 6, 5889-5902; Sanson, B. et al., 2014, *Acta Crystallogr D Biol Crystallogr.* 70, 290-298). In this work, the expression level of FABP4 in macrophages is much lower than FABP5 (**Fig. 4a**). So, BMS309403 was selected to inhibit FABP5 in our experiments.

As the reviewer mentioned, considering the specificity of BMS309403 to FABP5, A novel α -truxillic acid 1-naphthyl mono-ester, SBF126, which has been proved to exhibit strong binding (K_i 0.9360.08 mM) and better inhibition of FABP5 than BMS309403, was used as FABP5 inhibitor to confirm the results obtained from BMS309403. (Berger, W. T. et al., 2012, *PLoS One.* 7, e50968;

Kaczocha, M. et al., 2014, *PLoS One*. 9, e94200). We have added the reference regarding the two inhibitors. (**Page 12, Line 237; Page 14, Line 272-277**)

Although BMS309409 is used as FABP5 inhibitor in 222 line, BMS309403 is used in line 224 and Fig. 5. The authors should describe which is correct.

Thank you for pointing this out. We have corrected it. (**Page 12, Line 235**)

6. In Fig. 5C, EDTA (18 μmol) should be changed to description of concentration.

We appreciate the reviewer's suggestion. We have corrected it (**Figure legend 5c**).

7. In Fig. 5D, the images of only in a limited area analyzed by the immunofluorescence staining are not convincing, and the differences between the groups are not clear. The authors need to quantify positive (FABP5 nuclear translocated) cells in each group.

We understand the concerns and appreciate the suggestions from reviewer. As indicated below, we have improved the resolution of **Fig. 5d** and quantified the nuclear accumulation of FABP5 (**As shown below, also see in revised Fig. 5e**).

8. In Fig. 5G, all the bands detected in the nucleus of BMDM are weak and unclear.

Thank you for pointing this out. We have repeated this experiment with modified conditions (**as indicated below, also see in revised Fig. 5h**).

9. In Fig. 7A and 7C, the font size is too small to read.

Thank you for pointing this out. We have made the font size bigger.

10. As mice null for one FABP compensate by increasing expression of other FABPs, careful interpretation of the data is required when using FABPs KO mice. Therefore, the authors need to describe detailed information of FABP5 KO BMDMs (Fig. 7A-D) and *Fabp5^{fl/fl}LysM^{cre}* mice (Fig. 7F and 7G). At least, the expression levels of other FABPs should be shown.

We appreciate this concern suggested by the reviewer. As indicated below (also shown in **Supplementary Fig.4b-d**), In the BMDM from FABP5 KO and *Grx1^{fl/fl}LysM^{cre}* mice, the expression levels of other FABPs did not change significantly in FABP5-deficient macrophages.

11. *The abbreviations should be used consistently throughout the manuscript. For example, ALI or “acute lung inflammation” is mixed together in the text (lines 62, 86, 91 and 449). In addition, Pr-SSG (line 34) and PSSG (line 510), UFA (line 953) and SFA (line 958) were used in Fig. 5 legend, ULCFA and SLCFA were used in the text. They should be revised correctly.*

Thank you for pointing these out. In the revised manuscript, PSSG has been changed to Pr-SSG (**Page 28, Line 567**); ALI has been changed to “acute lung injury” (**Page 24, Line 492**); UFA has been changed to ULCFA (**Figure legends 5a, c**); SFA changed to SLCFA (**Figure legends 5b**).

12. *In line 249, mainly should be changed to abundantly, because FABP5 is also abundantly expressed in keratinocytes, adipocytes, T cells, and cancer cells.*

Thank you for pointing this out. We have corrected it. (**Page 13, Line 262**)

13. *In line 267, “FABP5” should be changed to “Homo sapience FABP5”.*

Thank you for pointing this out. We have corrected it. (**Page 14, Line 284**)

14. *In line 318, ERR α should be added, because the paper concerning FABP5-ERR α crosstalk in the nucleus was published recently (Oncotarget 9, 31753-31770, 2018).*

Thank you for pointing this out. ERR α has been added in the revised manuscript. (**Page 17, Line 344**)

15. *In lines 336-340, mutation of FABP5 (G114R) is found in the diseases accompanied by inflammation? What is the relationship between this mutation and oxidative stress and/or inflammation?*

We appreciate the reviewer for raising this question. FABP5 G114R, a mutation identified from schizophrenia and autism spectrum disorder, enhanced the S-glutathionylation of FABP5. A wide body of evidence has been accumulating for aberrant reactive oxygen species and inflammation in schizophrenia and autism spectrum disorder (Sawa, A. et al., 2016, *Schizophr Res.* 176, 1-2; Pangrazzi, L. et al., 2020, *Int J Mol Sci.* 21, 3293). But the regulatory mechanisms remain unclear.

Data to date indicate that FABP5 is multi-functional protein, plays a key role in lipid-related metabolic processes, cell signaling, cell growth, and differentiation, as well as regulating inflammation. FABP5 is expressed in brain, and altered mRNA expression level of FABP5 was detected in psychiatric illnesses. However, the precise mechanism remains largely unclear. As a intracellular lipid chaperone that bind to fatty acids, FABP5 plays an important role in trafficking fatty acids and related lipids to specific cellular compartments.

Disturbances of lipid metabolism have been implicated in psychiatric illnesses. Recently, fatty acids, particularly ω 3 and ω 6 polyunsaturated fatty acids (PUFAs) have attracted attention in the pathophysiology of schizophrenia and ASD (Peet M., et al., 1995, *J. Psychiatr. Res.* 29, 227–232; Arvindakshan M., 2003, *Schizophr. Res.* 62, 195–204.). As PUFAs are extremely lipophilic molecules, FABPs are needed for their intracellular trafficking. Thus, we speculate that, in psychiatric illnesses, G114R enhanced the S-glutathionylation of FABP5, affecting the lipid metabolism in brain cells, not just the anti-inflammatory ability of macrophages (or microglia). The detailed molecular mechanisms need to be further studied.

16. In line 358, “the fatty acid-binding protein” might be deleted.

Thank you for pointing this out. We have deleted it (**Page 19, Line 391**).

17. In lines 398-405, the part regarding S-glutathionylation of GAPDH might not be necessarily described in Discussion.

Thank you for pointing this out. We have deleted the description of GAPDH-SSG.

18. Materials and Methods is partly incomplete. More detailed information of all the mice, reagents and procedures used in this study should be provided. For example, the experiments regarding PPRE-driven reporter assay, nucleofection, preparation of FABP5 KO BMDM with overexpression of FABP5 C127S and FABP5 WT, ELISA and molecular docking should be described in more detail.

We thank the reviewer for pointing out this issue. When we submitted the manuscript, we choose the **double-blind peer review**. So, the reviewers cannot see the detailed information of mice for the moment. We have carefully checked all relevant information to ensure that it has been included in the submitted manuscript.

In conclusion, although the findings reported here are potentially quite interesting and significant, sufficient mechanistic details and major revision are needed to substantiate the findings at this time.

We greatly appreciate the comments and suggestions from the reviewer. As suggested by the reviewer, we have revised the manuscript and reformatted the figures.

REVIEWERS' COMMENTS

Reviewer #1 (Remarks to the Author):

The authors have addressed almost all the concerns raised from the initial review. However, they have not fully addressed the first comment from the review "It is interesting that some of

the results obtained in this hyperoxic experiment are very similar to those mentioned in the introduction with exposure to cigarette smoke (25) and COPD (27) i.e. reduction in Grx1 expression and decreased GSH:GSSG ratio, could the authors comment in the discussion on whether both hypoxia and hyperoxia would provoke similar responses and why?

I would like to see the authors address this comment, the revised manuscript contains some additional information (lines 411-414) but not whether the mechanisms are similar.

Reviewer #2 (Remarks to the Author):

The authors have done a thorough and effective job of responding to the review of the originally submitted version. I do not have any further major concerns with the revised manuscript. I encourage the authors / journal staff to carefully proofread the current version for minor errors in sentence syntax and other grammatical shortcomings, especially in the revised sections.

Reviewer #3 (Remarks to the Author):

The authors have adequately revised the manuscript, by improving the technical presentation, revising texts, and performing additional experiments. The manuscript remains a highly interesting mechanistic study on the role of a specific s-glutathionylated target, FABP5, in protection from acute lung injury, that uses state of the art approaches.

The authors have performed the in vivo validation experiment that I asked for namely demonstrating FABP5 S-glutathionylation in lung tissues of LPS-exposed Grx1fl/flLysMcre mice.

The main flaw of the paper was the multi-model approach (hypoxia, LPS, and H₂O₂) to define individual pieces of the pathway. This is not incorrect but lends to a patchwork presentation rather than a single focus on one model of inflammation. The authors have now provided adequate justification and transitions to support this approach.

Some of the western blot panels in figs. 5-7 are legible but appear at suboptimal image quality/resolution.

Please ensure that all institutional, mouse, and reagent acquisition details are present in the final submitted manuscript, if they were hidden for review purposes, as noted by the authors.

Reviewer #4 (Remarks to the Author):

I have carefully assessed whether the previous version of the manuscript entitled "S-glutathionylation of fatty acid-binding protein 5 protects against acute lung injury by suppressing inflammation in macrophages" is appropriately revised according to the reviewer's comments.

Overall, I feel that the authors have satisfactorily addressed not only the minor points but also the major issues that the reviewer indicated. According to the reviewer's comments, they have shown new data in the rebuttal and revised manuscript in more detail. Although absolute proof is still lacking for the transcriptional mechanism underlying anti-inflammatory activity mediated by FABP5/PPAR β/δ and other cofactors in macrophages, I feel the data presented are now more compelling on the issue of significance of S-glutathionylation of FABP5.

Ms. No. NCOMMS-20-48677A

Ms. Title: S-glutathionylation of fatty acid-binding protein 5 protects against acute lung injury by suppressing inflammation in macrophages

Point-by-point responses to the reviewers' comments on manuscript

Reviewer #1 (Remarks to the Author):

The authors have addressed almost all the concerns raised from the initial review. However, they have not fully addressed the first comment from the review "It is interesting that some of the results obtained in this hyperoxic experiment are very similar to those mentioned in the introduction with exposure to cigarette smoke (25) and COPD (27) i.e. reduction in Grx1 expression and decreased GSH:GSSG ratio, could the authors comment in the discussion on whether both hypoxia and hyperoxia would provoke similar responses and why? I would like to see the authors address this comment, the revised manuscript contains some additional information (lines 411-414) but not whether the mechanisms are similar.

We thank the reviewer for this insightful suggestion. Both hyperoxia and hypoxic conditions favour the increase of ROS, although the specific mechanisms may be different. The major contributor of hyperoxia-induced ROS is activation of the multiprotein enzyme complex NADPH oxidase. Most ROS are generated in cells by the mitochondrial respiratory chain under hypoxic conditions. Oxidative stress occurs when the production of ROS exceeds their catabolism, and the GSH/GSSG ratio is used as a general marker of oxidative stress. Cells attempt to counteract the oxidant effects and restore the redox balance by activation or silencing of genes encoding defensive enzymes, transcription factors, and structural proteins, such as Grx1, an important component of the endogenous antioxidant defense system.

We have added some comments in the discussion section (**Page 20, Line 433-439**).

Reviewer #2 (Remarks to the Author):

The authors have done a thorough and effective job of responding to the review of the originally submitted version. I do not have any further major concerns with the revised manuscript. I encourage the authors / journal staff to carefully proofread the current version for minor errors in sentence syntax and other grammatical shortcomings, especially in the revised sections.

We greatly appreciate the comments from the reviewer. We have carefully checked

the manuscript and corrected the grammatical errors.

Reviewer #3 (Remarks to the Author):

The authors have adequately revised the manuscript, by improving the technical presentation, revising texts, and performing additional experiments. The manuscript remains a highly interesting mechanistic study on the role of a specific s-glutathionylated target, FABP5, in protection from acute lung injury, that uses state of the art approaches.

The authors have performed the in vivo validation experiment that I asked for namely demonstrating FABP5 S-glutathionylation in lung tissues of LPS-exposed Grx1fl/flLysMcre mice.

The main flaw of the paper was the multi-model approach (hypoxia, LPS, and H₂O₂) to define individual pieces of the pathway. This is not incorrect but lends to a patchwork presentation rather than a single focus on one model of inflammation. The authors have now provided adequate justification and transitions to support this approach.

We greatly appreciate the positive comments from the reviewer.

Some of the western blot panels in figs. 5-7 are legible but appear at suboptimal image quality/resolution.

We appreciate this helpful suggestion. We have improved the resolution of **Fig. 5-8**.

Please ensure that all institutional, mouse, and reagent acquisition details are present in the final submitted manuscript, if they were hidden for review purposes, as noted by the authors.

We have carefully checked all relevant information to ensure that it has been included in the final submitted manuscript.

Reviewer #4 (Remarks to the Author):

I have carefully assessed whether the previous version of the manuscript entitled "S-glutathionylation of fatty acid-binding protein 5 protects against acute lung injury by suppressing inflammation in macrophages" is appropriately revised according to the reviewer's comments.

Overall, I feel that the authors have satisfactorily addressed not only the minor points but also the major issues that the reviewer indicated. According to the reviewer's

comments, they have shown new data in the rebuttal and revised manuscript in more detail. Although absolute proof is still lacking for the transcriptional mechanism underlying anti-inflammatory activity mediated by FABP5/PPAR β/δ and other cofactors in macrophages, I feel the data presented are now more compelling on the issue of significance of S-glutathionylation of FABP5.

We are grateful that you recognize the importance of our work and appreciate your comments.